# Faithful Relational Reasoning with Region-based Embeddings: Expressivity of Convex Coordinate-wise Models

Victor Charpenay [1]    Steven Schockaert [2]

## Abstract

Embedding methods are among the most efficient approaches for learning to reason about relational knowledge. In this paper, we focus on the framework of region-based embeddings, where relations are encoded as geometric regions. The spatial arrangement of these regions allows such models to capture symbolic rules, enabling them to simulate some forms of symbolic reasoning. A crucial consideration is how the regions are parameterized, as this affects which rule bases can be captured. Most methods use convex regions which are defined in terms of coordinate-wise comparisons. This makes them highly efficient, but the implications of this choice have thus far remained unclear. We present a series of results that shed light on this issue, showing that convex coordinate-wise models indeed have important limitations, while at the same time showing that there is still room for pushing the expressivity of existing coordinate-wise models.

## 1. Introduction

Relational reasoning is the task of inferring how different entities are related, based on a given set of relational facts (i.e. a knowledge graph). Relational reasoning is central for predicting missing links in knowledge graphs (Zhu et al., 2021), enables abstract visual reasoning (Webb et al., 2024), is used for learning general policies (Ståhlberg et al., 2025) and for providing guarantees (Debot et al., 2025) in reinforcement learning, and is paramount for neurosymbolic AI more generally (Rocktäschel & Riedel, 2017; Manhaeve et al., 2021). In practice, the rules which govern relational inferences typically need to be learned. Accordingly, a wide

variety of Graph Neural Networks (Zhu et al., 2021; Khalid & Schockaert, 2025; Ståhlberg et al., 2025), Transformer models (Bergen et al., 2021; Webb et al., 2023) and Neurosymbolic approaches (Minervini et al., 2020; Lu et al., 2022; Cheng et al., 2023; Chen et al., 2023) have been proposed for learning to reason in relational domains. Many of these approaches, however, are computationally expensive. For instance, the number of vectors computed by Edge Transformers (Cheng et al., 2023), in each layer, is cubic in the number of entities, which drastically limits their scalability. GNN-based approaches often require a separate forward pass for each link prediction query (Zhu et al., 2021; Khalid & Schockaert, 2025), which can also become infeasible.

When more efficiency is needed, most approaches rely on learning *entity embeddings*, either using a GNN encoder (Schlichtkrull et al., 2018; Vashishth et al., 2020; Chen et al., 2022a; Pavlovic et al., 2025; Morris et al., 2025) or through direct optimization (Bordes et al., 2013; Trouillon et al., 2016; Sun et al., 2019). An entity embedding $v$ is a mapping from entities $e$ to vectors $v(e) \in \mathbb{R}^n$. They are typically learned together with a scoring function $s_r : \mathbb{R}^n \times \mathbb{R}^n \to \mathbb{R}$ for each relation $r$, such that $s_r(v(e), v(f))$ reflects the plausibility that the relational fact $r(e, f)$ holds. Entity embeddings can be efficiently learned, even for knowledge graphs (KGs) involving tens of millions of entities (Zheng et al., 2020), although this typically comes at the expense of lower accuracy. It remains unclear, however, whether this reflects particular weaknesses of current approaches or inherent limitations in the expressivity of embedding models. Indeed, while there has been extensive work on the expressivity of Graph Neural Networks (Morris et al., 2019; Barceló et al., 2020; Huang et al., 2023) and Transformers (Müller et al., 2024), only a few previous works have studied embedding methods from this point of view (Gutiérrez-Basulto & Schockaert, 2018; Bourgaux et al., 2024).

To formally study the expressivity of embedding models, we need to consider a threshold $\theta_r$ such that $r(a, b)$ is predicted to hold iff $s_r(v(a), v(b)) \geq \theta_r$. The representation of a relation $r$ can then be viewed as a region $A_r \subseteq \mathbb{R}^{2n}$:

$$A_r = \{\mathbf{x} \oplus \mathbf{y} \mid \mathbf{x}, \mathbf{y} \in \mathbb{R}^n, s_r(\mathbf{x}, \mathbf{y}) \geq \theta_r\}$$

where we write $\oplus$ for vector concatenation. The spatial

---

[1]Mines Saint-Etienne, UMR 6158 LIMOS, Saint-Étienne, France [2]Cardiff University, Cardiff, UK. Correspondence to: Victor Charpenay <victor.charpenay@emse.fr>, Steven Schockaert <schockaerts1@cardiff.ac.uk>.

*Proceedings of the 43rd International Conference on Machine Learning*, Seoul, South Korea. PMLR 306, 2026. Copyright 2026 by the author(s).

configuration of these regions determines what rules are captured. As a simple example, if $A_r \subseteq A_s$, then whenever a fact $r(a, b)$ is predicted, we also have that $s(a, b)$ is predicted. Such an embedding thus captures the rule $r(X, Y) \to s(X, Y)$, where $X$ and $Y$ denote variables.

A central question is whether, for a given rule base $\mathcal{P}$, we can always find region-based relation embeddings such that all rules that can be inferred from $\mathcal{P}$ (in a bounded number of inference steps) are captured, and *only* these rules. Pavlovic et al. (2025) introduced a model which has this property for the important case of closed path rules. However, their model relies on *cross-coordinate comparisons*, i.e. to determine whether $\mathbf{x} \oplus \mathbf{y} \in A_r$, we may need to compare any coordinate of $\mathbf{x}$ with any coordinate of $\mathbf{y}$. In contrast, most popular models are *coordinate-wise*, i.e. the $i^{\text{th}}$ coordinate of $\mathbf{x}$ is only compared with the $i^{\text{th}}$ coordinate of $\mathbf{y}$. Such coordinate-wise models are more efficient, and they typically perform better, as the larger number of parameters needed to learn cross-coordinate comparisons can lead to overfitting. Unfortunately, only few results are known about the expressivity of coordinate-wise models. Most notably, Charpenay & Schockaert (2024) showed that a model based on axis-aligned octagons is capable of capturing sets of so-called regular rule bases, but these octagons are not capable of capturing more general sets of closed path rules.

In this paper, we study whether coordinate-wise models, in general, have limitations when it comes to capturing closed path rules. Our main findings can be summarized as follows:

- Without further constraints, we find that arbitrary sets of closed path rules can be faithfully captured using coordinate-wise models. We therefore focus our main analysis on *convex* coordinate-wise regions.

- We find that convex coordinate-wise models can capture arbitrary acyclic sets of closed path rules, but only when models are allowed to use different regions for learning the embeddings and for evaluating queries.

- Convex coordinate-wise models are inherently limited when looking beyond closed path rules. In particular, certain sets of closed path rules can only be captured by additionally capturing spurious semantic dependencies.

- We empirically show that existing models often fail to faithfully capture rule bases in practice, even when they should be capable of doing so in theory.

## 2. Background

**Preliminaries** Let $\mathcal{E}$ and $\mathcal{R}$ be finite sets of entities and relations, respectively. A *relational fact* is an expression of the form $r(e, f)$, encoding that entity $e$ has relationship $r$ to entity $f$. A set of relational facts $\mathcal{G}$ is called a *knowledge graph* (KG). Following previous work on the expressivity of region-based embeddings (Pavlovic & Sallinger, 2023; Charpenay & Schockaert, 2024; Pavlovic et al., 2025), we focus on closed path rules in this paper. This is motivated by the central importance of such rules for KG reasoning, together with the fact that capturing such rules with embeddings is particularly challenging. Indeed, existing models (Abboud et al., 2020; Pavlovic & Sallinger, 2023; Charpenay & Schockaert, 2024) are already capable of capturing arbitrary sets of intersection and hierarchy rules, for instance, but capturing arbitrary sets of closed path rules with coordinate-wise models has thus far proven elusive.

A *closed path rule* is an expression of the form $r_1(X_1, X_2) \wedge r_2(X_2, X_3) \wedge ... \wedge r_k(X_k, X_{k+1}) \to s(X_1, X_{k+1})$. Such a rule encodes that whenever $r_1(e_1, e_2), ..., r_k(e_k, e_{k+1})$ hold, for some entities $e_1, ..., e_{k+1}$, then $s(e_1, e_{k+1})$ must also hold. For brevity, we will write such a rule as $r_1 \circ ... \circ r_k \subseteq s$. A set of closed path rules (i.e. a rule base) is called acyclic if there exists an enumeration $r_1, ..., r_m$ of the relations in $\mathcal{P}$ such that for every rule $r_{i_1} \circ ... \circ r_{i_k} \subseteq r_{i_{k+1}}$, we have that $\max(i_1, ..., i_k) < i_{k+1}$. A rule base which is not acyclic is said to have cyclic dependencies.

Let $\mathcal{P}$ be a set of closed path rules. We write $\mathcal{P} \cup \mathcal{G} \models_k r(e, f)$ to denote that $r(e, f)$ can be derived from the facts in $\mathcal{G}$ and the rules in $\mathcal{P}$ in at most $k$ inference steps. Formally, we have $\mathcal{P} \cup \mathcal{G} \models_0 r(e, f)$ if $r(e, f) \in \mathcal{G}$. For $k > 0$, we have $\mathcal{P} \cup \mathcal{G} \models_k r(e, f)$ if there is a rule $r_1 \circ ... \circ r_m \subseteq r$ in $\mathcal{P}$ such that $\mathcal{P} \models_{k_1} r_1(e, e_1), ..., \mathcal{P} \models_{k_m} r_m(e_{m-1}, f)$ for some entities $e_1, ..., e_{m-1}$, such that $k \geq k_1 + ... + k_m + 1$. We write $\mathcal{P} \models_k r_1 \circ ... \circ r_m \subseteq r$ to denote $\mathcal{P} \cup \{r_1(e_1, e_2), ..., r_m(e_m, e_{m+1})\} \models_k r(e_1, e_{m+1})$. We say that $\mathcal{P} \cup \mathcal{G}$ entails $r(e, f)$, written $\mathcal{P} \cup \mathcal{G} \models r(e, f)$ if there is some $k \in \mathbb{N}$ such that $\mathcal{P} \cup \mathcal{G} \models_k r(e, f)$, and similar for $\mathcal{P} \models r_1 \circ ... \circ r_m \subseteq r$.

**Region-based Embeddings** An ($n$-dimensional) entity embedding is a mapping $v : \mathcal{E} \to \mathbb{R}^n$. An ($n$-dimensional) relation embedding is a mapping $\eta : \mathcal{R} \to 2^{\mathbb{R}^{2n}}$. A knowledge graph embedding is a pair $(v, \eta)$, with $v$ an entity embedding and $\eta$ a relation embedding of the same dimensionality. We say that $(v, \eta)$ captures a relational fact $r(e, f)$ iff $v(e) \oplus v(f) \in \eta(r)$. We also write this as $(v, \eta) \models r(e, f)$. If $(v, \eta)$ captures every relational fact in a knowledge graph $\mathcal{G}$ we also say that $(v, \eta)$ captures $\mathcal{G}$. A relation embedding is called *convex* if $\eta(r)$ is a convex region for every $r \in \mathcal{R}$.

**Example 2.1.** *In TransE (Bordes et al., 2013), embeddings are learned such that positive and examples are separated by the score $s_r(v(e), v(f)) = -d(v(e) + \mathbf{r}, v(f))$, where $\mathbf{r} \in \mathbb{R}^n$ encodes the relation $r$. A region-based relation embedding is then obtained by fixing a threshold $\theta_r \in \mathbb{R}$:*

$$\eta(r) = \{\mathbf{x} \oplus \mathbf{y} \mid s_r(\mathbf{x}, \mathbf{y}) \geq \theta_r\}$$

*Note that for the analysis in this paper, the scoring function*

**Symbolic reasoning**

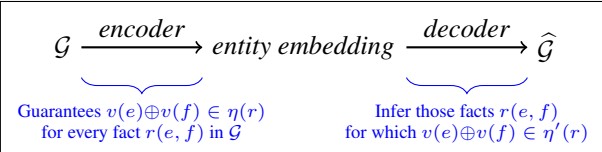

**Region-based embeddings**

*Figure 1.* Schematic overview of how region-based embeddings simulate symbolic reasoning.

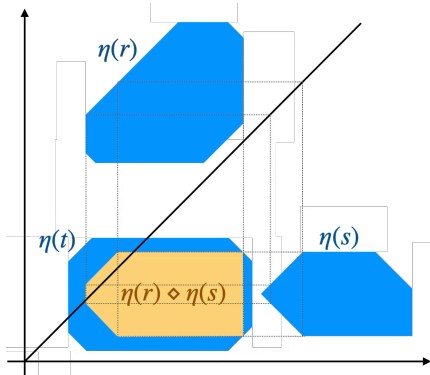

*Figure 2.* Region embeddings capture the rule $r \circ s \subseteq t$.

*that was used for learning the embeddings is irrelevant, as our focus is on the regularities that are captured by the resulting regions $\eta(r)$. Some models directly learn regions. For instance, ReshufflE (Pavlovic et al., 2025) learns relation embeddings of the following form ($\mathbf{A_r} \in \mathbb{R}^{n \times n}$):*

$$\eta(r) = \{\mathbf{x} \oplus \mathbf{y} \mid \max(\mathbf{A_r x}, \mathbf{y}) = \mathbf{y}\} \quad (1)$$

A relation embedding $\eta$ is called coordinate-wise if all regions $\eta(r)$ are of the following form:

$$\{(x_1, ..., x_n, y_1, ..., y_n) \mid \forall 1 \leq i \leq n . (x_i, y_i) \in \mathsf{A}_r^i\}$$

where $\mathsf{A}_r^1, ..., \mathsf{A}_r^n \subseteq \mathbb{R}^2$. A coordinate-wise relation embedding $\eta$ is thus determined by two-dimensional regions $\mathsf{A}_r^i$. We write $\eta_i : \mathcal{R} \to 2^{\mathbb{R}^2}$ for the mapping that defines these two-dimensional regions, i.e. $\eta_i(r) = \mathsf{A}_r^i$.

**Example 2.2.** *A variant of TransE can be defined by considering two-dimensional regions of the following form:*

$$\eta_i(r) = \{(x, y) \in \mathbb{R}^2 \mid |y - x - r_i| \leq \theta_{i,r}\}$$

*ExpressivE (Pavlovic & Sallinger, 2023) learns two-dimensional regions that correspond to parallelograms, while Charpenay & Schockaert (2024) introduced a coordinate-wise model which uses axis-aligned octagons.*

## 3. Reasoning with Region-based Embeddings

**Encoder-Decoder View of Reasoning**   A rule base $\mathcal{P}$ defines a closure operator which maps KGs $\mathcal{G}$ onto their corresponding deductive closure $\widehat{\mathcal{G}} = \{r(e, f) \mid \mathcal{P} \cup \mathcal{G} \models r(e, f)\}$. We can think of KG embedding models as approximating this symbolic reasoning process in two steps: first an *encoder* is used to learn an entity embedding $v$ from the given knowledge graph $\mathcal{G}$ and then a *decoder* is used to infer the corresponding set of relational facts. In the region-based framework, the encoder and decoder are parameterized by a relation embedding $\eta$. Relation embeddings thus serve as the counterpart of the rule base $\mathcal{P}$, capturing the semantic regularities of the domain. The encoder can take many

forms but essentially aims to learn an entity embedding $v$ such that $v(e) \oplus v(f) \in \eta(r)$ holds for every relational fact $r(a, b)$ from the given KG. The decoder then uses this entity embedding to infer all relational facts $r(e, g)$ for which $v(e) \oplus v(f) \in \eta'(r)$. In practice, the parameters of the encoder and decoder are often tied, in which case $\eta = \eta'$. This process is illustrated in Figure 1.

**Example 3.1.** *ReshufflE uses relation embeddings of the form (1). An associated GNN encoder can be defined as:*

$$\mathbf{y}^{(l+1)} = \max(\mathbf{y}^{(l)}, \max\{\mathbf{A_r x} \mid r(x, y) \in \mathcal{G}\})$$

*where $\mathbf{y}^{(l)}$ denotes the representation of entity $y$ in layer $l$ and the input embeddings $\mathbf{y}^{(0)}$ are randomly initialized (Pavlovic et al., 2025). The decoder uses the same parameters for determining the inferred triples, i.e.:*

$$\widehat{\mathcal{G}} = \{r(x, y) \mid \max(\mathbf{A_r x}^{(L)}, \mathbf{y}^{(L)}) = \mathbf{y}^{(L)}\}$$

*where we write $L$ for the final layer of the GNN encoder. In ReshufflE, a GNN with max-pooling can be used because the regions have a particularly simple form. For other types of regions, GNN encoders with sum-pooling can be used, by essentially using message-passing to simulate gradient-based optimization steps (Chen et al., 2022b).*

In models such as ExpressivE (Pavlovic & Sallinger, 2023) and Octagons (Charpenay & Schockaert, 2024), the entity embedding $v$ and relation embedding $\eta$ are jointly learned by optimizing a loss function. In such *transductive* settings, the encoder part of the pipeline is thus implicit.

**Capturing Rule Bases**   We now define what it means for a relation embedding $\eta$ to capture a rule base $\mathcal{P}$. We first consider what it means to capture a single rule.

**Definition 3.2.** *Let $\eta$ and $\eta'$ be $n$-dimensional relation embeddings. We say that the pair $(\eta, \eta')$ captures $r_1 \circ ... \circ r_k \subseteq s$ if for any $\mathbf{x_1}, ..., \mathbf{x_{k+1}} \in \mathbb{R}^n$ such that $\mathbf{x_1} \oplus \mathbf{x_2} \in \eta(r_1), ..., \mathbf{x_k} \oplus \mathbf{x_{k+1}} \in \eta(r_k)$ we have $\mathbf{x_1} \oplus \mathbf{x_{k+1}} \in \eta'(s)$.*

**Example 3.3.** *Consider the two-dimensional relation embedding $\eta$ depicted in Figure 2. For $\mathbf{x}, \mathbf{y}, \mathbf{z} \in \mathbb{R}^2$, if $\mathbf{x} \oplus \mathbf{y} \in \eta(r)$ and $\mathbf{y} \oplus \mathbf{z} \in \eta(s)$ then it follows that $\mathbf{x} \oplus \mathbf{z}$ belongs to the orange region, and thus in particular also $\mathbf{x} \oplus \mathbf{z} \in \eta(t)$. We thus have that $\eta$ captures the rule $r \circ s \subseteq t$.*

We will refer to the pair $(\eta, \eta')$ as an accordant relation embedding if $\eta = \eta'$. If an accordant relation embedding $(\eta, \eta)$ captures a rule, then we simply say that $\eta$ captures that rule. For the ease of presentation, we define:

$$\mathsf{A} \diamond \mathsf{B} = \{\mathbf{x} \oplus \mathbf{z} \mid \exists \mathbf{z} \in \mathbb{R}^n . \mathbf{x} \oplus \mathbf{y} \in \mathsf{A} \wedge \mathbf{y} \oplus \mathbf{z} \in \mathsf{B}\}$$

where $\mathsf{A}, \mathsf{B} \subseteq \mathbb{R}^{2n}$. We have that $(\eta, \eta')$ captures $r_1 \circ ... \circ r_k \subseteq s$ iff $\eta(r_1) \diamond ... \diamond \eta(r_k) \subseteq \eta'(s)$.

Let $\mathcal{P}$ be a set of closed path rules. It is straightforward to construct relation embeddings that capture every rule in $\mathcal{P}$ (e.g. by defining $\eta(r) = [0,1] \times [0,1]$ for every relation $r$). The main challenge is to ensure that *only* those rules that are entailed by $\mathcal{P}$ are captured. For instance, a region-based model based on TransE can capture the rule $r_1 \circ r_2 \subseteq s$, but it cannot do so without also capturing the rule $r_2 \circ r_1 \subseteq s$.

**Definition 3.4.** *Let $\mathcal{P}$ be a set of closed path rules, and let $\eta$ and $\eta'$ be relation embeddings. We say that $(\eta, \eta')$ faithfully captures $\mathcal{P}$ if for any relations $r_1, ..., r_k, s$ we have:*

$$(\mathcal{P} \models r_1 \circ ... \circ r_k \subseteq s) \Leftrightarrow (\eta(r_1) \diamond ... \diamond \eta(r_k) \subseteq \eta'(s))$$

*Similarly, we say that $(\eta, \eta')$ faithfully captures $\mathcal{P}$ up to depth $m$ if for any relations $r_1, ..., r_k, s$ we have:*

$$(\mathcal{P} \models_m r_1 \circ ... \circ r_k \subseteq s) \Leftrightarrow (\eta(r_1) \diamond ... \diamond \eta(r_k) \subseteq \eta'(s))$$

Ideally, we want to capture reasoning up to any depth. However, for rule bases with cyclic dependencies, this may not always be possible, even for models with cross-coordinate comparisons (Pavlovic et al., 2025).

# 4. Convexity Limits Expressivity

As the next result reveals, when convexity is not required, coordinate-wise relation embeddings are capable of faithfully capturing arbitrary sets of closed path rules.

**Proposition 4.1.** *Let $\mathcal{P}$ be a set of closed path rules. There exists a one-dimensional coordinate-wise relation embedding $\eta$ which faithfully captures $\mathcal{P}$.*

The proof of Proposition 4.1 relies on a construction in which each relation $r$ is represented by a union of lines, with one line for each composition $r_1 \circ ... \circ r_k$ satisfying $\mathcal{P} \models r_1 \circ ... \circ r_k \subseteq r$. Unfortunately, this means that for rule bases with cyclic dependencies, some regions consist of a countably infinite number of lines. Even for acyclic rule bases, the number of lines in a given region can be

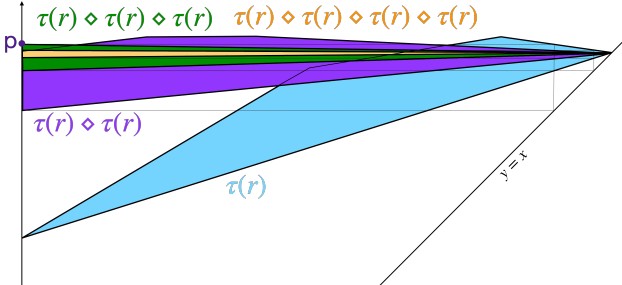

*Figure 3.* The point $p$ belongs to $\tau(r) \diamond \tau(r) \diamond \tau(r)$ but not to the convex hull of $\tau(r)$, $\tau(r) \diamond \tau(r)$ and $\tau(r) \diamond \tau(r) \diamond \tau(r) \diamond \tau(r)$.

exponential in the number of relations that appear in $\mathcal{P}$. Hence, while Proposition 4.1 establishes that capturing sets of closed path rules is possible with coordinate-wise models, this may not reflect the expressivity of models that can actually be learned. In practice, we need regions $\mathsf{A}$ that allow us to efficiently test whether $v(e) \oplus v(f) \in \mathsf{A}$. Furthermore, to enable gradient-based learning, we need to be able to assess some kind of distance between $v(e) \oplus v(f)$ and $\mathsf{A}$, in cases where $v(e) \oplus v(f) \notin \mathsf{A}$. A natural requirement is then to only allow *convex* regions. As the following result shows, however, imposing convexity fundamentally restricts the expressivity of coordinate-wise models.

**Proposition 4.2.** *Let $\mathcal{P}$ consist of the following rules:*

$$r_1 \subseteq r_2 \qquad\qquad r_1 \circ r_1 \circ r_1 \subseteq r_2$$
$$r_1 \circ r_1 \subseteq r_3 \qquad\qquad r_1 \circ r_1 \circ r_1 \subseteq r_3$$
$$r_1 \subseteq r_4 \qquad\qquad r_2 \circ r_2 \subseteq r_4$$
$$r_3 \circ r_3 \subseteq r_4$$

*If $\eta$ is a convex coordinate-wise relation embedding capturing every rule in $\mathcal{P}$, then $\eta$ also captures $r_1 \circ r_1 \circ r_1 \subseteq r_4$.*

This shows that some rule bases cannot be faithfully captured by *accordant* convex coordinate-wise relation embeddings (even without the presence of cyclic dependencies).

# 5. Capturing Rule Bases with Polygons

Following the negative results from Section 4, we now establish two positive results: (i) we show that *non-accordant* relation embeddings can faithfully capture rule bases up to any given depth $m$ and (ii) we establish sufficient conditions under which *accordant* relation embeddings can do this.

To capture arbitrary sets of closed path rules (up to some depth $m$), we need regions which are expressive enough to ensure that each relational composition (up to a given length) can be distinguished from all other relational compositions. More precisely, let $(r_1, ..., r_l)$ be a tuple of relations and let $\mathcal{S}$ be any set of such tuples, not containing $(r_1, ..., r_l)$ itself. We need to be able to ensure that $\eta(r_1) \diamond ... \diamond \eta(r_l)$ contains

a point $p$ which does not belong to the following region:

$$\text{CH}\left(\bigcup\{\eta(s_1) \diamond ... \diamond \eta(s_p) \,|\, (s_1, ..., s_p) \in \mathcal{S}\}\right)$$

where CH denotes the convex hull operator. The results in this section hinge on the fact that we can always construct convex polygons that have this property (where the number of vertices is determined by the length of the relational compositions involved). Figure 3 illustrates the main idea for a single relation $r$. The point $p$ belongs to the region $\eta(r) \diamond \eta(r) \diamond \eta(r)$ but not to the convex hull of $\eta(r)$, $\eta(r) \diamond \eta(r)$ and $\eta(r) \diamond \eta(r) \diamond \eta(r) \diamond \eta(r)$. This construction allows us, for instance, to design relation embeddings which capture the rules $r \subseteq s$, $r \circ r \subseteq s$ and $r \circ r \circ r \circ r \subseteq s$, without capturing $r \circ r \circ r \subseteq s$. We have the following result.

**Proposition 5.1.** *Let $\mathcal{P}$ be a set of closed path rules and $m \in \mathbb{N}$. There is a convex coordinate-wise relation embedding $(\eta, \eta')$ which faithfully captures $\mathcal{P}$ up to depth $m$.*

We know from Proposition 4.2 that this result cannot be generalized to accordant relation embeddings. Unfortunately, however, non-accordant embeddings have some limitations in practice. For instance, standard strategies for learning KG embeddings, which optimize a loss based on some scoring function, can only be used for learning accordant embeddings. Moreover, non-accordant embeddings are also unsuitable when compositional generalization is needed. For instance, suppose the training data contains sufficient evidence to capture the rules $r_1 \circ r_2 \subseteq s_1$ and $s_1 \circ s_2 \subseteq t$. Accordant relation embeddings capturing these rules will also capture the rule $r_1 \circ r_2 \circ s_2 \subseteq t$ (which is logically entailed by the former two), but for non-accordant relation embeddings this is not guaranteed. While accordant relation embeddings cannot faithfully capture arbitrary sets of closed path rules, there are still important special cases for which this is possible. Let us consider the following definitions.

**Definition 5.2.** *Let $\mathcal{P}$ be a set of closed path rules. A relation $r$ is called* regular *in $\mathcal{P}$ if for every rule $r_1 \circ ... \circ r_k \subseteq r$ entailed by $\mathcal{P}$, except for $r \subseteq r$, the relations $r_1, ..., r_k, r$ are all distinct (Charpenay & Schockaert, 2024).*

**Definition 5.3.** *Let $\mathcal{P}$ be a set of closed path rules, with $\mathcal{R}$ the set of relations occurring in $\mathcal{P}$. A weighting $\omega : \mathcal{R} \to \mathbb{N} \setminus \{0\}$ is said to* balance *the relation $r$ if $\omega(r_1) + ... + \omega(r_k) = \omega(r)$ for every rule $r_1 \circ ... \circ r_k \subseteq r$ entailed by $\mathcal{P}$.*

We already know from Charpenay & Schockaert (2024) that octagons can be used to capture rule bases in which every relation is regular. We now generalize this result as follows.

**Proposition 5.4.** *Let $\mathcal{P}$ be an acyclic set of closed path rules, with $\mathcal{R}$ the set of relations occurring in $\mathcal{P}$. Assume that for every $r \in \mathcal{R}$ at least one of the following conditions is satisfied:*

- *$r$ is regular in $\mathcal{P}$;*

- *there exists a weighting $\omega_r : \mathcal{R} \to \mathbb{N} \setminus \{0\}$ such that for every rule $r_1 \circ ... \circ r_k \subseteq r$ entailed by $\mathbb{P}$ it holds that $\omega_r$ balances $r_1, ..., r_k$.*

*Then there exists a convex coordinate-wise relation embedding $\eta$ which faithfully captures $\mathcal{P}$.*

Note that the second condition only requires that the relations $r_1, ..., r_k$ are balanced, not the relation $r$ itself. For instance, if $\mathcal{P}$ is such that any relation which appears in the head of a rule never appears in the body of a rule, then the second condition is always satisfied. Furthermore note that a different weighting $\omega_r$ can be used for each relation $r$.

**Example 5.5.** *Let $\mathcal{P}$ consist of the following rules:*

$$
\begin{array}{ll}
r_1 \circ r_2 \subseteq s_1 & r_1 \subseteq s_1 \\
s_1 \circ s_2 \subseteq t_1 & r_1 \circ r_1 \subseteq s_3 \\
r_2 \subseteq s_3 & s_3 \circ s_3 \subseteq t_2 \\
s_4 \circ s_4 \subseteq t_3 & s_4 \subseteq t_3
\end{array}
$$

*Then $\mathcal{P}$ satisfies the conditions of Proposition 5.4. Indeed, the relations $r_1, r_2, s_2, s_4$ do not appear in the head of any rule, and thus trivially satisfy the conditions. The relations $s_1$ and $t_1$ are regular. The relation $s_3$ satisfies the second condition, as it only appears in the head of rules with $r_1$ and $r_2$ in the body. As $r_1$ and $r_2$ do not appear in the head of any rule, they are trivially balanced. The same holds for $t_3$. Note that while $t_3$ itself cannot be balanced, it is sufficient for $s_4$ to be balanced. Finally, $t_2$ satisfies the second condition, since $r_2$ is trivially balanced and $s_3$ is balanced by any weighting $\omega$ satisfying $\omega(s_3) = \omega(r_2) = 2\omega(r_1)$.*

**Example 5.6.** *The rule base $\mathcal{P}$ from Proposition 4.2 does not satisfy the conditions of Proposition 5.4. Indeed, since $r_4$ is not regular, the only way to satisfy these conditions would be to construct a weighting $\omega$ that balances $r_1$, $r_2$ and $r_3$. This would require $\omega(r_1) = 3\omega(r_1) = \omega(r_2)$ and $2\omega(r_1) = 3\omega(r_1) = \omega(r_3)$, which cannot be satisfied, given that only strictly positive weights are allowed.*

A general procedure for checking the conditions from Proposition 5.4 is provided in Appendix F.

## 6. Beyond Closed Path Rules

If a relation embedding faithfully captures a rule base, we know that no unwanted closed path rules are captured (see Definition 3.4). However, such a relation embedding might still capture other unwanted semantic dependencies (which cannot be expressed as closed path rules). Unfortunately, as the next result reveals, it is not always possible to avoid this with convex coordinate-wise models, even for non-accordant embeddings and acyclic rule bases.

**Proposition 6.1.** *Suppose the convex coordinate-wise rela-*

*tion embedding $(\eta, \eta')$ captures the following rules:*

$$r_1 \circ s_1 \subseteq t \qquad r_2 \circ s_2 \subseteq t \qquad r_3 \circ s_3 \subseteq t$$

*Let $v$ be an entity embedding such that $(v, \eta)$ captures the following relational facts:*

$$
\begin{array}{llll}
r_1(a, b_{12}) & s_2(b_{12}, c) & r_1(a, b_{13}) & s_3(b_{13}, c) \\
r_2(a, b_{21}) & s_1(b_{21}, c) & r_2(a, b_{23}) & s_3(b_{23}, c) \\
r_3(a, b_{31}) & s_1(b_{31}, c) & r_3(a, b_{32}) & s_2(b_{32}, c)
\end{array}
$$

*Then $(v, \eta')$ captures the fact $t(a, c)$.*

A closely related question is whether it is possible to faithfully capture more general classes of rule bases (w.r.t. a straightforward generalization of Definition 3.4). For instance, in addition to closed path rules, previous work has considered intersection rules of the following form:

$$r_1(X, Y) \wedge r_2(X, Y) \rightarrow s(X, Y)$$

which we can abbreviate as $r_1 \cap r_2 \subseteq s$. Rules of the form $r \subseteq s$ are often called hierarchy rules in this setting (which we consider to be special cases of closed path rules in this paper). Most region-based embedding models can faithfully capture arbitrary sets of intersection and hierarchy rules (Abboud et al., 2020; Pavlovic & Sallinger, 2023; Charpenay & Schockaert, 2024; Pavlovic et al., 2025).

We may also wonder about rules that mix compositions and intersections. For instance, we can consider a rule like $r_1(X, Y_1) \wedge r_2(Y_1, Z) \wedge r_3(X, Y_2) \wedge r_4(Y_2, Z) \rightarrow s(X, Z)$, which we can abbreviate as $(r_1 \circ r_2) \cap (r_3 \circ r_4) \subseteq s$. Unfortunately, we already know from Proposition 6.1 that relation embeddings cannot faithfully capture sets of such generalized rules. In particular, it follows from this proposition that any relation embedding capturing the three given rules will also capture the rule $(r_1 \circ s_2) \cap (r_1 \circ s_3) \cap (r_2 \circ s_1) \cap (r_2 \circ s_3) \cap (r_3 \circ s_1) \cap (r_3 \circ s_2) \subseteq t$.

Another limitation arises when some relations are required to be symmetric. Let us define, for $A \subseteq \mathbb{R}^2$:

$$inv(A) = \{(y, x) \mid (x, y) \in A\}$$

If $r$ is a symmetric relation, then we should have $\eta_i(r) = inv(\eta_i(r))$ for every coordinate $i$. If this condition is satisfied, then we say that $r$ is symmetric in $\eta$. Unfortunately, imposing symmetry in this way significantly limits the expressivity of coordinate-wise embeddings. Let us write $r^{\uparrow i}$ for a composition of $r$ with itself of length $i$, i.e. $r^{\uparrow 1} = r$ and for $i > 1$ we define $r^{\uparrow i} = r^{\uparrow (i-1)} \circ r$.

**Proposition 6.2.** *Let $\eta$ be a coordinate-wise relation embedding such that $r$ is symmetric in $\eta$. Suppose $(\eta, \eta')$ captures the rule $r^{\uparrow i} \subseteq s$ for some $i \geq 3$. Then $(\eta, \eta')$ also captures the rule $r^{\uparrow (i-2)} \subseteq s$.*

Note that this result holds even if convexity is not imposed.

*Table 1.* Rule sets of the FAIRE benchmark.

| Dataset | Rule base $\mathcal{P}^+$ | Distractors $\mathcal{P}^-$ |
|---|---|---|
| PERM2 | $r \circ s \subseteq t$ | $s \circ r \subseteq t$ |
| PERM3 | $r \circ s \circ t \subseteq u$ | $r \circ t \circ s \subseteq u, \quad s \circ r \circ t \subseteq u,$ $s \circ t \circ r \subseteq u, \quad t \circ r \circ s \subseteq u,$ $t \circ s \circ r \subseteq u$ |
| MIX2 | $r_1 \circ s_1 \subseteq t,$ $r_2 \circ s_2 \subseteq t$ | $r_2 \circ s_1 \subseteq t,$ $r_1 \circ s_2 \subseteq t$ |
| MIX3 | $r_1 \circ s \circ t_1 \subseteq u,$ $r_2 \circ s \circ t_2 \subseteq u$ | $r_1 \circ s \circ t_2 \subseteq u,$ $r_2 \circ s \circ t_1 \subseteq u$ |
| REP12 | $r \circ r \circ s \subseteq t,$ $r \circ s \subseteq t$ | $s \subseteq t,$ $r \circ r \circ r \circ s \subseteq t$ |
| REP13 | $r \circ s \subseteq t,$ $r \circ r \circ r \circ s \subseteq t$ | $r \circ r \circ s \subseteq t$ |
| COMB | $r \circ s \circ r \subseteq t$ | $r \subseteq t, \quad s \subseteq t, \quad r \circ r \subseteq t,$ $r \circ r \circ s \subseteq t, \quad s \circ r \circ r \subseteq t$ |

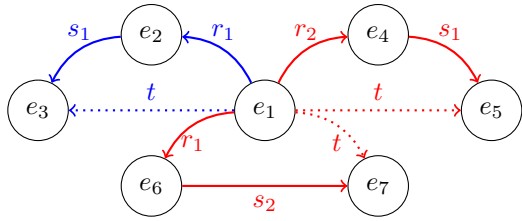

*Figure 4.* Instantiation from the MIX2 dataset. Solid blue edges depict facts from $\mathcal{A}_{\mathcal{P}^+}$; the dotted blue edge depicts a fact from $\mathcal{I}_{\mathcal{P}^+}$; the solid red edges depict facts from $\mathcal{A}_{\mathcal{P}^-}$; and the dotted red edges depict facts from $\mathcal{I}_{\mathcal{P}^-}$. Models that faithfully capture the rule base $\mathcal{P}^+$ should infer $t(e_1, e_3)$ but not $t(e_1, e_5)$ or $t(e_1, e_7)$.

## 7. Experimental Results

Despite their theoretical limitations, convex coordinate-wise models have obtained competitive results on real-world KGs (Pavlovic & Sallinger, 2023). However, complex constructs, such as the rule base introduced in Proposition 4.2, are rarely needed for such KGs, making them ill-suited for assessing whether models are prone to making spurious inferences. Other applications, for instance in the context of reinforcement learning (Ståhlberg et al., 2025), are more reasoning intensive, but they do not readily allow us to assess the reasoning abilities of models in a systematic way. To support further work on this topic, we therefore designed a synthetic rule reasoning benchmark. This benchmark, made of seven datasets, intends to evaluate the ability of a model to faithfully capture a rule base by explicitly considering a number of distractors, i.e. rules that the model should not capture.

Table 1 describes the different datasets from the benchmark, which we call FAIRE (Faithful Reasoning). Each of the datasets in FAIRE involves a rule base $\mathcal{P}^+$ and a set of distractor rules $\mathcal{P}^-$, with all rules sharing the same head. PERM2, PERM3, MIX2 and MIX3 only involve regular

*Table 2.* Hits@1 results for FAIRE, averaged across 20 runs, with the minimum and maximum values shown in parentheses.

|  | PERM2 | PERM3 | MIX2 | MIX3 | REP12 | REP13 | COMB |
|---|---|---|---|---|---|---|---|
| Octagons | **1.00** (1.00–1.00) | 0.58 (0.00–1.00) | 0.47 (0.00–1.00) | 0.25 (0.03–0.48) | 0.42 (0.11–0.58) | 0.60 (0.35–1.00) | 0.17 (0.00–0.50) |
| ExpressivE | 0.99 (0.92–1.00) | **0.97** (0.70–1.00) | **0.94** (0.82–1.00) | **0.71** (0.24–1.00) | **0.58** (0.38–0.85) | **0.69** (0.30–1.00) | 0.32 (0.15–0.61) |
| Polygons (2) | 0.96 (0.44–1.00) | 0.60 (0.26–0.94) | 0.77 (0.29–1.00) | 0.50 (0.23–0.96) | 0.37 (0.20–0.55) | 0.53 (0.41–0.62) | 0.28 (0.08–0.82) |
| Polygons (4) | 0.99 (0.80–1.00) | 0.75 (0.20–1.00) | 0.76 (0.20–1.00) | 0.45 (0.22–0.73) | 0.40 (0.24–0.54) | 0.62 (0.44–0.87) | 0.35 (0.14–0.65) |
| Polygons (6) | **1.00** (1.00–1.00) | 0.69 (0.00–1.00) | 0.78 (0.23–1.00) | 0.57 (0.35–0.82) | 0.35 (0.14–0.61) | 0.51 (0.20–0.85) | 0.40 (0.03–0.99) |
| Polygons (8) | 0.97 (0.60–1.00) | 0.71 (0.22–1.00) | 0.78 (0.26–0.98) | 0.44 (0.22–0.76) | 0.42 (0.16–0.62) | 0.63 (0.34–0.90) | 0.34 (0.02–0.69) |
| Polygons (10) | **1.00** (1.00–1.00) | 0.89 (0.56–1.00) | 0.74 (0.29–0.97) | 0.47 (0.17–0.92) | 0.41 (0.24–0.68) | 0.53 (0.38–0.81) | **0.44** (0.08–0.88) |
| Random | 0.50 | 0.17 | 0.33 | 0.33 | 0.33 | 0.50 | 0.17 |

relations, and can thus in theory be captured by existing region-based models (see Section 8). For every rule base, we have that the relations which appear in the body of a rule are trivially balanced, meaning that the conditions of Proposition 5.4 are satisfied, hence they can all be faithfully captured by accordant polygon-based embeddings.

Each dataset is associated with four KGs: $\mathcal{A}_{\mathcal{P}+}$, $\mathcal{I}_{\mathcal{P}+}$, $\mathcal{A}_{\mathcal{P}-}$, and $\mathcal{I}_{\mathcal{P}-}$. Every rule $r_1 \circ \ldots \circ r_k \subseteq r$ in $\mathcal{P}^+$ is instantiated multiple times with non-overlapping entities, such that for every instantiation with entities $e_1, \ldots, e_{k+1}$, it holds that $r_1(e_1, e_2), \ldots, r_k(e_k, e_{k+1}) \in \mathcal{A}_{\mathcal{P}+}$ and $r(e_1, e_{k+1}) \in \mathcal{I}_{\mathcal{P}+}$. Moreover, $e_1$ is reused in a further instantiation of every distractor $r'_1 \circ \ldots \circ r'_j \subseteq r$ in $\mathcal{P}^-$ such that $r'_1(e_1, e'_2), \ldots, r'_j(e'_j, e'_{j+1}) \in \mathcal{A}_{\mathcal{P}-}$ and $r(e_1, e'_{j+1}) \in \mathcal{I}_{\mathcal{P}-}$. We refer to $e_1$ as the anchor for this instantiation. Note that each instantiation involves *one* rule from $\mathcal{P}^+$ and *all* distractors from $\mathcal{P}^-$. An example of such an instantiation is shown in Figure 4. Models are trained on $\mathcal{A}_{\mathcal{P}+} \cup \mathcal{A}_{\mathcal{P}-} \cup \mathcal{I}_{\mathcal{P}+}$. They are thus exposed to many instances of every rule in $\mathcal{P}^+$ and no instances of distractors from $\mathcal{P}^-$. The models are then evaluated in an inference step in which entity embeddings are reinitialized and retrained on $\mathcal{A}_{\mathcal{P}+} \cup \mathcal{A}_{\mathcal{P}-}$. During this inference step, relation embeddings are frozen. The setting we describe here is similar to inductive link prediction, which requires an inference step to encode entities not shown during training.

To evaluate models, we use link prediction queries of the form $r(e_1, ?)$, where $r$ is the head of some rule in $\mathcal{P}^+$ and $e_1$ is the anchor of some instantiation of that rule. The model is asked to rank $|\mathcal{P}^-| + 1$ candidate entities: the correct answer and the answers that would be inferred when using the distractor rules in $\mathcal{P}^-$. For instance, for the instantiation from Figure 4, the query would be $t(e_1, ?)$ and the candidate entities would be $e_3, e_5, e_7$. We report results in terms of hits@1, i.e. the proportion of instantiations for which the correct entity is ranked higher than the distractor entities.

We evaluated[1] two state-of-the-art convex coordinate-wise models: Octagons (Charpenay & Schockaert, 2024) and

ExpressivE (Pavlovic & Sallinger, 2023). Inspired by the results from Section 5, we also experimented with a variant that parameterizes regions as polygons. All these models have accordant relation embeddings and were trained with a standard binary cross-entropy loss. Every rule has 100 instantiations and all models use 10-dimensional entity embeddings. We evaluated the polygon model in several configurations, where polygons have 2 to 10 edges. From the proof of Proposition 5.4, we know that the polygon model is expressive enough to achieve perfect results (i.e. hits@1 = 1) if the number of dimensions is higher than the number of distractors and the number of edges is at least $k + 2$, with $k$ the largest number of relations in the body of a distractor rule. In particular, the polygon model can achieve perfect results on all datasets provided that at least 6 dimensions are used and polygons have at least 6 edges. More details on the experimental setup can be found in Appendix I.

The results of our experiments are summarized in Table 2. All models show highly unstable results across runs, sometimes performing worse than random. This suggests that standard strategies for learning KG embeddings may not be sufficient for region-based models, and that stronger inductive biases may need to be imposed to improve their empirical performance. Focusing on the *best* results across 20 runs (which gives an indication of the potential of each model), Octagons and ExpressivE perform well on PERM2, PERM3, MIX2 and REP13, but poorly on REP12 and COMB. The polygon model underperforms ExpressivE, regardless of the number of edges, except on COMB. On this dataset, all polygons outperform Octagons and ExpressivE in at least one run, reaching an almost perfect score for polygons with 6 edges. Overall, our results show that the proposed benchmark is deceptively challenging for existing models, and that further work is needed to learn effective region-based models for settings that require faithful relational reasoning. Additional experiments show that increasing dimensionality does not consistently improve the results (see Appendix J).

---

[1] Our code for replicating these experiments is available on Github: https://github.com/vcharpenay/FaiRe.

## 8. Related Work

Early work on the expressivity of KG embeddings focused on whether a given embedding model is capable of capturing any knowledge graph, a property called *full expressiveness* (Wang et al., 2018; Kazemi & Poole, 2018). Popular KG embedding models, such as TransE and Distmult, were found not to be fully expressive (Wang et al., 2018), which served as a central motivation for models such as ComplEx (Trouillon et al., 2017) and SimplE (Kazemi & Poole, 2018).

Full expressivity refers to the ability of an embedding model to capture facts. The study of region-based models has emerged from the desire to also capture rules. Gutiérrez-Basulto & Schockaert (2018) studied region-based models from a theoretical perspective, showing that convex regions (with unrestricted cross-coordinate comparisons) are capable of capturing particular types of existential rules. Abboud et al. (2020) proposed a region-based model based on hyperboxes and analyzed which types of "inference patterns", i.e. types of rules, this model can capture. They showed, for instance, that arbitrary sets of intersection and hierarchy rules can be faithfully captured using box embeddings. In fact, they claimed this was still true in the presence of disjointness constraints, but this has since been disproven (Charpenay & Schockaert, 2024; Leemhuis & Kutz, 2025). A key limitation of box embeddings is that they cannot capture closed path rules. Pavlovic & Sallinger (2023) therefore proposed to use parallelograms instead of axis-aligned boxes. They showed that the resulting model, called ExpressivE, can capture individual closed path rules but did not study under which conditions faithfully capturing *sets* of such rules is possible. Charpenay & Schockaert (2024) advocated axis-aligned octagons, because such regions are closed under intersection and composition. They showed that the octagon model can faithfully capture sets of so-called regular closed path rules (i.e. rule bases where every relation is regular in the sense of Definition 5.2). Charpenay & Schockaert (2025) studied a simpler region-based model based on MuRE (Balazevic et al., 2019). They found that this model is also capable of faithfully capturing sets of closed path rules, provided that the coordinates of the entity embeddings are bounded. ReshuffIE (Pavlovic et al., 2025) uses a GNN encoder to learn region embeddings based on ordering constraints. This model is capable of capturing bounded reasoning with arbitrary sets of closed path rules. However, as it relies on cross-coordinate comparisons, learning and inference are somewhat inefficient. As we found in this paper, however, such cross-coordinate comparisons may be essential for faithfully capturing arbitrary sets of closed path rules, unless non-convex or non-accordant embeddings are used, which come with their own disadvantages.

While the aforementioned works have primarily focused on representing KGs, region-based models are also used for embedding ontologies (Mondal et al., 2021; Xiong et al., 2022; Jackermeier et al., 2023; Özçep et al., 2023; Bourgaux et al., 2024; Leemhuis & Kutz, 2025). In these cases, a set of rules (i.e. ontology axioms) is provided as part of the input, and the task of interest may involve predicting plausible missing rules, in addition to predicting missing facts.

The expressivity of GNNs (and graph transformers) has been extensively studied (Barceló et al., 2020), including for link prediction in KGs (Cucala et al., 2022; 2023; Huang et al., 2023; Morris & Horrocks, 2025; Morris et al., 2025). The ability of standard GNNs to learn rules for link prediction is limited (Zhang et al., 2021). A common approach to enable more expressive reasoning, pioneered in NBFNet (Zhu et al., 2021), is to learn node embeddings that are conditioned on a given query. This enables expressive reasoning (Huang et al., 2023) but can be inefficient in practice, as a separate forward pass of the GNN is needed for each query. Edge Transformers (Bergen et al., 2021) enable expressive reasoning by modifying the attention mechanism from the Transformer to compute relational compositions. They compute relationships between any pair of entities in a single pass, but the required number of computations is cubic in the number of entities, which severely limits their scalability. Some authors have therefore proposed approximation strategies, which trade-off some of the expressivity of edge transformers for better scalability (Ståhlberg et al., 2025). Another approach is to use the so-called labelling trick (Zhang et al., 2021). This essentially involves using standard GNNs but relying on input representations that distinguish the different entities. ReshuffIE, for instance, relies on randomly initialized entity embeddings.

The state-of-the-art in KG reasoning has seen several advances in recent years, including through integration with Large Language Models (Zhu et al., 2024) and by pretraining foundation models (Galkin et al., 2024). Such approaches are tangential to the aims of this paper, which are about the theoretical expressivity of embedding models.

## 9. Conclusions

We have studied the expressivity of convex coordinate-wise region-based embeddings. We showed that non-accordant relation embeddings can faithfully capture bounded reasoning with arbitrary sets of closed path rules. For accordant relation embeddings, this is not true in general, but we identified rather flexible sufficient conditions under which rule bases can be faithfully captured. Unfortunately, we also found that convex coordinate-wise models sometimes capture spurious semantic dependencies (which cannot be expressed as closed path rules). Models with cross-coordinate comparisons can overcome these limitations (Pavlovic et al., 2025), but such models are less efficient and can be difficult to train. Further work is thus needed to find minimal exten-

sions of convex coordinate-wise models (e.g. based on non-convex regions or limited cross-coordinate comparisons) that overcome such limitations. Empirically, we found that KG embedding models struggle to faithfully capture rule bases, even when they should be capable of doing so in theory. These results show that there is still substantial scope for improving the performance of convex coordinate-wise models, despite their theoretical limitations, for instance by studying ways of imposing stronger inductive biases.

## Acknowledgments

Computations have been performed on the supercomputer facilities of the Mésocentre Clermont-Auvergne of the Université Clermont Auvergne. Steven Schockaert was supported by EPSRC Grant EP/W003309/1.

## Impact Statement

This paper presents work whose goal is to advance the field of Machine Learning. There are many potential societal consequences of our work, none which we feel must be specifically highlighted here.

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

# A. Proof of Proposition 4.1

Let $r_1, ..., r_n$ be an enumeration of the different relations that appear in $\mathcal{P}$. Let $q_1, ..., q_n$ be distinct prime numbers. Let us consider the regions $\mathsf{L}_1, ..., \mathsf{L}_n$ defined as follows:

$$\mathsf{L}_i = \{(x, y) \in \mathbb{R}^2 \mid 2y - x = \log q_i\} \tag{2}$$

We define the one-dimensional relation embedding $\eta$ as follows ($r \in \{r_1, ..., r_n\}$):

$$\eta(r) = \bigcup \{\mathsf{L}_{i_1} \diamond ... \diamond \mathsf{L}_{i_m} \mid i_1, ..., i_m \in \{1, ..., n\}, \mathcal{P} \models r_{i_1} \circ ... \circ r_{i_m} \subseteq r\}$$

Note that for rule bases with cyclic dependencies, the number of arguments in the union may be countably infinite. We show that the relation embedding $\eta$ captures $\mathcal{P}$.

**Lemma A.1.** *If $\mathcal{P} \models s_1 \circ ... \circ s_m \subseteq r$ then we have:*

$$\eta(s_1) \diamond ... \diamond \eta(s_m) \subseteq \eta(r)$$

*Proof.* Suppose $(x_1, x_{m_1}) \in \eta(s_1) \diamond ... \diamond \eta(s_m)$. Then there are $x_2, ..., x_m \in \mathbb{R}$ such that $(x_1, x_2) \in \eta(s_1), ..., (x_m, x_{m+1}) \in \eta(s_m)$. For each $i \in \{1, ..., m\}$, the fact that $(x_i, x_{i+1}) \in \eta(r_i)$ means that there exist relations $r_{j_{i1}}, ..., r_{j_{ip_i}}$ such that $\mathcal{P} \models r_{j_{i1}} \circ ... \circ r_{j_{ip_i}} \subseteq s_i$ and $(x_i, x_{i+1}) \in \mathsf{L}_{j_{i1}} \diamond ... \diamond \mathsf{L}_{j_{ip_i}}$. Since $\mathcal{P} \models s_1 \circ ... \circ s_m \subseteq r$ we then also have $\mathcal{P} \models r_{j_{11}} \circ ... \circ r_{j_{1p_1}} \circ r_{j_{21}} \circ ... \circ r_{j_{mp_m}} \subseteq r$. This means $\mathsf{L}_{j_{11}} \diamond ... \diamond \mathsf{L}_{j_{mp_m}} \in \eta(r)$. We furthermore have $(x_1, x_{m+1}) \in \mathsf{L}_{j_{11}} \diamond ... \diamond \mathsf{L}_{j_{mp_m}}$ and thus $(x_1, x_{m+1}) \in \eta(r)$. $\qquad\square$

**Lemma A.2.** *Let $i_1, ..., i_m \in \{1, ..., n\}$. It holds that*

$$\mathsf{L}_{i_1} \diamond ... \diamond \mathsf{L}_{i_m} = \{(x, y) \in \mathbb{R}^2 \mid 2^m y - x = \sum_{j=1}^{m} 2^{j-1} \log q_{i_j}\}$$

*Proof.* We show the result by induction. We clearly have that the result holds if $m = 1$. Now suppose $m > 1$. Using the induction hypothesis, we then find:

$$(x, z) \in \mathsf{L}_{i_1} \diamond ... \diamond \mathsf{L}_{i_m} \Leftrightarrow \exists y \, . \, ((x, y) \in \mathsf{L}_{i_1} \diamond ... \diamond \mathsf{L}_{i_{m-1}}) \wedge ((y, z) \in \mathsf{L}_{i_m})$$

$$\Leftrightarrow \exists y \, . \, (2^{m-1} y - x = \sum_{j=1}^{m-1} 2^{j-1} \log q_{i_j}) \wedge (2z - y = \log q_{i_m})$$

$$\Leftrightarrow 2^{m-1}(2z - \log q_{i_m}) - x = \sum_{j=1}^{m-1} 2^{j-1} \log q_{i_j}$$

$$\Leftrightarrow 2^m z - x = \sum_{j=1}^{m} 2^{j-1} \log q_{i_j}$$

$\qquad\square$

**Lemma A.3.** *Let $i_1, ..., i_m, i'_1, ..., i'_{m'} \in \{1, ..., n\}$. Suppose that*

$$\mathsf{L}_{i_1} \diamond ... \diamond \mathsf{L}_{i_m} = \mathsf{L}_{i'_1} \diamond ... \diamond \mathsf{L}_{i'_{m'}}$$

*Then we have $m = m'$ and $i_1 = i'_1, ..., i_m = i'_m$.*

*Proof.* From $\mathsf{L}_{i_1} \diamond ... \diamond \mathsf{L}_{i_m} = \mathsf{L}_{i'_1} \diamond ... \diamond \mathsf{L}_{i'_{m'}}$ it follows that

$$\sum_{j=1}^{m} 2^{j-1} \log q_{i_j} = \sum_{j=1}^{m'} 2^{j-1} \log q_{i'_j}$$

$$\Leftrightarrow \exp\left(\sum_{j=1}^{m} 2^{j-1} \log q_{i_j}\right) = \exp\left(\sum_{j=1}^{m'} 2^{j-1} \log q_{i'_j}\right)$$

$$\Leftrightarrow \prod_{j=1}^{m} \exp\left(\log q_{i_j}^{2^{j-1}}\right) = \prod_{j=1}^{m'} \exp\left(\log q_{i'_j}^{2^{j-1}}\right)$$

$$\Leftrightarrow \prod_{j=1}^{m} q_{i_j}^{2^{j-1}} = \prod_{j=1}^{m'} q_{i'_j}^{2^{j-1}}$$

$$\Leftrightarrow \prod_{l=1}^{n} q_l^{\sum_{j=1}^{m} \mathbb{1}[i_j=l]2^{j-1}} = \prod_{l=1}^{n} q_l^{\sum_{j=1}^{m'} \mathbb{1}[i'_j=l]2^{j-1}}$$

where $\mathbb{1}[\alpha] = 1$ if the condition $\alpha$ is true and $\mathbb{1}[\alpha] = 0$ otherwise. Since integers have a unique prime factorization, it follows that for each $l \in \{1, ..., m\}$ we have:

$$\sum_{j=1}^{m} \mathbb{1}[i_j = l]2^{j-1} = \sum_{j=1}^{m'} \mathbb{1}[i'_j = l]2^{j-1}$$

This is only possible if $m = m'$ and $i_j = i'_j$ for all $j \in \{1, ..., m\}$. $\qquad\square$

**Lemma A.4.** *Suppose* $\eta(r_{i_1}) \diamond ... \diamond \eta(r_{i_m}) \subseteq \eta(r)$. *It holds that*

$$\mathcal{P} \models r_{i_1} \circ ... \circ r_{i_m} \subseteq r$$

*Proof.* We clearly have $\mathsf{L}_{i_1} \diamond ... \diamond \mathsf{L}_{i_m} \subseteq \eta(r_{i_1}) \diamond ... \diamond \eta(r_{i_m})$ and thus also $\mathsf{L}_{i_1} \diamond ... \diamond \mathsf{L}_{i_m} \in \eta(r)$. By construction, we can only have $\mathsf{L}_{i_1} \diamond ... \diamond \mathsf{L}_{i_m} \in \eta(r)$ if $\mathsf{L}_{i_1} \diamond ... \diamond \mathsf{L}_{i_m} = \mathsf{L}_{i'_1} \diamond ... \diamond \mathsf{L}_{i'_{m'}}$ such that $\mathcal{P} \models r_{i'_1} \circ ... \circ r_{i'_{m'}} \subseteq r$. By Lemma A.3, we find $\mathcal{P} \models r_{i_1} \circ ... \circ r_{i_m} \subseteq r$. $\qquad\square$

Proposition 4.1 follows immediately from Lemma A.1 and A.4.

## B. Proof of Proposition 4.2

Suppose there is a convex coordinate wise relation embedding $\eta$ that captures the rules in $\mathcal{P}$ but not the following rule:

$$r_1 \circ r_1 \circ r_1 \subseteq r_4 \tag{3}$$

By assumption, there must exist some coordinate $i$ and point $(x_1, y_3) \in \mathbb{R}^2$ satisfying:

$$(x_1, y_3) \in \big(\eta_i(r_1) \diamond \eta_i(r_1) \diamond \eta_i(r_1)\big) \setminus \eta_i(r_4) \tag{4}$$

Let us fix such a coordinate $i$. Since $(x_1, y_3) \in \eta_i(r_1) \diamond \eta_i(r_1) \diamond \eta_i(r_1)$, there must exist $y_1, y_2 \in \mathbb{R}$ such that

$$(x_1, y_1) \in \eta_i(r_1)$$
$$(y_1, y_2) \in \eta_i(r_1)$$
$$(y_2, y_3) \in \eta_i(r_1)$$

First assume that $x_1 \le y_1 \le y_2$ or $x_1 \ge y_1 \ge y_2$. We find:

- We have assumed $(x_1, y_3) \in \eta_i(r_1) \diamond \eta_i(r_1) \diamond \eta_i(r_1)$. Since $r_1 \circ r_1 \circ r_1 \subseteq r_2$ is captured by $\eta$, this means $(x_1, y_3) \in \eta_i(r_2)$.

- We have $(y_2, y_3) \in \eta_i(r_1)$. Since $r_1 \subseteq r_2$ is captured, this means $(y_2, y_3) \in \eta_i(r_2)$.

- From $(x_1, y_3) \in \eta_i(r_2)$ and $(y_2, y_3) \in \eta_i(r_2)$, using the convexity of $\eta_i(r_2)$ and the fact that we assumed $x_1 \le y_1 \le y_2$ or $x_1 \ge y_1 \ge y_2$, we obtain $(y_1, y_3) \in \eta_i(r_2)$.

- From $(x_1, y_1) \in \eta_i(r_1)$ and the fact that $r_1 \subseteq r_2$ is captured, we obtain $(x_1, y_1) \in \eta_i(r_2)$.

- From $(x_1, y_1) \in \eta_i(r_2)$ and $(y_1, y_3) \in \eta_i(r_2)$, we obtain $(x_1, y_3) \in \eta_i(r_2) \diamond \eta_i(r_2)$, and thus, since $r_2 \circ r_2 \subseteq r_4$ is captured, $(x_1, y_3) \in \eta_i(r_4)$, a contradiction.

Let us now assume that $x_1 \leq y_2 \leq y_1$ or $x_1 \geq y_2 \geq y_1$. We find:

- We have $(x_1, y_2) \in \eta_i(r_1) \diamond \eta_i(r_1)$. Since $r_1 \circ r_1 \subseteq r_3$ is captured, it follows that $(x_1, y_2) \in \eta_i(r_3)$.

- We have $(y_1, y_3) \in \eta_i(r_1) \diamond \eta_i(r_1)$. Since $r_1 \circ r_1 \subseteq r_3$ is captured, it follows that $(y_1, y_3) \in \eta_i(r_3)$.

- We assumed $(x_1, y_3) \in \eta_i(r_1) \diamond \eta_i(r_1) \diamond \eta_i(r_1)$. Since $r_1 \circ r_1 \circ r_1 \subseteq r_3$ is captured, it follows that $(x_1, y_3) \in \eta_i(r_3)$.

- From $(x_1, y_3) \in \eta_i(r_3)$ and $(y_1, y_3) \in \eta_i(r_3)$, given the convexity of $\eta_i(r_3)$ and the fact that we assumed $x_1 \leq y_2 \leq y_1$ or $x_1 \geq y_2 \geq y_1$, it follows that $(y_2, y_3) \in \eta_i(r_3)$.

- From $(x_1, y_2) \in \eta_i(r_3)$ and $(y_2, y_3) \in \eta_i(r_3)$, it follow that $(x_1, y_3) \in \eta_i(r_3) \diamond \eta_i(r_3)$, and thus, since $r_3 \circ r_3 \subseteq r_4$ is captured, that $(x_1, y_4) \in \eta_i(r_4)$, a contradiction.

Finally, let us assume that $y_1 \leq x_1 \leq y_2$ or $y_1 \geq x_1 \geq y_2$. We find:

- We have $(y_1, y_3) \in \eta_i(r_1) \diamond \eta_i(r_1)$. Since $r_1 \subseteq r_2$ and $r_2 \circ r_2 \subseteq r_4$ are captured, we thus also have $(y_1, y_3) \in \eta_i(r_2) \diamond \eta_i(r_2)$ and $(y_1, y_3) \in \eta_i(r_4)$.

- We have $(y_2, y_3) \in \eta_i(r_1)$. Since $r_1 \subseteq r_4$ is captured, we thus also have $(y_2, y_3) \in \eta_i(r_4)$.

- Since $y_1 \leq x_1 \leq y_2$ or $y_1 \geq x_1 \geq y_2$, and given the convexity of $\eta_i(r_4)$, from $(y_1, y_3) \in \eta_i(r_4)$ and $(y_2, y_3) \in \eta_i(r_4)$ we infer $(x_1, y_3) \in \eta_i(r_4)$, a contradiction.

In all cases we thus arrive at a contradiction, meaning that a point $(x_1, y_3)$ satisying (4) cannot exist. It follows that $\eta$ must capture the rule (3).

## C. Parameterized Polygons

Before presenting the proof of Proposition 5.1 in the next section, we first study a class of parameterized polygons. After introducing this class of polygons, we establish some fundamental properties that the proof of the latter proposition will rely on.

### C.1. Intuitions

The region $\eta(r)$ in Figure 3 was not chosen arbitrarily. Essentially, we need polygons with three key properties. First, the polygon needs to contain some point of the form $(x^*, x^*)$, as otherwise we would have $\eta(r) \diamond ... \diamond \eta(r) = \emptyset$ for some finite number of compositions of $\eta(r)$. This is undesirable, as the corresponding relation embedding would then capture rules of the form $r \circ ... \circ r \subseteq s$, for any relation $s$. Second, we can think of the polygons as defining an upper bound on the $y$-coordinate and a lower bound. In the case of the region $\eta(r)$ in Figure 3, this upper bound consists of an increasing part followed by a decreasing part. This is needed to make it possible to have $\max\{z \mid (0, z) \in \eta(r) \diamond \eta(r)\} < \max\{z \mid (0, z) \in \eta(r) \diamond \eta(r) \diamond \eta(r)\}$ while at the same time also having $\max\{z \mid (0, z) \in \eta(r) \diamond \eta(r) \diamond \eta(r)\} > \max\{z \mid (0, z) \in \eta(r) \diamond \eta(r) \diamond \eta(r) \diamond \eta(r)\}$. Finally, we also need polygons where the lower bound is sufficiently close to the upper bound. The parameterized polygons that we introduce in this section are based on these key intuitions.

### C.2. Definition

Let us first consider two-dimensional regions of the following form ($x_1 \neq x_2$):

$$U(x_1, y_1, x_2, y_2) = \{(x, y) \mid y - y_1 \leq \frac{y_2 - y_1}{x_2 - x_1}(x - x_1)\}$$

$$L(x_1, y_1, x_2, y_2) = \{(x, y) \mid y - y_1 \geq \frac{y_2 - y_1}{x_2 - x_1}(x - x_1)\}$$

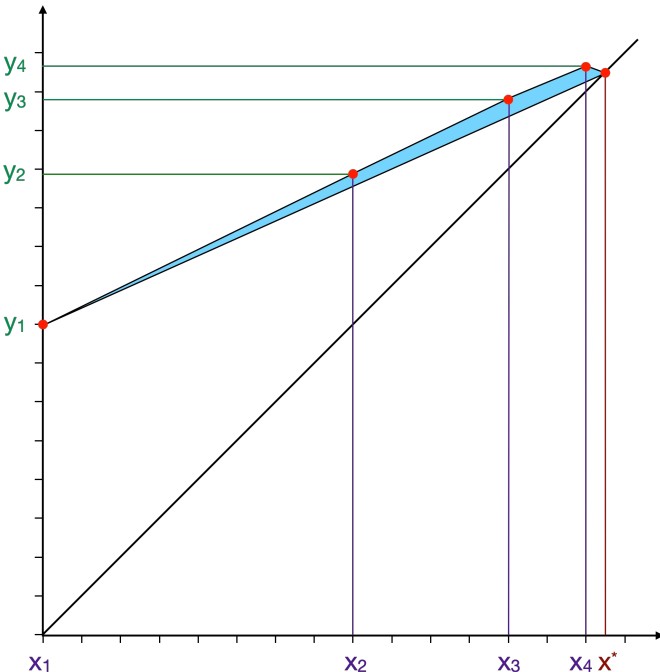

*Figure 5.* Illustration of a parameterized polygon with $k = 4$.

In other words, $U(x_1, y_1, x_2, y_2)$ represents the region which is upper bounded by the line that goes through the points $(x_1, y_1)$ and $(x_2, y_2)$. Similarly, $L(x_1, y_1, x_2, y_2)$ represents the region that is lower bounded by this line. Using these basic building blocks, we now construct polygons of the following form:

$$\mathsf{Reg}(x_1, y_1, ..., x_k, y_k; x^*) = \bigcap_{i=1}^{k-1} U(x_i, y_i, x_{i+1}, y_{i+1}) \cap U(x_k, y_k, x^*, x^*) \cap L(x_1, y_1, x^*, x^*)$$

We will focus on regions of this form which are parameterized by a set of indices $I \subseteq \{1, ..., k\}$. In particular, we write $R_I$ to denote the region $\mathsf{Reg}(x_1, y_1, ..., x_k, y_k; x^*)$ where:

- $x_1 = 0$;

- $i \in \{2, ..., k+1\}$ we have:

$$x_i = 1 + \lambda + ... + \lambda^{i-2} = \frac{1 - \lambda^{i-1}}{1 - \lambda}$$

- for $i \in \{1, ..., k\}$ we have:

$$y_i = \begin{cases} x_{i+1} - (x_i^2 - \lambda^{3k})\varepsilon & \text{if } i \in I \\ x_{i+1} - x_i^2 \varepsilon & \text{otherwise} \end{cases}$$

- we have

$$x^* = x_{k+1} - \delta$$

An example of a polygon of this form is shown in Figure 5. The constants $\varepsilon, \lambda, \delta > 0$ will be fixed for all regions. However, we will assume that these constants are sufficiently small (as made more precise in the results below). In particular, $\lambda$ has to be sufficiently small relative to $k$, $\delta$ has to be sufficiently small relative to $\lambda$, and $\varepsilon$ has to be sufficiently small relative to $\delta$.

## C.3. Basic Properties of the Parameterized Polygons

Let $R_I = \mathsf{Reg}(x_1, y_1, ..., x_k, y_k; x^*)$. We show the following key properties:

- $x_1 < x_2 < .... < x_k < x^*$.

- $y_1 < y_2 < .... < y_k$ and $y_1 < x^* < y_k$.

- The points $(x_1, y_1), ...(x_k, y_k), (x^*, x^*)$ are all vertices of $R_I$. In particular, we show that $(x_i, y_i) \in R_I$, for $i \in \{1, ..., k\}$ and $(x^*, x^*) \in R_I$.

In the proofs we will often rely on the following equalities:

$$x_{i+1} - x_i = \lambda^{i-1} \tag{5}$$
$$x_{i+1}^2 - x_i^2 = (x_{i+1} - x_i)(x_{i+1} + x_i) = \lambda^{i-1}(x_{i+1} + x_i) \tag{6}$$
$$x_{i+1} - x_{j+1} = \lambda(x_i - x_j) \tag{7}$$

**Lemma C.1.** *Let $k \geq 2$ and $I \subseteq \{1, ..., k\}$. Let $R_I = \mathsf{Reg}(x_1, y_1, ..., x_k, y_k; x^*)$. Assume $0 < \delta < \lambda^{k-1}$. It holds that $x_1 < x_2 < ... < x_k < x^*$.*

*Proof.* We clearly have, since $\lambda > 0$, that $x_1 < x_2 < ... < x_k$. We furthermore have:

$$x_k < x^* \Leftrightarrow x_k < x_{k+1} - \delta$$
$$\Leftrightarrow 0 < \lambda^{k-1} - \delta$$
$$\Leftrightarrow \delta < \lambda^{k-1}$$

$\square$

**Lemma C.2.** *Let $k \geq 2$ and $I \subseteq \{1, ..., k\}$. Let $R_I = \mathsf{Reg}(x_1, y_1, ..., x_k, y_k; x^*)$. Assume $0 < \delta < \lambda^{k-1}$ and $0 < \varepsilon < \min(\frac{\delta}{x_k^2}, \frac{1}{\lambda^{-1}(x_k + x_{k-1}) + \lambda^k}, 1)$. It holds that $y_1 < ... y_{k-1} < x^* < y_k$.*

*Proof.* We first show that $y_i < y_{i+1}$ for $i \in \{1, ..., k-1\}$. Assume that $i \in I$ and $i + 1 \notin I$. We find:

$$y_i < y_{i+1} \Leftrightarrow x_{i+1} - (x_i^2 - \lambda^{3k})\varepsilon < x_{i+2} - x_{i+1}^2\varepsilon$$
$$\Leftrightarrow (x_{i+1}^2 - x_i^2 + \lambda^{3k})\varepsilon < \lambda^i$$
$$\Leftrightarrow \varepsilon < \frac{\lambda^i}{x_{i+1}^2 - x_i^2 + \lambda^{3k}}$$
$$\Leftrightarrow \varepsilon < \frac{\lambda^i}{\lambda^{i-1}(x_{i+1} + x_i) + \lambda^{3k}}$$
$$\Leftrightarrow \varepsilon < \frac{1}{\lambda^{-1}(x_{i+1} + x_i) + \lambda^{3k-i}}$$

The latter inequality is satisfied since we have $\varepsilon < \frac{1}{\lambda^{-1}(x_k + x_{k-1}) + \lambda^k} \leq \frac{1}{\lambda^{-1}(x_{i+1} + x_i) + \lambda^{3k-i}}$ for $i \leq k - 1$. The other cases, where $i \notin I$ and/or $i + 1 \in I$ follow a fortiori, since for $i \notin I$ we have $y_i < x_{i+1} - (x_i^2 - \lambda^{3k})\varepsilon$ and for $i + 1 \in I$ we have $y_{i+1} > x_{i+2} - x_{i+1}^2\varepsilon$. Next, assuming $k - 1 \in I$, we have:

$$y_{k-1} < x^* \Leftrightarrow 1 + ... + \lambda^{k-2} - (x_{k-1}^2 - \lambda^{3k})\varepsilon < 1 + \lambda + ... + \lambda^{k-1} - \delta$$
$$\Leftrightarrow \delta < \lambda^{k-1} + (x_{k-1}^2 - \lambda^{3k})\varepsilon$$

which holds since we assumed $\delta < \lambda^{k-1}$ and $(x_{k-1}^2 - \lambda^{3k})\varepsilon > 0$. The case where $1 \notin I$ again follows a fortiori. We also find for $k \notin I$:

$$x^* < y_k \Leftrightarrow x_{k+1} - \delta < x_{k+1} - x_k^2\varepsilon$$
$$\Leftrightarrow \varepsilon < \frac{\delta}{x_k^2}$$

which is satisfied by assumption. The case where $k \in I$ follows a fortiori. $\square$

**Lemma C.3.** *Let $k \geq 2$ and $I \subseteq \{1, ..., k\}$. Let $R_I = \mathsf{Reg}(x_1, y_1, ..., x_k, y_k; x^*)$. Assume $0 < \delta < \lambda^{k-1}$, $0 < \varepsilon < \min(\frac{\delta}{x_k^2}, \frac{1}{\lambda^{-1}(x_k + x_{k-1}) + \lambda^k})$ and $\lambda \leq \frac{1}{2}$. Let $i \in \{1, ..., k\}$. It holds that $(x_i, y_i) \in U(x_j, y_j, x_{j+1}, y_{j+1})$ for $j \in \{1, ..., k-1\}$.*

*Proof.* Clearly we have, for $i < k$, that $(x_i, y_i) \in U(x_i, y_i, x_{i+1}, y_{i+1})$ and, for $i > 1$, that $(x_i, y_i) \in U(x_{i-1}, y_{i-1}, x_i, y_i)$. We now show that $(x_i, y_i) \in U(x_j, y_j, x_{j+1}, y_{j+1})$ for $j \in \{1, ..., k-1\} \setminus \{i, i-1\}$. We find using (7):

$$(x_i, y_i) \in U(x_j, y_j, x_{j+1}, y_{j+1}) \Leftrightarrow y_i - y_j \leq \frac{y_{j+1} - y_j}{x_{j+1} - x_j}(x_i - x_j)$$

$$\Leftrightarrow y_i - y_j \leq \frac{(x_i - x_j)(y_{j+1} - y_j)}{\lambda^{j-1}}$$

First suppose $i \in I$, $j \in I$ and $j + 1 \in I$. Then we find:

$$y_i - y_j \leq \frac{(x_i - x_j)(y_{j+1} - y_j)}{\lambda^{j-1}}$$

$$\Leftrightarrow x_{i+1} - x_{j+1} + (x_j^2 - x_i^2)\varepsilon \leq \frac{(x_i - x_j)(x_{j+2} - x_{j+1} + (x_j^2 - x_{j+1}^2)\varepsilon)}{\lambda^{j-1}}$$

$$\Leftrightarrow x_{i+1} - x_{j+1} + (x_j^2 - x_i^2)\varepsilon \leq \frac{(x_i - x_j)(\lambda^j - \lambda^{j-1}(x_j + x_{j+1})\varepsilon)}{\lambda^{j-1}}$$

$$\Leftrightarrow (x_j^2 - x_i^2)\varepsilon \leq -\frac{\lambda^{j-1}(x_i - x_j)(x_j + x_{j+1})\varepsilon}{\lambda^{j-1}}$$

$$\Leftrightarrow x_j^2 - x_i^2 \leq (x_j - x_i)(x_j + x_{j+1})$$

$$\Leftrightarrow x_j^2 - x_i^2 \leq x_j^2 + x_j x_{j+1} - x_i x_j - x_i x_{j+1}$$

$$\Leftrightarrow 0 \leq x_i(x_i - x_j) - x_{j+1}(x_i - x_j)$$

$$\Leftrightarrow 0 \leq (x_i - x_{j+1})(x_i - x_j)$$

which is always satisfied.

Now suppose $i \in I$, $j \notin I$ and $j + 1 \notin I$. Then we similarly find

$$y_i - y_j \leq \frac{(x_i - x_j)(y_{j+1} - y_j)}{\lambda^{j-1}}$$

$$\Leftrightarrow x_{i+1} - x_{j+1} + (\lambda^{3k} + x_j^2 - x_i^2)\varepsilon \leq \frac{(x_i - x_j)(x_{j+2} - x_{j+1} + (x_j^2 - x_{j+1}^2)\varepsilon)}{\lambda^{j-1}}$$

$$\Leftrightarrow \lambda^{3k} + x_j^2 - x_i^2 \leq (x_j - x_i)$$

$$\Leftrightarrow \lambda^{3k} \leq (x_i - x_{j+1})(x_i - x_j)$$

Since $i \neq j + 1$ and $i \neq j$ we have $|x_i - x_{j+1}| \geq \lambda^k$ and $|x_i - x_j| \geq \lambda^k$. Furthermore, the signs of $x_i - x_{j+1}$ and $x_i - x_j$ are the same, hence the inequality must hold.

Next suppose $i \in I$, $j \notin I$ and $j + 1 \in I$. Then we find:

$$y_i - y_j \leq \frac{(x_i - x_j)(y_{j+1} - y_j)}{\lambda^{j-1}}$$

$$\Leftrightarrow x_{i+1} - x_{j+1} + (\lambda^{3k} + x_j^2 - x_i^2)\varepsilon \leq \frac{(x_i - x_j)(x_{j+2} - x_{j+1} + (\lambda^{3k} + x_j^2 - x_{j+1}^2)\varepsilon)}{\lambda^{j-1}}$$

$$\Leftrightarrow x_{i+1} - x_{j+1} + (\lambda^{3k} + x_j^2 - x_i^2)\varepsilon \leq \frac{(x_i - x_j)(\lambda^j + (\lambda^{3k} + x_j^2 - x_{j+1}^2)\varepsilon)}{\lambda^{j-1}}$$

$$\Leftrightarrow (\lambda^{3k} + x_j^2 - x_i^2)\varepsilon \leq \frac{(x_i - x_j)(\lambda^{3k} + x_j^2 - x_{j+1}^2)\varepsilon}{\lambda^{j-1}}$$

$$\Leftrightarrow \lambda^{3k} + x_j^2 - x_i^2 \leq (x_i - x_j)\lambda^{3k-j+1} - (x_i - x_j)(x_j + x_{j+1})$$

$$\Leftrightarrow \lambda^{3k} \leq (x_i - x_{j+1} + \lambda^{3k-j+1})(x_i - x_j)$$

If $i > j + 1$ then the latter inequality follows from $x_i - x_{j+1} + \lambda^{3k-j+1} \geq x_i - x_{j+1} \geq \lambda^k$ and $x_i - x_j \geq \lambda^k$. If $i < j$, then we have $x_i - x_{j+1} + \lambda^{3k-j+1} \leq -\lambda^{j-1} + \lambda^{3k-j+1} < 0$ and $x_i - x_j \leq -\lambda^{j-2}$. And thus we have:

$$(x_i - x_{j+1} + \lambda^{3k-j+1})(x_i - x_j) \geq (\lambda^{j-1} - \lambda^{3k-j+1})\lambda^{j-2}$$
$$= \lambda^{2j-3} - \lambda^{3k-1}$$

and thus it is sufficient to show

$$\lambda^{3k} \leq \lambda^{2j-3} - \lambda^{3k-1}$$
$$\Leftrightarrow \lambda^{3k-2j+3} + \lambda^{3k-2j+2} \leq 1$$

which is satisfied as $\lambda^{3k-2j+3} \leq \lambda \leq \frac{1}{2}$ and $\lambda^{3k-2j+2} \leq \lambda \leq \frac{1}{2}$.

Now consider the case where $i \in I$, $j \in I$ and $j + 1 \notin I$. We find:

$$y_i - y_j \leq \frac{(x_i - x_j)(y_{j+1} - y_j)}{\lambda^{j-1}}$$

$$\Leftrightarrow x_{i+1} - x_{j+1} + (x_j^2 - x_i^2)\varepsilon \leq \frac{(x_i - x_j)(x_{j+2} - x_{j+1} + (x_j^2 - x_{j+1}^2 - \lambda^{3k})\varepsilon)}{\lambda^{j-1}}$$

$$\Leftrightarrow x_{i+1} - x_{j+1} + (x_j^2 - x_i^2)\varepsilon \leq \frac{(x_i - x_j)(\lambda^j + (x_j^2 - x_{j+1}^2 - \lambda^{3k})\varepsilon)}{\lambda^{j-1}}$$

$$\Leftrightarrow (x_j^2 - x_i^2)\varepsilon \leq \frac{(x_i - x_j)(x_j^2 - x_{j+1}^2 - \lambda^{3k})\varepsilon}{\lambda^{j-1}}$$

$$\Leftrightarrow (x_j^2 - x_i^2)\varepsilon \leq \frac{(x_i - x_j)(-(x_j + x_{j+1})\lambda^{j-1} - \lambda^{3k})\varepsilon}{\lambda^{j-1}}$$

$$\Leftrightarrow x_j^2 - x_i^2 \leq -(x_i - x_j)(x_j + x_{j+1}) - (x_i - x_j)\lambda^{3k-j+1}$$

$$\Leftrightarrow 0 \leq x_i^2 - x_i x_j - x_i x_{j+1} + x_j x_{j+1} - (x_i - x_j)\lambda^{3k-j+1}$$

$$\Leftrightarrow 0 \leq (x_i - x_{j+1} - \lambda^{3k-j+1})(x_i - x_j)$$

If $i > j + 1$ then this follows because $x_i > x_j$ and $x_i - x_{j+1} \geq \lambda^{i-2} \geq \lambda^{3k-j+1}$. If $i < j$ then we have $x_i - x_{j+1} - \lambda^{3k-j+1} \leq 0$ and $x_i - x_j \leq 0$ and the inequality again holds.

Finally, the cases where $i \notin I$ follow a fortiori. $\qquad\square$

**Lemma C.4.** *Let $k \geq 2$ and $I \subseteq \{1, ..., k\}$. Let $R_I = \mathrm{Reg}(x_1, y_1, ..., x_k, y_k; x^*)$. Assume $0 < \delta < \lambda^{k-1}$, $0 < \varepsilon < \min(\frac{\delta}{x_k^2}, \frac{1}{\lambda^{-1}(x_k + x_{k-1}) + \lambda^k})$ and $\lambda \leq \frac{1}{2}$. Let $i \in \{1, ..., k\}$. It holds that $(x_i, y_i) \in U(x_k, y_k, x^*, x^*)$.*

*Proof.* We have:

$$(x_i, y_i) \in U(x_k, y_k, x^*, x^*) \Leftrightarrow y_i - y_k \leq \frac{x^* - y_k}{x^* - x_k}(x_i - x_k)$$
$$\Leftrightarrow (y_i - y_k)(x^* - x_k) \leq (x^* - y_k)(x_i - x_k)$$

From Lemmas C.1 and C.2, we know that the left-hand side is negative while the right-hand side is positive, meaning that the inequality is satisfied. $\qquad\square$

**Lemma C.5.** *Let $k \geq 2$ and $I \subseteq \{1, ..., k\}$. Let $R_I = \mathrm{Reg}(x_1, y_1, ..., x_k, y_k; x^*)$. Assume $0 < \delta < \lambda^{k-1}$, $0 < \varepsilon < \min(\frac{\delta}{x_k^2}, \frac{1}{\lambda^{-1}(x_k + x_{k-1}) + \lambda^k}, \frac{\lambda_k}{x_{k+1}x_k^2})$ and $\lambda \leq \frac{1}{2}$. Let $i \in \{1, ..., k\}$. It holds that $(x_i, y_i) \in L(x_1, y_1, x^*, x^*)$.*

*Proof.* We find (using $x_1 = 0$):

$$(x_i, y_i) \in L(x_1, y_1, x^*, x^*)$$
$$\Leftrightarrow y_i - y_1 \geq \frac{x^* - y_1}{x^* - x_1}(x_i - x_1)$$

$$\Leftrightarrow y_i - y_1 \geq \frac{x^* - y_1}{x^*} x_i$$

$$\Leftrightarrow y_i x^* - y_1 x^* \geq x^* x_i - x_i y_1$$

We show this for the case where $i \in I$ and $1 \in I$. The other cases follow a fortiori. We find:

$$y_i x^* - y_1 x^* \geq x^* x_i - x_i y_1$$
$$\Leftrightarrow (x_{i+1} + (\lambda^{3k} - x_i^2)\varepsilon - x_i - 1 - \lambda^{3k}\varepsilon)x^* + x_i(1 + \lambda^{3k}\varepsilon) \geq 0$$
$$\Leftrightarrow (x_{i+1} - x_i^2\varepsilon - x_i - 1)x^* + x_i(1 + \lambda^{3k}\varepsilon) \geq 0$$
$$\Leftrightarrow (x_{i+1} - x_i^2\varepsilon - x_i - 1)(x_{k+1} - \delta) + x_i(1 + \lambda^{3k}\varepsilon) \geq 0$$

We have:

$$x_{i+1} - x_i^2\varepsilon - x_i - 1 \leq 0 \Leftrightarrow \lambda^{i-1} - 1 \leq x_i^2\varepsilon$$

The latter inequality is always satisfied given that $\lambda^{i-1} - 1 < 0$ and $x_i^2\varepsilon > 0$. We also have $x_i \geq 0$. It is thus sufficient to show:

$$(x_{i+1} - x_i^2\varepsilon - x_i - 1)x_{k+1} + x_i \geq 0$$
$$\Leftrightarrow (\lambda^{i-1} - x_i^2\varepsilon - 1)x_{k+1} + x_i \geq 0$$
$$\Leftrightarrow (\lambda^{i-1} - 1)x_{k+1} + x_i \geq x_{k+1}x_i^2\varepsilon$$
$$\Leftrightarrow (\lambda^{i-1} + ... + \lambda^{k+i-2}) - (1 + ... + \lambda^{k-1}) + (1 + \lambda + ... + \lambda^{i-2}) \geq x_{k+1}x_i^2\varepsilon$$
$$\Leftrightarrow \lambda_k + ... + \lambda_{k+i-2} \geq x_{k+1}x_i^2\varepsilon$$

We have $\lambda_k + ... + \lambda_{k+i-2} \geq \lambda_k$ and $x_{k+1}x_i^2\varepsilon \leq x_{k+1}x_k^2\varepsilon$, hence the latter inequality follows from our assumption that $\varepsilon < \frac{\lambda_k}{x_{k+1}x_k^2}$.

$\square$

**Lemma C.6.** *Let $k \geq 2$ and $I \subseteq \{1, ..., k\}$. Let $R_I = \mathsf{Reg}(x_1, y_1, ..., x_k, y_k; x^*)$. Assume $0 < \delta < \lambda^{k-1}$, $0 < \varepsilon < \min(\frac{\delta}{x_k^2}, \frac{1}{\lambda^{-1}(x_k + x_{k-1}) + \lambda^k})$ and $\lambda \leq \frac{1}{2}$. It holds that $(x^*, x^*) \in R_I$.*

*Proof.* We trivially find that $(x^*, x^*) \in U(x_k, y_k, x^*, x^*)$ and $(x^*, x^*) \in L(x_1, y_1, x^*, x^*)$. For $j \in \{1, ..., k-1\}$ we need to show

$$(x^*, x^*) \in U(x_j, y_j, x_{j+1}, y_{j+1})$$
$$\Leftrightarrow x^* - y_j \leq \frac{y_{j+1} - y_j}{x_{j+1} - x_j}(x^* - x_j)$$

However, from Lemma C.3 we already know that:

$$y_k - y_j \leq \frac{y_{j+1} - y_j}{x_{j+1} - x_j}(x_k - x_j)$$

The result thus immediately follows from the fact that we have $x^* > x_k$, $x^* < y_k$, $x_{j+1} > x_j$ and, by Lemma C.2, $y_{j+1} > y_j$.

$\square$

### C.4. Composing Parameterized Polygons

We now establish a number of results about the composition of regions, focusing on the considered family of parameterized polygons. These results will be used in the proofs of Propositions 5.1 and 5.4, in the following sections.

**Lemma C.7.** *Let $k \geq 2$, $\ell \geq 1$ and $p \in \{1, ..., k - \ell\}$. Let*

$$x = x_p + \sum_{i=1}^{m_1} a_i \varepsilon^i$$

$$z_p = x_{p+\ell} + \sum_{i=1}^{m_2} b_i \varepsilon^i$$

$$z_{p+1} = x_{p+1+\ell} + \sum_{i=1}^{m_2} c_i \varepsilon^i$$

for some $m_1, m_2 \geq 0$, and coefficients $a_i, b_i, c_i$ which do not depend on $\varepsilon$. For sufficiently small $\varepsilon$, it holds that

$$\max\{z \mid (x, z) \in U(x_p, z_p, x_{p+1}, z_{p+1})\} = x_{p+\ell} + (b_1 + a_1\lambda^\ell)\varepsilon + \sum_{i=2}^{m_1+m_2} a_i' \varepsilon^i$$

for some coefficients $a_2', ..., a_{m+2}'$ that do not depend on $\varepsilon$.

*Proof.* We find:

$$\max\{z \mid (x, z) \in U(x_p, z_p, x_{p+1}, z_{p+1})\}$$

$$= z_p + \frac{(x - x_p)(z_{p+1} - z_p)}{x_{p+1} - x_p}$$

$$= x_{p+\ell} + \sum_{i=1}^{m_2} b_i \varepsilon^i + \frac{(x_p + \sum_{i=1}^{m_1} a_i \varepsilon^i - x_p)(x_{p+\ell+1} - x_{p+\ell} + \sum_{i=1}^{m_2}(c_i - b_i)\varepsilon^i)}{x_{p+1} - x_p}$$

$$= x_{p+\ell} + \sum_{i=1}^{m_2} b_i \varepsilon^i + \frac{(\sum_{i=1}^{m_1} a_i \varepsilon^i)(\lambda^{p+\ell-1} + \sum_{i=1}^{m_2}(c_i - b_i)\varepsilon^i)}{\lambda^{p-1}}$$

$$= x_{p+\ell} + (b_1 + a_1\lambda^\ell)\varepsilon + \sum_{i=2}^{m_1+m_2} a_i' \varepsilon^i$$

□

**Lemma C.8.** *Let $k \geq 2$, $\ell \geq 1$ and $p \in \{1, ..., k - \ell\}$. Let*

$$x = x_{p+1} + \sum_{i=1}^{m_1} a_i \varepsilon^i$$

$$z_p = x_{p+\ell} + \sum_{i=1}^{m_2} b_i \varepsilon^i$$

$$z_{p+1} = x_{p+1+\ell} + \sum_{i=1}^{m_2} c_i \varepsilon^i$$

*for some $m_1, m_2 \geq 0$, and coefficients $a_i, b_i, c_i$ (which do not depend on $\varepsilon$). For sufficiently small $\varepsilon$, it holds that*

$$\max\{z \mid (x, z) \in U(x_p, z_p, x_{p+1}, z_{p+1})\} = x_{p+\ell+1} + (a_1\lambda^\ell + c_1)\varepsilon + \sum_{i=2}^{m_1+m_2} a_i' \varepsilon^i$$

*for some coefficients $a_2', ..., a_{m+2}'$ that do not depend on $\varepsilon$.*

*Proof.* We find:

$$\max\{z \mid (x, z) \in U(x_p, z_p, x_{p+1}, z_{p+1})\}$$

$$= z_p + \frac{(x - x_p)(z_{p+1} - z_p)}{x_{p+1} - x_p}$$

$$= x_{p+\ell} + \sum_{i=1}^{m_2} b_i \varepsilon^i + \frac{(x_{p+1} + \sum_{i=1}^{m_1} a_i \varepsilon^i - x_p)(x_{p+\ell+1} - x_{p+\ell} + \sum_{i=1}^{m_2}(c_i - b_i)\varepsilon^i)}{x_{p+1} - x_p}$$

$$= x_{p+\ell} + \sum_{i=1}^{m_2} b_i \varepsilon^i + \frac{(\lambda^{p-1} + \sum_{i=1}^{m_1} a_i \varepsilon^i)(\lambda^{p+\ell-1} + \sum_{i=1}^{m_2}(c_i - b_i)\varepsilon^i)}{\lambda^{p-1}}$$

$$= x_{p+\ell} + \lambda^{p+\ell-1} + \sum_{i=1}^{m_2} b_i \varepsilon^i + \frac{(\sum_{i=1}^{m_1} a_i \varepsilon^i)(\lambda^{p+\ell-1} + \sum_{i=1}^{m_2}(c_i - b_i)\varepsilon^i)}{\lambda^{p-1}} + \sum_{i=1}^{m_2}(c_i - b_i)\varepsilon^i$$

$$= x_{p+\ell+1} + \sum_{i=1}^{m_2} b_i \varepsilon^i + \frac{(\sum_{i=1}^{m_1} a_i \varepsilon^i)(\lambda^{p+\ell-1} + \sum_{i=1}^{m_2}(c_i - b_i)\varepsilon^i)}{\lambda^{p-1}} + \sum_{i=1}^{m_2}(c_i - b_i)\varepsilon^i$$

$$= x_{p+\ell+1} + (b_1 + a_1\lambda^\ell + (c_1 - b_1))\varepsilon + \sum_{i=2}^{m_1+m_2} a_i' \varepsilon^i$$

$$= x_{p+\ell+1} + (a_1\lambda^\ell + c_1)\varepsilon + \sum_{i=2}^{m_1+m_2} a_i' \varepsilon^i$$

$\square$

**Corollary C.9.** *Let $I \subseteq \{1, ..., k\}$. Let $u_i = 1$ if $i \in I$ and $u_i = 0$ otherwise. Let $p \in \{1, ..., k-1\}$ and*

$$x = x_{p+1} + \sum_{i=1}^{m} a_i \varepsilon^i$$

*for some $m \geq 0$ and coefficients $a_1, ..., a_m$ (which do not depend on $\varepsilon$). Assume that $x_p < x \leq x_{p+1}$ (i.e. either $a_1 < 0$ and $x > x_p$, or $a_1 = ... = a_m = 0$). For sufficiently small $\varepsilon$, it holds that*

$$\max\{z \,|\, (x,z) \in R_I\} = x_{p+2} + (\lambda a_1 + u_{p+1}\lambda^{3k} - x_{p+1}^2)\varepsilon + \sum_{i=2}^{m+1} a_i' \varepsilon^i$$

*for some coefficients $a_2', ..., a_{m+1}'$ which do not depend on $\varepsilon$.*

**Lemma C.10.** *Let $\ell \in \{2, ..., k\}$. Let $I_1, ..., I_\ell \subseteq \{1, ..., k\}$. Let $u_{i,j} = 1$ if $i \in I_j$ and $u_{i,j} = 0$ otherwise. Let $p \in \{1, ..., k-\ell\}$ and*

$$x = x_{p+1} + \sum_{i=1}^{m} a_i \varepsilon^i$$

*for some $m \geq 0$ such that $x_p < x \leq x_{p+1}$ (i.e. either $a_1 < 0$ and $x > x_p$, or $a_1 = ... = a_m = 0$). Let $X_i$ ($i \in \{1, ..., \ell\}$) be defined as:*

$$X_i = \{(x,y) \,|\, x \geq \theta_i\}$$

*Assume that $\theta_i \leq x_i$. For sufficiently small $\varepsilon$, it holds that*

$$\max\{z \,|\, (x,z) \in (R_{I_1} \cap X_1) \diamond ... \diamond (R_{I_\ell} \cap R_\ell)\} = x_{p+\ell+1} + \left(\lambda^\ell a_1 + \sum_{j=1}^{\ell} u_{p+j,j}\lambda^{3k+\ell-j} - \sum_{j=0}^{\ell-1} \lambda^j x_{p+\ell-j}^2\right)\varepsilon + \sum_{i=2}^{m+\ell} a_i' \varepsilon^i$$

*for some coefficients $a_2', ..., a_{m+\ell}'$ which do not depend on $\varepsilon$.*

*Proof.* By repeatedly applying Corollary C.9, we find:

$$\max\{z \,|\, (x,z) \in R_{I_1} \diamond ... \diamond R_{I_\ell}\}$$

$$= \max\{z \,|\, (x_{p+2} + (\lambda a_1 + u_{p+1,1}\lambda^{3k} - x_{p+1}^2)\varepsilon + \sum_{i=2}^{m+1} a_i^{(1)}\varepsilon^i, z) \in R_{I_2} \diamond ... \diamond R_{I_\ell}\}$$

$$= \max\{z \,|\, (x_{p+3} + (\lambda^2 a_1 + u_{p+1,1}\lambda^{3k+1} + u_{p+2,2}\lambda^{3k} - \lambda x_{p+1}^2 - x_{p+2}^2)\varepsilon + \sum_{i=2}^{m+2} a_i^{(2)}\varepsilon^i, z) \in R_{I_3} \diamond ... \diamond R_{I_\ell}\}$$

$$= x_{p+\ell+1} + (\lambda^\ell a_1 + \sum_{j=1}^{\ell} u_{p+j,j}\lambda^{3k+\ell-j} - \sum_{j=0}^{\ell-1} \lambda^j x_{p+\ell-j}^2)\varepsilon + \sum_{i=2}^{m+\ell} a_i^{(\ell)}\varepsilon^i$$

$\square$

**Lemma C.11.** *Assume $k \geq 2$. Let $p \in \{1, ..., k\}$ and let $\ell, \xi \in \{1, ..., k\}$ be such that $p + \ell + \xi \leq k$. Assume that we have:*

$$z_p = x_{p+\ell} + \left( \sum_{j=1}^{\ell} u_j \lambda^{3k+\ell-j} - \sum_{j=0}^{\ell-1} \lambda^j x_{p-1+\ell-j}^2 \right) \varepsilon + \sum_{i=2}^{\ell} a_i \varepsilon^i$$

$$z_{p+1} = x_{p+1+\ell} + \left( \sum_{j=1}^{\ell} u_j' \lambda^{3k+\ell-j} - \sum_{j=0}^{\ell-1} \lambda^j x_{p+\ell-j}^2 \right) \varepsilon + \sum_{i=2}^{\ell} b_i \varepsilon^i$$

$$z_{p+\ell}' = x_{p+\ell+\xi} + \left( \sum_{j=1}^{\xi} u_j'' \lambda^{3k+\xi-j} - \sum_{j=0}^{\xi-1} \lambda^j x_{p+\ell-1+\xi-j}^2 \right) \varepsilon + \sum_{i=2}^{\xi} c_i \varepsilon^i$$

$$z_{p+\ell+1}' = x_{p+\ell+1+\xi} + \left( \sum_{j=1}^{\xi} u_j''' \lambda^{3k+\xi-j} - \sum_{j=0}^{\xi-1} \lambda^j x_{p+\ell+\xi-j}^2 \right) \varepsilon + \sum_{i=2}^{\xi} d_i \varepsilon^i$$

*for some coefficients $a_i, b_i, c_i, d_i \in \mathbb{R}$ which do not depend on $\varepsilon$, and some values $u_j, u_j', u_j'', u_j''' \in \{0, 1\}$. If $\delta$, $\lambda$ and $\varepsilon$ are sufficiently small, for each $i \in \{1, ..., k\}$, it holds that:*

$$U(x_p, z_p, x_{p+1}, z_{p+1}) \diamond U(x_{p+\ell}, z_{p+\ell}', x_{p+\ell+1}, z_{p+\ell+1}') \subseteq U(x_p, z_p'', x_{p+1}, z_{p+1}'')$$

*where*

$$z_p'' = x_{p+\ell+\xi} + \left( \sum_{j=1}^{\ell+\xi} v_j \lambda^{3k+\ell+\xi-j} - \sum_{j=0}^{\ell+\xi-1} \lambda^j x_{p-1+\ell+\xi-j}^2 \right) \varepsilon + \sum_{i=2}^{\ell+\xi} e_i \varepsilon^i$$

$$z_{p+1}'' = x_{p+\ell+1+\xi} + \left( \sum_{j=1}^{\ell+\xi} w_j \lambda^{3k+\ell+\xi-j} - \sum_{j=0}^{\ell+\xi-1} \lambda^j x_{p+\ell+\xi-j}^2 \right) \varepsilon + \sum_{i=2}^{\ell+\xi} f_i \varepsilon^i$$

*for some coefficients $e_i, f_i$ which do not depend on $\varepsilon$, and where:*

$$v_j = \begin{cases} u_j & \text{if } j \in \{1, ..., \ell\} \\ u_{j-\ell}'' & \text{otherwise} \end{cases} \qquad\qquad w_j = \begin{cases} u_j' & \text{if } j \in \{1, ..., \ell\} \\ u_{j-\ell}''' & \text{otherwise} \end{cases}$$

*Proof.* From Lemma C.7 we find:

$$\max\{z \mid (z_p, z) \in U(x_{p+\ell}, z_{p+\ell}', x_{p+\ell+1}, z_{p+\ell+1}'))$$

$$= x_{p+\ell+\xi} + \left( \left( \sum_{j=1}^{\ell} u_j \lambda^{3k+\ell-j} - \sum_{j=0}^{\ell-1} \lambda^j x_{p-1+\ell-j}^2 \right) \lambda^\xi + \sum_{j=1}^{\xi} u_j'' \lambda^{3k+\xi-j} - \sum_{j=0}^{\xi-1} \lambda^j x_{p+\ell-1+\xi-j}^2 \right) \varepsilon + \sum_{i=2}^{\ell+\xi} a_i' \varepsilon^i$$

$$= x_{p+\ell+\xi} + \left( \sum_{j=1}^{\ell} u_j \lambda^{3k+\ell+\xi-j} - \sum_{j=0}^{\ell-1} \lambda^{j+\xi} x_{p-1+\ell-j}^2 + \sum_{j=1}^{\xi} u_j'' \lambda^{3k+\xi-j} - \sum_{j=0}^{\xi-1} \lambda^j x_{p+\ell-1+\xi-j}^2 \right) \varepsilon + \sum_{i=2}^{\ell+\xi} a_i' \varepsilon^i$$

$$= x_{p+\ell+\xi} + \left( \sum_{j=1}^{\ell} u_j \lambda^{3k+\ell+\xi-j} - \sum_{j=\xi}^{\ell+\xi-1} \lambda^j x_{p-1+\ell+\xi-j}^2 + \sum_{j=1+\ell}^{\xi+\ell} u_{j-\ell}'' \lambda^{3k+\xi+\ell-j} - \sum_{j=0}^{\xi-1} \lambda^j x_{p+\ell-1+\xi-j}^2 \right) \varepsilon + \sum_{i=2}^{\ell+\xi} a_i' \varepsilon^i$$

$$= x_{p+\ell+\xi} + \left( \sum_{j=1}^{\ell+\xi} v_j \lambda^{3k+\ell+\xi-j} - \sum_{j=0}^{\ell+\xi-1} \lambda^j x_{p-1+\ell+\xi-j}^2 \right) \varepsilon + \sum_{i=2}^{\ell+\xi} a_i' \varepsilon^i$$

for some coefficients $e_i$, and where $v_j = u_j$ for $j \in \{1, ..., \ell\}$ and $v_j = u_{j-\ell}''$ for $j \in \{\ell+1, ..., \ell+\xi\}$.

From Lemma C.8 we find:

$$\max\{z \mid (z_{p+1}, z) \in U(x_{p+\ell}, z_{p+\ell}', x_{p+\ell+1}, z_{p+\ell+1}'))$$

$$= x_{p+\ell+1+\xi} + \left( \left( \sum_{j=1}^{\ell} u_j' \lambda^{3k+\ell-j} - \sum_{j=0}^{\ell-1} \lambda^j x_{p+\ell-j}^2 \right) \lambda^\xi + \sum_{j=1}^{\xi} u_j''' \lambda^{3k+\xi-j} - \sum_{j=0}^{\xi-1} \lambda^j x_{p+\ell+\xi-j}^2 \right) \varepsilon + \sum_{i=2}^{\ell+\xi} a_i'' \varepsilon^i$$

$$= x_{p+\ell+1+\xi} + \left( \sum_{j=1}^{\ell} u_j' \lambda^{3k+\ell+\xi-j} - \sum_{j=0}^{\ell-1} \lambda^{j+\xi} x_{p+\ell-j}^2 + \sum_{j=1}^{\xi} u_j''' \lambda^{3k+\xi-j} - \sum_{j=0}^{\xi-1} \lambda^j x_{p+\ell+\xi-j}^2 \right) \varepsilon + \sum_{i=2}^{\ell+\xi} a_i'' \varepsilon^i$$

$$= x_{p+\ell+1+\xi} + \left( \sum_{j=1}^{\ell} u_j' \lambda^{3k+\ell+\xi-j} - \sum_{j=\xi}^{\ell+\xi-1} \lambda^j x_{p+\ell+\xi-j}^2 + \sum_{j=\ell+1}^{\xi+\ell} u_{j-\ell}''' \lambda^{3k+\xi+\ell-j} - \sum_{j=0}^{\xi-1} \lambda^j x_{p+\ell+\xi-j}^2 \right) \varepsilon + \sum_{i=2}^{\ell+\xi} a_i'' \varepsilon^i$$

$$= x_{p+\ell+1+\xi} + \left( \sum_{j=1}^{\ell+\xi} w_j \lambda^{3k+\ell+\xi-j} - \sum_{j=0}^{\ell+\xi-1} \lambda^j x_{p+\ell+\xi-j}^2 \right) \varepsilon + \sum_{i=2}^{\ell+\xi} a_i'' \varepsilon^i$$

for some coefficients $f_i$, and where $w_j = u_j'$ for $j \in \{1, ..., \ell\}$ and $w_j = u_{j-\ell}'''$ for $j \in \{\ell+1, ..., \ell+\xi\}$. We thus have:

$$\max\{z \,|\, (x_p, z) \in U(x_p, z_p, x_{p+1}, z_{p+1}) \diamond U(x_{p+\ell}, z_{p+\ell}', x_{p+\ell+1}, z_{p+\ell+1}')\} = z_p''$$
$$\max\{z \,|\, (x_{p+1}, z) \in U(x_p, z_p, x_{p+1}, z_{p+1}) \diamond U(x_{p+\ell}, z_{p+\ell}', x_{p+\ell+1}, z_{p+\ell+1}')\} = z_{p+1}''$$

The result now follows from the linearity of the upper bounds in the regions $U(x_p, z_p, x_{p+1}, z_{p+1})$, $U(x_{p+\ell}, z_{p+\ell}', x_{p+\ell+1}, z_{p+\ell+1}')$ and $U(x_p, z_p'', x_{p+1}, z_{p+1}'')$. $\qquad\square$

**Lemma C.12.** *Let* $j \in \{1, ..., k\}$ *and* $\ell \geq 1$. *Let* $I_1, ..., I_\ell \subseteq \{1, ..., k\}$. *It holds that:*

$$(x_j, z) \in R_{I_1} \diamond ... \diamond R_{I_\ell} \Rightarrow z \geq x^* + (x_j - x^*) \left( \frac{x^* - y_1}{x^*} \right)^\ell$$

*Proof.* Let us write $z_i$ for the bound corresponding to $\ell = i$. We show the result by induction. For the base case, we find

$$\begin{aligned}
z_1 &= \min\{z \,|\, (x_j, z) \in R_{I_1}\} \\
&= \min\{z \,|\, (x_j, z) \in L(x_1, y_1, x^*, x^*)\} \\
&= \min\{z \,|\, z - y_1 \geq \frac{x^* - y_1}{x^* - x_1}(x_j - x_1)\} \\
&= y_1 + \frac{x^* - y_1}{x^* - x_1}(x_j - x_1) \\
&= y_1 + \left( \frac{x^* - y_1}{x^*} \right) x_j \\
&= x^* - \left( \frac{x^* - y_1}{x^*} \right) x^* + \left( \frac{x^* - y_1}{x^*} \right) x_j \\
&= x^* + (x_j - x^*) \left( \frac{x^* - y_1}{x^*} \right)
\end{aligned}$$

Now suppose the result already holds for $\ell = i - 1$. We find:

$$\begin{aligned}
&\min\{z \,|\, (x_j, z) \in R_{I_1} \circ R_{I_2} \diamond ... \diamond R_{I_i}\} \\
&\min\{z \,|\, (z_{i-1}, z) \in R_{I_i}\} \\
&= \min\{z \,|\, z - y_1 \geq \frac{x^* - y_1}{x^* - x_1}(z_{i-1} - x_1)\} \\
&= y_1 + \frac{x^* - y_1}{x^* - x_1}(z_{i-1} - x_1) \\
&= y_1 + \frac{x^* - y_1}{x^*} z_{i-1} \\
&= y_1 + \frac{x^* - y_1}{x^*} \left( x^* + (x_j - x^*) \left( \frac{x^* - y_1}{x^*} \right)^{i-1} \right) \\
&= x^* + (x_j - x^*) \left( \frac{x^* - y_1}{x^*} \right)^i
\end{aligned}$$

$\qquad\square$

# D. Proof of Proposition 5.1

Let $\mathcal{P}$ be a set of closed path rules, and let $\mathcal{R}$ be the set of relations appearing in $\mathcal{P}$. We will construct a relation embedding $(\eta, \eta')$ that captures inferences from $\mathcal{P}$ up to depth $m$. We will construct this embedding by incrementally adding coordinates. Each coordinate will ensure that some unwanted rules are not captured, while ensuring that the rules that can be entailed from $\mathcal{P}$ (up to depth $m$) remain captured by the embedding.

For every relation $r$ there are only a finite number of rules $r_1 \circ ... \circ r_k \subseteq r$ such that $\mathcal{P} \models_m r_1 \circ ... \circ r_k \subseteq r$. For a relational composition $\alpha = r_1 \circ ... \circ r_k$, we write $size(\alpha) = k$ to denote the number of relations. Let us furthermore define:

$$bound_r = \max\{size(\alpha) \,|\, \mathcal{P} \models_m \alpha \subseteq r\}$$

In other words, $bound_r$ is the size of the longest relational composition which entails $r$ (up to depth $m$). We choose the first coordinate of the embeddings such that all rules of the form $\alpha \subseteq r$ with $size(\alpha) > bound_r$ are excluded. Specifically, we define ($r \in \mathcal{R}$):

$$\eta_1(r) = \{(x, y) \in \mathbb{R}^2 \,|\, 0 \leq y - x \leq 1\}$$
$$\eta_1'(r) = \{(x, y) \in \mathbb{R}^2 \,|\, 0 \leq y - x \leq bound_r\}$$

We have the following result.

**Lemma D.1.** *We have $\eta_1(r_1) \diamond ... \diamond \eta_1(r_k) \subseteq \eta_1'(r)$ iff $k \leq bound_r$.*

*Proof.* It is straightforward to verify that

$$\eta_1(r_1) \diamond ... \diamond \eta_1(r_k) = \{(x, y) \,|\, 0 \leq y - x \leq k\}$$

from which the result immediately follows. $\square$

**Corollary D.2.** *If $\mathcal{P} \models_m r_1 \circ ... \circ r_k \subseteq r$ then $\eta_r(r_1) \diamond ... \diamond \eta_r(r_k) \subseteq \eta_2(r)$.*

At this point, we have relation embedding which captures all the rules that can be inferred from $\mathcal{P}$ up to depth $m$. Moreover, there are only finitely many rules that are captured by the relation embedding while not being entailed by $\mathcal{P}$ up to depth $m$. Each of the coordinates that will be added in the construction below is aimed at ensuring that one of these unwanted rules is no longer captured.

Assume we have already defined the embeddings for coordinates $1, ..., \ell - 1$, such that every rule entailed by $\mathcal{P}$ up to depth $m$ is captured by this $\ell - 1$ dimensional relation embedding. Let $r_1 \circ ... \circ r_k \subseteq r$ be such that $\eta_j(r_1) \diamond ... \diamond \eta_j(r_k) \subseteq \eta_j'(r)$ for every $j \in \{1, ..., \ell - 1\}$ while $\mathcal{P} \not\models_m r_1 \circ ... \circ r_k \subseteq r$. We will define the relation embeddings $\eta$ and $\eta'$ in coordinate $\ell$ such that such that $\eta_\ell(r_1) \diamond ... \diamond \eta_\ell(r_k) \not\subseteq \eta_\ell'(r)$, while ensuring that every rule which is entailed by $\mathcal{P}$ is still captured by the embedding.

We define the relation embeddings $\eta_\ell(s)$ as follows:

- For $s \notin \{r_1, ..., r_k\}$ we define $\eta_\ell(s) = \{(x^*, x^*)\}$.

- We define $\eta_\ell(r) = R_{I_{r_1}} = \text{Reg}(x_1, y_1^{(1)}, ..., x_k, y_k^{(1)}; x^*)$.

- For $i \in \{2, ..., k\}$, we define $\eta_\ell(r_i) = R_{I_{r_i}} = \text{Reg}(x_1, y_1^{(i)}, ..., x_k, y_k^{(i)}; x^*) \cap ([\varepsilon, x^*] \times \mathbb{R})$.

where the parameters $I_{r_i}$ of the region $R_{I_{r_i}}$ are chosen as follows:

$$I_{r_i} = \{j \,|\, r_j = r_i, j > 1\}$$

We exclude 1 from $I_{r_i}$ to ensure that $y_1^{(i)} \leq x_2$.

The embedding $\eta_\ell'$ is defined as follows ($s \in \mathcal{R}$):

$$\eta_\ell'(s) = CH\left(\bigcup\{\eta_\ell(s_1) \diamond ... \diamond \eta_\ell(s_p) \,|\, \mathcal{P} \models_m s_1 \circ ... \circ s_p \subseteq s\}\right) \tag{8}$$

where *CH* denotes the convex hull operator.

We clearly have that every rule which is entailed by $\mathcal{P}$ up to depth $m$ is still captured by the resulting $\ell$-dimensional relation embedding. For the ease of presentation, let us write $\eta_\ell(r) = A_r$ and $\eta'_\ell(r) = A'_r$. We need to show that

$$A_{r_1} \diamond ... \diamond A_{r_k} \nsubseteq A'_r$$

In particular, we will show that there is some point $(x_1, z^*) \in (A_{r_1} \diamond ... \diamond A_{r_k}) \setminus A'_r$.

**Lemma D.3.** *It holds that $(x_1, z^*) \in A_{r_1} \diamond ... \diamond A_{r_k}$, with $z^* \in \mathbb{R}$ given by:*

$$z^* = x_{k+1} + (\sum_{j=2}^{k} \lambda^{4k-j} - \sum_{j=0}^{k-1} \lambda^j x_{k-j}^2)\varepsilon + \sum_{i=2}^{k-1} a'_i \varepsilon^i \tag{9}$$

*for some coefficients $a_2, ..., a_{k-1}$ which do not depend on $\varepsilon$.*

*Proof.* We have $(x_1, 1) \in A_{r_1}$. It thus suffices to show that

$$(1, z^*) \in A_{r_2} \diamond ... \diamond A_{r_k}$$

From Lemma C.10 it follows that this is the case for

$$z^* = x_{k+1} + (\sum_{j=1}^{k-1} u_{1+j,j} \lambda^{4k-1-j} - \sum_{j=0}^{k-2} \lambda^j x_{k-j}^2)\varepsilon + \sum_{i=2}^{k-1} a'_i \varepsilon^i$$

$$= x_{k+1} + (\sum_{j=2}^{k} u_{j,j-1} \lambda^{4k-j} - \sum_{j=0}^{k-2} \lambda^j x_{k-j}^2)\varepsilon + \sum_{i=2}^{k-1} a'_i \varepsilon^i$$

$$= x_{k+1} + (\sum_{j=2}^{k} \lambda^{4k-j} - \sum_{j=0}^{k-2} \lambda^j x_{k-j}^2)\varepsilon + \sum_{i=2}^{k-1} a'_i \varepsilon^i$$

$$= x_{k+1} + (\sum_{j=2}^{k} \lambda^{4k-j} - \sum_{j=0}^{k-1} \lambda^j x_{k-j}^2)\varepsilon + \sum_{i=2}^{k-1} a'_i \varepsilon^i$$

where the penultimate step follows because, as per Lemma C.10, $u_{j,j-1} = 1$ if $j \in I_j$ (noting that the $j^{th}$ region in the composition is $A_{r_{j+1}}$). The last step follows from $x_1 = 0$. $\qquad\square$

It remains to be shown that $(x_1, z^*) \notin A'_r$. To this end, it is sufficient to show that $(x_1, z^*) \notin A_{s_1} \diamond ... \diamond A_{s_p}$ for any $s_1, ..., s_p$ such that $\mathcal{P} \models_m s_1 \circ ... \circ s_p \subseteq r$. We show this separately for the case where $p \leq k$ (Lemma D.4) and for the case where $p > k$ (Lemma D.6).

**Lemma D.4.** *Let $s_1, ..., s_p \in \mathcal{R}$. Assume that $p \leq k$ and $(s_1, ..., s_p) \neq (r_1, ..., r_k)$ (i.e. if $p = k$ then $s_i \neq r_i$ for some $i \in \{1, ..., p\}$). It holds that $(x_1, z^*) \notin A_{s_1} \diamond ... \diamond A_{s_p}$ with $z^*$ defined as in Lemma D.3.*

*Proof.* If there is some $i \in \{1, ..., p\}$ such that $s_i \notin \{r_1, ..., r_k\}$, then we have $A_{s_1} \diamond ... \diamond A_{s_p} = \{(x^*, x^*)\}$ and the result holds trivially. Furthermore, if $s_1 \neq r_1$ then $(x_1, z) \notin A_{s_1}$ for any $z \in \mathbb{R}$, and thus also $(x_1, z) \notin A_{s_1} \diamond ... \diamond A_{s_p}$ for any $z$. Let us thus assume that $s_i \in \{r_1, ..., r_k\}$ for every $i \in \{1, ..., p\}$ and that $r_1 = s_1$.

By repeatedly applying Lemma C.11 we then find:

$$A_{s_1} \diamond ... \diamond A_{s_p} \subseteq U(x_1, z_1, x_2, z_2)$$

where

$$z_1 = x_{p+1} + \left(\sum_{j=1}^{p} v_j \lambda^{3k+p-j} - \sum_{j=0}^{p-1} \lambda^j x_{p-j}^2\right)\varepsilon + \sum_{i=2}^{p} a_i \varepsilon^i$$

$$z_2 = x_{p+2} + \left( \sum_{j=1}^{p} w_j \lambda^{3k+p-j} - \sum_{j=0}^{p-1} \lambda^j x_{p+1-j}^2 \right) \varepsilon + \sum_{i=2}^{p} b_i \varepsilon^i$$

for some coefficients $a_i, b_i$ which do not depend on $\varepsilon$, and where the coefficients $v_j$ are defined as:

$$v_j = \begin{cases} 1 & \text{if } s_j = r_l \text{ such that } j \in I_l \\ 0 & \text{otherwise} \end{cases}$$

$$= \begin{cases} 1 & \text{if } s_j = r_j \\ 0 & \text{otherwise} \end{cases}$$

The coefficients $w_j$ are defined similarly, but they do not play any role in this proof.

We have that $(x, z) \in A_{s_1} \diamond ... \diamond A_{s_p}$ can only hold if $x \leq z_1$. If $p < k$ we have $x_{p+1} < x_{k+1}$ and thus $z_1 < z^*$ for sufficiently small $\varepsilon$. Now assume $p = k$. Then we can only have $z^* \leq z_1$, for sufficiently small $\varepsilon$, if

$$\sum_{j=1}^{k} v_j \lambda^{4k+p-j} \geq \sum_{j=2}^{k} \lambda^{4k-j}$$

Since $s_1 = r_1$ we have $v_1 = 0$. The inequality can thus only hold if $v_2 = ... = v_k = 1$, in which case we have $(s_2, ..., s_k) = (r_2, ..., r_k)$. We have thus shown that whenever $(s_1, ..., s_k) \neq (r_1, ..., r_k)$ we have $(x_1, z^*) \notin A_{s_1} \diamond ... \diamond A_{s_p}$. $\square$

**Lemma D.5.** *Assume that $\varepsilon$, $\lambda$ and $\delta$ are sufficiently small. If $\ell \geq k - j + 1$, it holds that:*

$$x^* + (x_j - x^*) \left( \frac{x^* - 1}{x^*} \right)^\ell > x_k + \frac{1}{2} \lambda^{k-1}$$

*Proof.* We find:

$$x^* + (x_j - x^*) \left( \frac{x^* - 1}{x^*} \right)^\ell > x_k + \frac{1}{2} \lambda^{k-1}$$

$$\Leftrightarrow x_{k+1} - \delta + (x_j - x^*) \left( \frac{x^* - 1}{x^*} \right)^\ell > x_k + \frac{1}{2} \lambda^{k-1}$$

$$\Leftrightarrow \lambda^{k-1} - \delta + (x_j - x^*) \left( \frac{x^* - 1}{x^*} \right)^\ell > \frac{1}{2} \lambda^{k-1}$$

$$\Leftrightarrow \frac{1}{2} \lambda^{k-1} + (x_j - x^*) \left( \frac{x^* - 1}{x^*} \right)^\ell > \delta$$

A suitable $\delta$ can always be found as long as the left-hand side is strictly positive. We find:

$$\frac{1}{2} \lambda^{k-1} + (x_j - x^*) \left( \frac{x^* - 1}{x^*} \right)^\ell > 0$$

$$\Leftrightarrow \frac{1}{2} \lambda^{k-1} > (x^* - x_j) \left( \frac{x^* - 1}{x^*} \right)^\ell$$

$$\Leftrightarrow \frac{1}{2} \lambda^{k-1} > \frac{(x^* - x_j)(x^* - 1)^\ell}{(x^*)^\ell}$$

Since the denominator in the right-hand side is larger than 1 (for sufficiently small $\delta$), it is sufficient to show:

$$\frac{1}{2} \lambda^{k-1} > (x^* - x_j)(x^* - 1)^\ell$$

$$\Leftrightarrow \frac{1}{2} \lambda^{k-1} > (\lambda^{j-1} + ... + \lambda^{k-1} - \delta)(\lambda + ... + \lambda^{k-1} - \delta)^\ell$$

Since $\delta > 0$ and $\lambda > 0$ it is furthermore sufficient to show

$$\frac{1}{2}\lambda^{k-1} > \lambda^{j-1}\lambda^{\ell}$$

Since we can assume that $\lambda < \frac{1}{2}$, this is satisfied as soon as $k - 1 < j - 1 + \ell$, or equivalently, $k - j + 1 \leq \ell$. □

**Lemma D.6.** *Let $s_1, ..., s_p \in \mathcal{R}$. Assume that $p > k$. It holds that $(x_1, z^*) \notin A_{s_1} \diamond ... \diamond A_{s_p}$ with $z^*$ defined as in Lemma D.3.*

*Proof.* As in the proof of Lemma D.4 we find that the result trivially holds if $s_i \notin \{r_1, ..., r_k\}$. Let us therefore assume that $s_i \in \{r_1, ..., r_k\}$ for every $i \in \{1, ..., p\}$.

Suppose $(x_1, z^*) \in A_{s_1} \diamond ... \diamond A_{s_p}$. Then there must be some $z'$ such that $(x_1, z') \in A_{s_1} \diamond ... \diamond A_{s_{p-1}}$ and $(z', z^*) \in A_{s_p}$. From $(x_1, z')$ we derive using Lemmas C.12 and D.5:

$$z' \geq x^* + (x_1 - x^*)\left(\frac{x^* - y_1}{x^*}\right)^p > x_k + \frac{1}{2}\lambda^{k-1}$$

We then find that for some $l \in \{1, ..., k\}$ and $u \in \{0, 1\}$ we have:

$$(z', z^*) \in A_{s_p} \Rightarrow (z', z^*) \in U(x_k, y_k^{(l)}, x^*, x^*)$$

$$\Leftrightarrow z^* - y_k^{(l)} \leq \frac{x^* - y_k^{(l)}}{x^* - x_k}(z' - x_k)$$

$$\Leftrightarrow z^* \leq x_{k+1} - (x_k^2 - u\lambda^{3k})\varepsilon + \frac{x^* - y_k^{(l)}}{x^* - x_k}(z' - x_k)$$

$$\Rightarrow z^* \leq x_{k+1} - (x_k^2 - u\lambda^{3k})\varepsilon + \frac{x^* - y_k^{(l)}}{2(x^* - x_k)}\lambda^{k-1}$$

where the last step relies on the fact that $\frac{x^* - y_k^{(l)}}{x^* - x_k} < 0$. This also means that, for $\varepsilon$ sufficiently small, the latter inequality cannot be satisfied. □

We have shown that $(x_1, z^*) \notin A_{s_1} \diamond ... \diamond A_{s_p}$ for any $s_1, ..., s_p$ such that $\mathcal{P} \models_m s_1 \circ ... \circ s_p \subseteq r$. Since all points $(x, z)$ in these regions are such that $x \geq x_1$, and all points of the form $(x_1, z)$ are such that $z \leq z^*$, it follows that $(x_1, z^*)$ is not in the convex hull of these regions $A_{s_1} \diamond ... \diamond A_{s_p}$, and in particular that $(x_1, z^*) \notin A'_r$.

## E. Proof of Proposition 5.4

We follow the same high-level strategy as in the proof of Proposition 5.1. In particular, we will design a relation embedding $\eta$ whose first coordinate is aimed at excluding all rules beyond a certain length. Then for each unwanted rule that is still captured by the relation embedding, we will add a coordinate aimed at excluding that rule. However, the fact that we now need an accordant relation embedding complicates the construction.

### E.1. Excluding Long Rules

Since $\mathcal{P}$ is acyclic, all rules $r_1 \circ ... \circ r_k \subseteq r_{k+1}$ entailed by $\mathcal{P}$ are bounded in size. We assign values $\gamma(r) > 0$ to each relation in $\mathcal{R}$ as follows:

- We initially set $\gamma(r) = 1$ for every $r \in \mathcal{R}$.

- For every rule $r_1 \circ ... \circ r_k \subseteq s$ in $\mathcal{P}$ we update $\gamma(r_i)$ as follows:

$$\gamma(r_i) \leftarrow \min\left(\gamma(r_i), \frac{\gamma(s)}{k}\right)$$

We repeat this update step until convergence.

Note that this process must converge after a finite number of steps, given the fact that $\mathcal{P}$ is acyclic. Using the resulting mapping $\gamma$, we define the first coordinate of the relation embedding $\eta$ as follows:

$$\eta_1(r) = \{(x,y) \mid 0 \le y - x \le \gamma(r)\}$$

Note that we then have:

$$\eta_1(r_1) \diamond ... \diamond \eta_1(r_k) = \{(x,y) \mid 0 \le y - x \le \gamma(r_1) + ... + \gamma(r_k)\}$$

The following result trivially follows from this observation:

**Lemma E.1.** *If $\mathcal{P} \models r_1 \circ ... \circ r_k \subseteq s$ then then $\eta_1(r_1) \diamond ... \diamond \eta_1(r_k) \subseteq \eta_1(s)$.*

Furthermore, we also trivially have the following result.

**Lemma E.2.** *For each $s$ there exists some $K_s \in \mathbb{N}$ such that $\eta_1(r_1) \diamond ... \diamond \eta_1(r_k) \not\subseteq \eta_1(s)$ whenever $k > K_s$.*

At this point, we have a relation embedding which captures all the rules in $\mathcal{P}$ (given Lemma E.1) and where there are only finitely many rules that are captured by the embedding while not being entailed by $\mathcal{P}$ (given Lemma E.2).

### E.2. Excluding Specific Rules

Assume we have already defined the embeddings for coordinates $1, ..., \ell - 1$, such that every rule entailed by $\mathcal{P}$ is captured by this $\ell - 1$ dimensional relation embedding. Let $r_1 \circ ... \circ r_d \subseteq r$ be such that $\eta_j(r_1) \diamond ... \diamond \eta_j(r_d) \subseteq \eta_j(r)$ for every $j \in \{1, ..., \ell - 1\}$ while $\mathcal{P} \not\models r_1 \circ ... \circ r_d \subseteq r$. We will define the relation embedding $\eta$ in coordinate $\ell$ such that such that $\eta_\ell(r_1) \diamond ... \diamond \eta_\ell(r_d) \not\subseteq \eta_\ell(r)$, while ensuring that every rule which is entailed by $\mathcal{P}$ is still captured by the embedding.

By assumption, either $r$ is regular or there exists a weighting $\omega_r$ which balances all the relations in rules that entail $r$. If $r$ is regular, we can use the construction from Charpenay & Schockaert (2024) based on octagon embeddings to define the relation embedding $\eta_\ell$. In the remainder of the proof, we therefore focus on the second case. Let us thus assume that there is a weighting $\omega : \mathcal{R} \to \mathbb{N} \setminus \{0\}$ such that for every $s_1, ..., s_p$ satisfying $\mathcal{P} \models s_1 \circ ... \circ s_p \subseteq r$, it holds that $s_1, ..., s_p$ are balanced by $\omega$. Let us define:

$$k = \omega(r_1) + ... + \omega(r_d) \tag{10}$$

For $i \in \{1, ..., d\}$ we define:

$$z_i = 1 + \omega(r_1) + ... + \omega(r_{i-1}) \tag{11}$$

For each $s \in \{r_1, ..., r_d\}$ we define an auxiliary region $X_s$ as follows ($i \in \{1, ..., c\}$):

$$X_{r_1} = R_{I_{r_1}} = \mathsf{Reg}(x_1, y_1^{(1)}, ..., x_k, y_k^{(1)}; x^*) \cap ([\varepsilon, x^*] \times \mathbb{R}) \tag{12}$$

while for $r_i \ne r_1$ we define:

$$X_{r_i} = R_{I_{r_i}} = \mathsf{Reg}(x_1, y_1^{(i)}, ..., x_k, y_k^{(i)}; x^*) \tag{13}$$

where for $i \in \{1, ..., d\}$ we define:

$$I_{r_i} = \{z_j \mid r_j = r_i, j > 1\}$$

For the ease of presentation we will write $\eta_\ell(r)$ as $A_r$. For $i \in \{1, ..., d\}$ we initialize $A_{r_i}$ as follows:

$$A_{r_i} = X_{r_i}^{\uparrow \omega(r_i)}$$

where we define $X^{\uparrow 1} = X$ and $X^{\uparrow i} = X \diamond X^{\uparrow i-1}$ for $i > 1$. For $s \notin \{r_1, ..., r_d\}$ we initialize:

$$A_s = \{(x^*, x^*)\}$$

The initial regions are then updated as follows, until convergence:

$$A_r \leftarrow CH\left(A_r \cup \bigcup\{A_{s_1} \diamond ... \diamond A_{s_p} \mid (s_1 \wedge ... \wedge s_p \to r) \in \mathcal{P}\}\right) \tag{14}$$

Since we assumed $\mathcal{P}$ to be acyclic, this process is guaranteed to terminate.

We clearly have that every rule which is entailed by $\mathcal{P}$ is still captured by the resulting $\ell$-dimensional relation embedding. We need to show that

$$A_{r_1} \diamond ... \diamond A_{r_k} \not\subseteq A_r$$

As in the proof of Proposition 5.1, we will show that $(x_1, z^*) \in (A_{r_1} \diamond ... \diamond A_{r_k}) \setminus A_r$ for some $z^*$. The value of $z^*$ is established in the following counterpart to Lemma D.3.

**Lemma E.3.** *It holds that* $(x_1, z^*) \in A_{r_1} \diamond ... \diamond A_{r_k}$, *with* $z^* \in \mathbb{R}$ *given by:*

$$z^* = x_{k+1} + \Big(\sum_{i=2}^{d} \lambda^{4k-z_i} - \sum_{j=0}^{k-1} \lambda^j x_{k-j}^2\Big)\varepsilon + \sum_{i=2}^{k-1} a'_i \varepsilon^i \tag{15}$$

*for some coefficients* $a_2, ..., a_{k-1}$ *which do not depend on* $\varepsilon$.

*Proof.* We have

$$A_{r_1} \diamond ... \diamond A_{r_d} \supseteq X_{r_1}^{\uparrow \omega(r_1)} \diamond ... \diamond X_{r_d}^{\uparrow \omega(r_d)}$$

It is thus sufficient to show $(x_1, z^*) \in X_{r_1}^{\uparrow \omega(r_1)} \diamond ... \diamond X_{r_d}^{\uparrow \omega(r_d)}$. We have $(x_1, 1) \in X_{r_1}$. As in the proof of Lemma D.3, we then find $(1, z^*) \in X_{r_1}^{\uparrow \omega(r_1)-1} \diamond ... \diamond X_{r_d}^{\uparrow \omega(r_d)}$ with $z^*$ given by:

$$z^* = x_{k+1} + \Big(\sum_{j=2}^{k} u_{j,j-1}\lambda^{4k-j} - \sum_{j=0}^{k-2} \lambda^j x_{k-j}^2\Big)\varepsilon + \sum_{i=2}^{k-1} a'_i \varepsilon^i$$

where $u_{j,j-1} = 0$ unless $j = z_i$ for $i \in \{2, ..., d\}$. We thus find:

$$z^* = x_{k+1} + \Big(\sum_{i=2}^{d} \lambda^{4k-z_i} - \sum_{j=0}^{k-1} \lambda^j x_{k-j}^2\Big)\varepsilon + \sum_{i=2}^{k} a'_i \varepsilon^i$$

$\square$

**Lemma E.4.** *Let* $i \in \{1, ..., d\}$, $l \in \{1, ..., k-1\}$ *and* $m \in \{1, ..., k-l\}$. *Let us define* $u_q = 1$ *if* $q \in I_{r_i}$ *and* $u_q = 0$ *otherwise. It holds that:*

$$X_{r_i}^{\uparrow m} \subseteq U(x_l, z_l, x_{l+1}, z_{l+1})$$

*with*

$$z_l = x_{l+m} + \Big(\sum_{j=1}^{m} u_{l+j-1}\lambda^{3k+m-j} - \sum_{j=0}^{m-1} \lambda^j x_{l+m-j-1}^2\Big)\varepsilon + \sum_{j=2}^{m} a'_j \varepsilon^j$$

$$z_{l+1} = x_{l+m+1} + \Big(\sum_{j=1}^{m} u_{l+j}\lambda^{3k+m-j} - \sum_{j=0}^{m-1} \lambda^j x_{l+m-j}^2\Big)\varepsilon + \sum_{j=2}^{m} a''_j \varepsilon^j$$

*for some coefficients* $a'_2, ..., a'_m, a''_2, ..., a''_m$ *which do not depend on* $\varepsilon$.

*Proof.* By construction, for $q \in \{0, ..., k-l-1\}$, we have that

$$X_{r_i} \subseteq U(x_{l+q}, z_{l+q}^{(1)}, x_{l+q+1}, z_{l+q+1}^{(1)})$$

where for $q \in \{0, ..., k-l\}$

$$z_{l+q}^{(1)} = x_{l+q+1} + (u_{l+q}\lambda^{3k} - x_{l+q}^2)\varepsilon$$

We show the result by induction. The base case, where $m = 1$, follows immediately from the characterization of $z_{l+q}^{(1)}$ for $q = 0$ and $q = 1$. Now suppose the result is already known to hold for $X_{r_i}^{\uparrow m-1}$. We thus have $X_{r_i}^{\uparrow m-1} \subseteq U(x_l, z_l^{(m-1)}, x_{l+1}, z_{l+1}^{(m-1)})$ with

$$z_l^{(m-1)} = x_{l+m-1} + \Big( \sum_{j=1}^{m-1} u_{l+j-1}\lambda^{3k+m-1-j} - \sum_{j=0}^{m-2} \lambda^j x_{l+m-j-2}^2 \Big)\varepsilon + \sum_{j=2}^{m-1} b_j^{(m-1)}\varepsilon^j$$

$$z_{l+1}^{(m-1)} = x_{l+m} + \Big( \sum_{j=1}^{m-1} u_{l+j}\lambda^{3k+m-1-j} - \sum_{j=0}^{m-2} \lambda^j x_{l+m-1-j}^2 \Big)\varepsilon + \sum_{j=2}^{m-1} c_j^{(m-1)}\varepsilon^j$$

for some coordinates $b_j^{(m-1)}$ and $c_j^{(m-1)}$. We furthermore have $X_{r_i} \subseteq U(x_{l+m-1}, z_{l+m-1}^{(1)}, x_{l+m}, z_{l+m}^{(1)})$ with $z_{l+m-1}^{(1)}$ and $z_{l+m}^{(1)}$ as defined before. By applying Lemma C.11, we then find:

$$X_{r_i}^{\uparrow m} = X_{r_i}^{\uparrow m-1} \diamond X_{r_i} \subseteq U(x_l, z_l, x_{l+1}, z_{l+1})$$

with

$$z_l = x_{l+m} + \Big( \sum_{j=1}^{m} u_{l+j-1}\lambda^{3k+m-j} - \sum_{j=0}^{m-1} \lambda^j x_{l+m-j-1}^2 \Big)\varepsilon + \sum_{j=2}^{m} a_j'\varepsilon^j$$

$$z_{l+1} = x_{l+m+1} + \Big( \sum_{j=1}^{m} u_{l+j}\lambda^{3k+m-j} - \sum_{j=0}^{m-1} \lambda^j x_{l+m-j}^2 \Big)\varepsilon + \sum_{j=2}^{m} a_j''\varepsilon^j$$

for some coefficients $a_j'$ and $a_j''$. □

**Lemma E.5.** *Let $s_1, ..., s_p \in \mathcal{R}$. Let $m = \omega(s_1) + ... + \omega(s_p)$ and $l \leq k - m$. Assume that for each $i \in \{1, ..., p\}$ it holds that:*

$$A_{s_i} \subseteq U(x_{l_i}, z_{l_i}, x_{l_i+1}, z_{l_i+1})$$

*with $l_i = l + \omega(s_1) + ... + \omega(s_{i-1})$ and*

$$z_{l_i} = x_{l_i+\omega(s_i)} + \Big( \sum_{j=1}^{\omega(s_i)} u_{l_i+j-1}\lambda^{3k+\omega(s_i)-j} - \sum_{j=0}^{\omega(s_i)-1} \lambda^j x_{l_i+\omega(s_i)-j-1}^2 \Big)\varepsilon + \sum_{j=2}^{\omega(s_i)} a_j'\varepsilon^j$$

$$z_{l_i+1} = x_{l_i+\omega(s_i)+1} + \Big( \sum_{j=1}^{\omega(s_i)} u_{l_i+j}'\lambda^{3k+\omega(s_i)-j} - \sum_{j=0}^{\omega(s_i)-1} \lambda^j x_{l_i+\omega(s_i)-j}^2 \Big)\varepsilon + \sum_{j=2}^{\omega(s_i)} a_j''\varepsilon^j$$

*for some coefficients $a_j', a_j''$ which do not depend on $\varepsilon$, and some constants $u_{l_i+j-1}, u_{l_i+j}' \in \{0, 1\}$. Then it holds that:*

$$A_{s_1} \diamond ... \diamond A_{s_p} \subseteq U(x_l, z_l', x_{l+1}, z_{l+1}')$$

*with*

$$z_l' = x_{l+m} + \Big( \sum_{j=1}^{m} u_{l+j-1}\lambda^{3k+m-j} - \sum_{j=0}^{m-1} \lambda^j x_{l+m-j-1}^2 \Big)\varepsilon + \sum_{j=2}^{m} b_j'\varepsilon^j$$

$$z_{l+1}' = x_{l+m+1} + \Big( \sum_{j=1}^{m} u_{l+j}'\lambda^{3k+m-j} - \sum_{j=0}^{m-1} \lambda^j x_{l+m-j}^2 \Big)\varepsilon + \sum_{j=2}^{m} b_j''\varepsilon^j$$

*for some coefficients $b_j', b_j''$ which do not depend on $\varepsilon$.*

*Proof.* We show the result by induction. The base case, where $p = 1$, is trivial. For the inductive step, suppose we have already established that:

$$A_{s_1} \diamond ... \diamond A_{s_{p-1}} \subseteq U(x_l, z_l', x_{l+1}, z_{l+1}')$$

with

$$z_l' = x_{l+m-\omega(s_p)} + \Big( \sum_{j=1}^{m-\omega(s_p)} u_{l+j-1} \lambda^{3k+m-\omega(s_p)-j} - \sum_{j=0}^{m-1-\omega(s_p)} \lambda^j x_{l+m-j-1-\omega(s_p)}^2 \Big) \varepsilon + \sum_{j=2}^{m-\omega(s_p)} a_j' \varepsilon^j$$

$$z_{l+1}' = x_{l+m+1-\omega(s_p)} + \Big( \sum_{j=1}^{m-\omega(s_p)} u_{l+j}' \lambda^{3k+m-\omega(s_p)-j} - \sum_{j=0}^{m-1-\omega(s_p)} \lambda^j x_{l+m-j-\omega(s_p)}^2 \Big) \varepsilon + \sum_{j=2}^{m-\omega(s_p)} a_j'' \varepsilon^j$$

Together with the fact that $A_{s_p} \subseteq U(x_{l_p}, z_{l_p}, x_{l_p+1}, z_{l_p+1})$ the result then follows from Lemma C.11. $\qquad\square$

**Lemma E.6.** *Let $k \geq 2$, $p \in \{1, ..., k\}$ and $\ell \in \{1, ..., k - p + 1\}$. Assume that:*

$$y_p = x_{p+\ell} + \Big( \sum_{j=1}^{\ell} u_{j,p} \lambda^{3k+\ell-j} - \sum_{j=0}^{\ell-1} \lambda^j x_{p-1+\ell-j}^2 \Big) \varepsilon + \sum_{i=2}^{\ell} a_i^{(p)} \varepsilon^i$$

*for some coefficients $a_i^{(p)} \in \mathbb{R}$ which do not depend on $\varepsilon$, and some values $u_{j,p} \in \{0, 1\}$. If $\delta$, $\lambda$ and $\varepsilon$ are sufficiently small, for each $i \in \{1, ..., k\}$, it holds that $(x_i, y_i) \in U(x_j, y_j, x_{j+1}, y_{j+1})$ for $j \in \{1, ..., k - 1\}$.*

*Proof.* First note that when $\lambda$ and $\varepsilon$ are sufficiently small, we clearly have that $y_1 < y_2 < ... < y_k$. Furthermore, we clearly have that $(x_i, y_i) \in U(x_i, y_i, x_{i+1}, y_{i+1})$ and (for $i > 1$) that $(x_i, y_i) \in U(x_{i-1}, y_{i-1}, x_i, y_i)$. We now show that $(x_i, y_i) \in U(x_j, y_j, x_{j+1}, y_{j+1})$ for $j \in \{1, ..., k - 1\} \setminus \{i, i - 1\}$. We find using (7):

$$(x_i, y_i) \in U(x_j, y_j, x_{j+1}, y_{j+1})$$

$$\Leftrightarrow y_i - y_j \leq \frac{y_{j+1} - y_j}{x_{j+1} - x_j}(x_i - x_j)$$

$$\Leftrightarrow \frac{y_i \lambda^{j-1}}{y_{j+1} - y_j} \leq \frac{y_j \lambda^{j-1}}{y_{j+1} - y_j} + x_i - x_j$$

$$\Leftrightarrow y_i - y_j \leq \frac{(x_i - x_j)(y_{j+1} - y_j)}{\lambda^{j-1}}$$

$$\Leftrightarrow x_{i+\ell} - x_{j+\ell} + \Big( \sum_{q=1}^{\ell}(u_{q,i} - u_{q,j})\lambda^{3k+\ell-q} - \sum_{q=0}^{\ell-1} \lambda^q (x_{i-1+\ell-q}^2 - x_{j-1+\ell-q}^2) \Big) \varepsilon + \sum_{i=2}^{l} a_i' \varepsilon^i$$

$$\leq \frac{(x_i - x_j)\Big(x_{j+1+\ell} - x_{j+\ell} + \big( \sum_{q=1}^{\ell}(u_{q,j+1} - u_{q,j})\lambda^{3k+\ell-q} - \sum_{q=0}^{\ell-1} \lambda^q (x_{j+\ell-q}^2 - x_{j-1+\ell-q}^2) \big) \varepsilon\Big)}{\lambda^{j-1}}$$

$$\Leftrightarrow \Big( \sum_{q=1}^{\ell}(u_{q,i} - u_{q,j})\lambda^{3k+\ell-q} - \sum_{q=0}^{\ell-1} \lambda^q (x_{i-1+\ell-q}^2 - x_{j-1+\ell-q}^2) \Big) \varepsilon + \sum_{i=2}^{l} a_i' \varepsilon^i$$

$$\leq \frac{(x_i - x_j)\Big( \sum_{q=1}^{\ell}(u_{q,j+1} - u_{q,j})\lambda^{3k+\ell-q} - \sum_{q=0}^{\ell-1} \lambda^q (x_{j+\ell-q}^2 - x_{j-1+\ell-q}^2) \Big) \varepsilon}{\lambda^{j-1}}$$

$$\Leftrightarrow \Big( \sum_{q=1}^{\ell}(u_{q,i} - u_{q,j})\lambda^{3k+\ell-q} - \sum_{q=0}^{\ell-1} \lambda^q (x_{i-1+\ell-q}^2 - x_{j-1+\ell-q}^2) \Big) \varepsilon + \sum_{i=2}^{l} a_i' \varepsilon^i$$

$$\leq \frac{(x_i - x_j)\Big( \sum_{q=1}^{\ell}(u_{q,j+1} - u_{q,j})\lambda^{3k+\ell-q} - \sum_{q=0}^{\ell-1} \lambda^{j+\ell-2}(x_{j+\ell-q} + x_{j-1+\ell-q}) \Big) \varepsilon}{\lambda^{j-1}}$$

$$\Leftrightarrow \sum_{q=1}^{\ell}(u_{q,i} - u_{q,j})\lambda^{3k+\ell-q} - \sum_{q=0}^{\ell-1} \lambda^q (x_{i-1+\ell-q}^2 - x_{j-1+\ell-q}^2) + \sum_{i=2}^{l} a_i' \varepsilon^{i-1}$$

$$\leq (x_i - x_j)\Big(\sum_{q=1}^{\ell}(u_{q,j+1} - u_{q,j})\lambda^{3k+\ell-q-j+1} - \sum_{q=0}^{\ell-1}\lambda^{\ell-1}(x_{j+\ell-q} + x_{j-1+\ell-q})\Big)$$

$$\Leftrightarrow \sum_{q=1}^{\ell}(u_{q,i} - u_{q,j})\lambda^{3k+\ell-q} - (x_i - x_j)\Big(\sum_{q=1}^{\ell}(u_{q,j+1} - u_{q,j})\lambda^{3k+\ell-q-j+1}\Big) + \sum_{i=2}^{l}a_i'\varepsilon^{i-1}$$

$$\leq \sum_{q=0}^{\ell-1}\lambda^q(x_{i-1+\ell-q}^2 - x_{j-1+\ell-q}^2) - (x_i - x_j)\Big(\sum_{q=0}^{\ell-1}\lambda^{\ell-1}(x_{j+\ell-q} + x_{j-1+\ell-q})\Big)$$

First assume $i < j$. Focusing on the right-hand side, we then find:

$$\sum_{q=0}^{\ell-1}\lambda^q(x_{i-1+\ell-q}^2 - x_{j-1+\ell-q}^2) - (x_i - x_j)\Big(\sum_{q=0}^{\ell-1}\lambda^{\ell-1}(x_{j+\ell-q} + x_{j-1+\ell-q})\Big)$$

$$= -\sum_{q=0}^{\ell-1}\lambda^q(x_{i-1+\ell-q} + x_{j-1+\ell-q})(\lambda^{i-2+\ell-q} + ... + \lambda^{j-3+\ell-q})$$

$$+ (\lambda^{i-1} + ... + \lambda^{j-2})\Big(\sum_{q=0}^{\ell-1}\lambda^{\ell-1}(x_{j+\ell-q} + x_{j-1+\ell-q})\Big)$$

$$= -\sum_{q=0}^{\ell-1}(x_{i-1+\ell-q} + x_{j-1+\ell-q})(\lambda^{i-2+\ell} + ... + \lambda^{j-3+\ell}) + (\lambda^{i+\ell-2} + ... + \lambda^{j+\ell-3})\Big(\sum_{q=0}^{\ell-1}(x_{j+\ell-q} + x_{j-1+\ell-q})\Big)$$

$$= (\lambda^{i+\ell-2} + ... + \lambda^{j+\ell-3})\Big(\sum_{q=0}^{\ell-1}x_{j+\ell-q} - x_{i-1+\ell-q}\Big)$$

$$\geq (\lambda^{i+\ell-2} + ... + \lambda^{j+\ell-3})(x_{j+1} - x_i)$$

$$= (\lambda^{i+\ell-2} + ... + \lambda^{j+\ell-3})(\lambda^{i-1} + ... + \lambda^{j-1})$$

$$\geq \lambda^{2i+\ell-3}$$

In the following, we repeatedly use the following inequality:

$$\lambda^i \geq \sum_{j=i+1}^{m}\lambda^j$$

This inequality holds when $\lambda$ is sufficiently small (with $\lambda \leq \frac{1}{m-i}$ being a sufficient condition). For the left-hand side, we find:

$$\sum_{q=1}^{\ell}(u_{q,i} - u_{q,j})\lambda^{3k+\ell-q} - (x_i - x_j)\Big(\sum_{q=1}^{\ell}(u_{q,j+1} - u_{q,j})\lambda^{3k+\ell-q-j+1}\Big) + \sum_{i=2}^{l}a_i'\varepsilon^{i-1}$$

$$\leq \sum_{q=1}^{\ell}\lambda^{3k+\ell-q} - (x_i - x_j)\Big(\sum_{q=1}^{\ell}\lambda^{3k+\ell-q-j+1}\Big) + \sum_{i=2}^{l}a_i'\varepsilon^{i-1}$$

$$= \sum_{q=1}^{\ell}\lambda^{3k+\ell-q} + (\lambda^{i-1} + ... + \lambda^{j-2})\Big(\sum_{q=1}^{\ell}\lambda^{3k+\ell-q-j+1}\Big) + \sum_{i=2}^{l}a_i'\varepsilon^{i-1}$$

$$\leq \sum_{q=1}^{\ell}\lambda^{3k+\ell-q} + (\lambda^{i-1} + ... + \lambda^{j-2})\lambda^{3k-j}) + \sum_{i=2}^{l}a_i'\varepsilon^{i-1}$$

$$\leq \lambda^{3k-1} + \lambda^{3k-j+i-2} + \sum_{i=2}^{l}a_i'\varepsilon^{i-1}$$

Assuming $\lambda < \frac{1}{2}$ and $\varepsilon$ sufficiently small, we have:

$$\lambda^{3k-1} + \lambda^{3k-j+i-2} + \sum_{i=2}^{l} a_i' \varepsilon^{i-1} \leq \lambda^{2i+\ell-3}$$

given that $2i + \ell - 3 < 3k - 1$ and $2i + \ell - 3 < 3k - j + i - 2$.

Now assume $i > j$, and thus also $j < i - 1$ (since we assumed $j \notin \{i, i-1\}$). For the right-hand side, we find:

$$\sum_{q=0}^{\ell-1} \lambda^q (x_{i-1+\ell-q}^2 - x_{j-1+\ell-q}^2) - (x_i - x_j)\Big( \sum_{q=0}^{\ell-1} \lambda^{\ell-1}(x_{j+\ell-q} + x_{j-1+\ell-q}) \Big)$$

$$= \sum_{q=0}^{\ell-1} \lambda^q (x_{i-1+\ell-q} + x_{j-1+\ell-q})(\lambda^{j-2+\ell-q} + ... + \lambda^{i-3+\ell-q})$$

$$- (\lambda^{i-2} + ... + \lambda^{j-1})\Big( \sum_{q=0}^{\ell-1} \lambda^{\ell-1}(x_{j+\ell-q} + x_{j-1+\ell-q}) \Big)$$

$$= \sum_{q=0}^{\ell-1}(x_{i-1+\ell-q} + x_{j-1+\ell-q})(\lambda^{j-2+\ell} + ... + \lambda^{i-3+\ell}) - (\lambda^{i+\ell-3} + ... + \lambda^{j+\ell-2})\Big(\sum_{q=0}^{\ell-1}(x_{j+\ell-q} + x_{j-1+\ell-q})\Big)$$

$$= (\lambda^{j-2+\ell} + ... + \lambda^{i-3+\ell}) \sum_{q=0}^{\ell-1}(x_{i-1+\ell-q} - x_{j+\ell-q})$$

$$\geq (\lambda^{j-2+\ell} + ... + \lambda^{i-3+\ell})(x_i - x_{j+1})$$

$$= (\lambda^{j-2+\ell} + ... + \lambda^{i-3+\ell})(\lambda^j + ... + \lambda^{i-2})$$

$$\geq \lambda^{2j-2+\ell}$$

For the left-hand side, we find:

$$\sum_{q=1}^{\ell}(u_{q,i} - u_{q,j})\lambda^{3k+\ell-q} - (x_i - x_j)\Big(\sum_{q=1}^{\ell}(u_{q,j+1} - u_{q,j})\lambda^{3k+\ell-q-j+1}\Big) + \sum_{i=2}^{l} a_i'\varepsilon^{i-1}$$

$$\leq \sum_{q=1}^{\ell}\lambda^{3k+\ell-q} + (x_i - x_j)\Big(\sum_{q=1}^{\ell}\lambda^{3k+\ell-q-j+1}\Big) + \sum_{i=2}^{l} a_i'\varepsilon^{i-1}$$

$$= \sum_{q=1}^{\ell}\lambda^{3k+\ell-q} + (\lambda^{i-2} + ... + \lambda^{j-1})\Big(\sum_{q=1}^{\ell}\lambda^{3k+\ell-q-j+1}\Big) + \sum_{i=2}^{l} a_i'\varepsilon^{i-1}$$

$$\leq \lambda^{3k-1} + \lambda^{j-2}\lambda^{3k-j} + \sum_{i=2}^{l} a_i'\varepsilon^{i-1}$$

$$= \lambda^{3k-1} + \lambda^{3k-2} + \sum_{i=2}^{l} a_i'\varepsilon^{i-1}$$

which is satisfied, assuming $\lambda$ and $\varepsilon$ sufficiently small, since $2j - 2 + \ell < 3k - 1$ and $2j - 2 + \ell < 3k - 2$. $\square$

Let us write $\mathcal{R}_0$ for the set of all relations from $\mathcal{R}$ which do not appear in the head of any rule in $\mathcal{P}$. For $i > 0$, we define $\mathcal{R}_i$ as the set of relations $s$ from $\mathcal{R} \setminus (\mathcal{R}_0 \cup ... \cup \mathcal{R}_{i-1})$ which are such that for any rule $s_1 \circ ... \circ s_p \subseteq s$ in $\mathcal{P}$ it hold that $s_1, ..., s_p \in \mathcal{R}_0 \cup ... \cup \mathcal{R}_{i-1}$.

For the ease of presentation, we also introduce the following notation. We say that a set of closed path rules $\mathcal{P}$ entails $s_1 \circ ... \circ s_p \subseteq s_1' \circ ... \circ s_l'$, written $\mathcal{P} \models s_1 \circ ... \circ s_p \subseteq s_1' \circ ... \circ s_l'$, if there exist indices $i_1 < ... < i_{l+1}$ such that $i_{l+1} = p + 1$ and for every $j \in \{1, ..., l\}$ we have $i_j < i_{j+1}$ and

$$\mathcal{P} \models s_{i_j} \circ ... \circ s_{i_{j+1}-1} \subseteq s_j'$$

**Lemma E.7.** *Let $s \in \mathcal{R}$ and $l \leq k - m$. Assume that $A_s \neq \{(x^*, x^*)\}$. It holds that there exist relations $s_1, ..., s_p \in \{r_1, ..., r_d\}$ such that $\mathcal{P} \models s_1 \circ ... \circ s_p \subseteq s$ and:*

$$A_s \subseteq U(x_l, z_l, x_{l+1}, z_{l+1})$$

*where*

$$z_l = x_{l+m} + \Big( \sum_{j=1}^{m} u_{l+j-1} \lambda^{3k+m-j} - \sum_{j=0}^{m-1} \lambda^j x_{l+m-j-1}^2 \Big) \varepsilon + \sum_{j=2}^{m} a'_j \varepsilon^j$$

$$z_{l+1} = x_{l+m+1} + \Big( \sum_{j=1}^{m} u'_{l+j} \lambda^{3k+m-j} - \sum_{j=0}^{m-1} \lambda^j x_{l+m-j}^2 \Big) \varepsilon + \sum_{j=2}^{m} a''_j \varepsilon^j$$

*with $m = \omega(s_1) + ... + \omega(s_p)$, $a'_2, ..., a'_m, a''_2, ..., a''_m$ constants which do not depend on $\varepsilon$, $u_l, ..., u_{l+m-1} \in \{0, 1\}$, and where the constants $u'_{l+j}$ are defined as:*

$$u'_{l+j} = \begin{cases} 1 & \text{if } 0 < j \leq \omega(s_1) \text{ and } l + j \in I_{s_1} \\ 1 & \text{if } \omega(s_1) < j \leq \omega(s_1) + \omega(s_2) \text{ and } l + j \in I_{s_2} \\ ... \\ 1 & \text{if } \omega(s_1) + ... + \omega(s_{p-1}) < j \leq m \text{ and } l + j \in I_{s_p} \\ 0 & \text{otherwise} \end{cases} \tag{16}$$

*Proof.* We show the result by induction. First assume that $s \in \mathcal{R}_0$. Then there are no rules of the form $s_1 \circ ... \circ s_q \subseteq s$ in $\mathcal{P}$, which means that $A_s \neq \{(x^*, x^*)\}$ is only possible if $s = r_i$ for some $i \in \{1, ..., d\}$. By construction $A_s$ was initialized as $X_{r_i}^{\uparrow \omega(r_i)}$ and since $r_i \in \mathcal{R}_0$ this initialization was not changed by the recursive update process in (14). The result for the base case thus follows from Lemma E.4.

Now suppose that the result was already shown to hold for all relations in $\mathcal{R}_0 \cup ... \cup \mathcal{R}_{i-1}$. Let us consider the rules $s'_1 \circ ... \circ s'_q \subseteq s$ in $\mathcal{P} \cup \{s \subseteq s\}$ for which $A_{s'_1} \diamond ... \diamond A_{s'_q} \neq \{(x^*, x^*)\}$. Let $s'_1 \circ ... \circ s'_q \subseteq s$ be the rule, among all these rules, for which the combined weight of the relations in the body $\omega(s'_1) + ... + \omega(s'_q)$ is maximal. By the induction hypothesis, we know that there must be relations $s_1, ..., s_p \in \{r_1, ..., r_d\}$ such that $\mathcal{P} \models s_1 \circ ... \circ s_p \subseteq s'_1 \circ ... \circ s'_q$ and for $i \in \{1, ..., q\}$ we have

$$A_{s_i} \subseteq U(x_{l_i}, z_{l_i}, x_{l_i+1}, z_{l_i+1})$$

$l_i = l + \omega(s_1) + ... + \omega(s_{i-1})$ and

$$z_{l_i} = x_{l_i+\omega(s_i)} + \Big( \sum_{j=1}^{\omega(s_i)} u_{l_i+j-1} \lambda^{3k+\omega(s_i)-j} - \sum_{j=0}^{\omega(s_i)-1} \lambda^j x_{l_i+\omega(s_i)-j-1}^2 \Big) \varepsilon + \sum_{j=2}^{\omega(s_i)} a'_j \varepsilon^j$$

$$z_{l_i+1} = x_{l_i+\omega(s_i)+1} + \Big( \sum_{j=1}^{\omega(s_i)} u'_{l_i+j} \lambda^{3k+\omega(s_i)-j} - \sum_{j=0}^{\omega(s_i)-1} \lambda^j x_{l_i+\omega(s_i)-j}^2 \Big) \varepsilon + \sum_{j=2}^{\omega(s_i)} a''_j \varepsilon^j$$

where the weights $u'_{l_i+j}$ are given by (16). Then we know from Lemma E.5 that

$$A_{s_1} \diamond ... \diamond A_{s_p} \subseteq U(x_l, z'_l, x_{l+1}, z'_{l+1})$$

with

$$z'_l = x_{l+m} + \Big( \sum_{j=1}^{m} u_{l+j-1} \lambda^{3k+m-j} - \sum_{j=0}^{m-1} \lambda^j x_{l+m-j-1}^2 \Big) \varepsilon + \sum_{j=2}^{m} a'_j \varepsilon^j$$

$$z'_{l+1} = x_{l+m+1} + \Big( \sum_{j=1}^{m} u'_{l+j} \lambda^{3k+m-j} - \sum_{j=0}^{m-1} \lambda^j x_{l+m-j}^2 \Big) \varepsilon + \sum_{j=2}^{m} a''_j \varepsilon^j$$

If there are multiple choices for the rule $s'_1 \circ ... \circ s'_q \sqsubseteq$ with the same combined weight $\omega(s'_1) + ... + \omega(s'_q)$, we choose the one that maximizes the term $\sum_{j=1}^{m} u'_{p+j} \lambda^{3k+m-j}$.

For any rule $s''_1 \circ ... \circ s''_u \sqsubseteq s$ such that $\omega(s''_1) + ... + \omega(s''_u) < m$, for sufficiently small $\varepsilon$, we clearly have

$$A_{s''_1} \diamond ... \diamond A_{s''_u} \subseteq U(x_l, z'_l, x_{l+1}, z'_{l+1})$$

Finally consider a rule $s''_1 \circ ... \circ s''_u \sqsubseteq s$ such that $\omega(s''_1) + ... + \omega(s''_u) = m$. Then we have

$$A_{s''_1} \diamond ... \diamond A_{s''_u} \subseteq U(x_l, z''_l, x_{l+1}, z''_{l+1})$$

where $z''_l$ and $z''_{l+1}$ have the same form as $z'_l$ and $z'_{l+1}$, but where the weights $u_{p+j-1}$ and $u'_{p+j}$ are defined by the relations $s''_1, ..., s''_u$ instead. By assumption, we have $z''_{l+1} \leq z'_{l+1}$. From Lemma E.6 it then follows that:

$$\mathrm{CH}(A_{s_1} \diamond ... \diamond A_{s_p}, A_{s''_1} \diamond ... \diamond A_{s''_u}) \subseteq U(x_l, z'''_l, x_{l+1}, z'''_{l+1})$$

where $z'''_l$ and $z'''_{l+1}$ have the same form as $z'_l$ and $z'_{l+1}$, and where the weights $u'_{p+j}$ in $z'''_{l+1}$ are identical to those in $z'_{l+1}$ (i.e. defined by the relations $s_1, ..., s_p$). In other words, $z'_{l+1}$ and $z'''_{l+1}$ only differ in the coefficients $a''_2, ..., a''_m$. We can repeat this argument for every rule for which the relations in the body have the same combined weight. Hence we also obtain:

$$A_s \subseteq U(x_l, z^*_l, x_{l+1}, z^*_{l+1})$$

where $z^*_{l+1}$ differs only from $z'''_{l+1}$ in the coefficients $a''_2, ..., a''_m$. $\square$

**Lemma E.8.** *Let $s \in \mathcal{R}$ and assume that $(x_1, z) \in A_s$ for some $z \in \mathbb{R}$. There exist relations $s_2, ..., s_p \in \{r_1, ..., r_d\}$ such that $\mathcal{P} \models r_1 \circ s_2 \circ ... \circ s_p \sqsubseteq s$ and*

$$\max\{z \,|\, (x_1, z) \in A_s\} \leq x_{m+1} + \Big(\sum_{j=1}^{m} u_j \lambda^{3k+m-j} - \sum_{j=0}^{m-1} \lambda^j x^2_{m-j}\Big)\varepsilon + \sum_{j=2}^{m} a_j \varepsilon^j$$

*with $m = \omega(r_1) + \omega(s_2) + ... + \omega(s_p)$, $a_2, ..., a_m$ coefficients which do not depend on $\varepsilon$, and*

$$u_j = \begin{cases} 1 & \text{if } 0 < j \leq \omega(r_1) \text{ and } j \in I_{r_1} \\ 1 & \text{if } \omega(r_1) < j \leq \omega(r_1) + \omega(s_2) \text{ and } j \in I_{s_2} \\ ... \\ 1 & \text{if } \omega(r_1) + ... + \omega(s_{p-1}) < j \leq m \text{ and } j \in I_{s_p} \\ 0 & \text{otherwise} \end{cases}$$

*Proof.* We show the result by induction. If $s \in \mathcal{R}_0$ then we can only have $(x_1, z) \in A_s$ for some $z \in \mathbb{R}$ if $s = r_1$. We then have $A_s = X_{r_1}^{\uparrow \omega(r_1)}$. From Lemma E.4 we find:

$$\max\{z \,|\, (x_1, z) \in A_s\} \leq x_{1+\omega(r_1)} + \Big(\sum_{j=1}^{\omega(r_1)} u_j \lambda^{3k+\omega(r_1)-j} - \sum_{j=0}^{\omega(r_1)-1} \lambda^j x^2_{\omega(r_1)-j}\Big)\varepsilon + \sum_{j=2}^{\omega(r_1)} a'_j \varepsilon^j$$

Now suppose the result already holds for all relations in $\mathcal{R}_0 \cup ... \cup \mathcal{R}_{i-1}$ and suppose that $s \in \mathcal{R}_i$. For $(x_1, z) \in A_s$ to hold, there must be a rule $s_1 \circ ... \circ s_q \sqsubseteq s$ in $\mathcal{P} \cup \{s \sqsubseteq s\}$ such that $(x_1, z) \in A_{s_1} \diamond ... \diamond A_{s_q}$. Let $s_1 \circ ... \circ s_q \sqsubseteq s$ be the rule for which $\max\{z \,|\, (x_1, z) \in A_{s_1} \diamond ... \diamond A_{s_q}\}$ is maximal. Note that we then must have $(x_1, z) \in A_{s_1}$ for some $z \in \mathbb{R}$. By the induction hypothesis, we know that there must be some rule $s'_{11} \circ ... \circ s'_{1p_1} \sqsubseteq s_1$ such that

$$\max\{z \,|\, (x_1, z) \in A_{s_1}\} \leq x_{m_1+1} + \Big(\sum_{j=1}^{m_1} u_j \lambda^{3k+m_1-j} - \sum_{j=0}^{m_1-1} \lambda^j x^2_{m_1-j}\Big)\varepsilon + \sum_{j=2}^{m_1} a_j^{(1)} \varepsilon^j$$

with $m_1 = \omega(s'_{11}) + ... + \omega(s'_{1p_1})$, $a_2^{(1)}, ..., a_{m_1}^{(1)}$ coefficients which do not depend on $\varepsilon$, and

$$u_j = \begin{cases} 1 & \text{if } 0 < j \leq \omega(s'_{11}) \text{ and } j \in I_{s'_{11}} \\ 1 & \text{if } \omega(s'_{11}) < j \leq \omega(s'_{11}) + \omega(s'_{12}) \text{ and } j \in I_{s'_{12}} \\ ... & \\ 1 & \text{if } \omega(s'_{11}) + ... + \omega(s'_{1(p_1-1)}) < j \leq m_1 \text{ and } j \in I_{s'_{1p_1}} \\ 0 & \text{otherwise} \end{cases}$$

Furthermore, from Lemma E.7 we know that there are rules $s'_{f1} \circ ... \circ s'_{fp_f} \subseteq s_f$ for $f \in \{2, ..., q\}$ such that

$$A_{s_f} \subseteq U(x_{l_f}, z_{l_f}^{(f)}, x_{l_f+1}, z_{l_f+1}^{(f)})$$

where $l_f = m_1 + ... + m_{f-1}$, $m_f = \omega(s'_{f1}) + ... + \omega(s'_{fp_f})$ and

$$z_{l_f}^{(f)} = x_{l_f+m_f} + \left( \sum_{j=1}^{m_f} u'_{l_f+j-1} \lambda^{3k+m_f-j} - \sum_{j=0}^{m_f-1} \lambda^j x_{l_f+m_f-j-1}^2 \right) \varepsilon + \sum_{j=2}^{m_f} a_j^{(f)} \varepsilon^j$$

$$z_{l_f+1}^{(f)} = x_{l_f+m_f+1} + \left( \sum_{j=1}^{m_f} u_{l_f+j} \lambda^{3k+m_f-j} - \sum_{j=0}^{m_f-1} \lambda^j x_{l_f+m_f-j}^2 \right) \varepsilon + \sum_{j=2}^{m_f} b_j^{(f)} \varepsilon^j$$

with $a_j^{(f)}$ and $b_j^{(f)}$ constants which do not depend on $\varepsilon$, $u'_l, ..., u'_{l+m-1} \in \{0, 1\}$, and where the constants $u_{l_f+j}$ are defined as:

$$u'_{l_f+j} = \begin{cases} 1 & \text{if } 0 < j \leq \omega(s'_{f1}) \text{ and } l_f + j \in I_{s'_{f1}} \\ 1 & \text{if } \omega(s'_{f1}) < j \leq \omega(s'_{f1}) + \omega(s'_{f2}) \text{ and } l_f + j \in I_{s'_{f2}} \\ ... & \\ 1 & \text{if } \omega(s'_{f1}) + ... + \omega(s'_{f(p_f-1)}) < j \leq m_f \text{ and } l_f + j \in I_{s'_{fp}} \\ 0 & \text{otherwise} \end{cases} \tag{17}$$

$$= \begin{cases} 1 & \text{if } l_f < l_f + j \leq l_f + \omega(s'_{f1}) \text{ and } l_f + j \in I_{s'_{f1}} \\ 1 & \text{if } l_f + \omega(s'_{f1}) < l_f + j \leq l_f + \omega(s'_{f1}) + \omega(s'_{f2}) \text{ and } l_f + j \in I_{s'_{f2}} \\ ... & \\ 1 & \text{if } l_f + \omega(s'_{f1}) + ... + \omega(s'_{f(p_f-1)}) < l_f + j \leq l_{f+1} \text{ and } l_f + j \in I_{s'_{fp}} \\ 0 & \text{otherwise} \end{cases} \tag{18}$$

From Lemma C.11 we then find:

$$A_{s_2} \diamond ... \diamond A_{s_q} \subseteq U(x_{m_1}, z_{m_1}^*, x_{m_1+1}, z_{m_1+1}^*)$$

with

$$z_{m_1}^* = x_m + \left( \sum_{j=1}^{m-m_1} u'_{m_1+j-1} \lambda^{3k+m-m_1-j} - \sum_{j=0}^{m-m_1-1} \lambda^j x_{m-j-1}^2 \right) \varepsilon + \sum_{j=2}^{m-m_1} a_j^* \varepsilon^j$$

$$z_{m_1+1}^* = x_{m+1} + \left( \sum_{j=1}^{m-m_1} u_{m_1+j} \lambda^{3k+m-m_1-j} - \sum_{j=0}^{m-m_1-1} \lambda^j x_{m-j}^2 \right) \varepsilon + \sum_{j=2}^{m-m_1} b_j^* \varepsilon^j$$

where $m = m_1 + ... + m_q$, $a_j^{(f)}$ and $b_j^{(f)}$ are constants which do not depend on $\varepsilon$, and the constants $u'_{m_1+j-1}$ and $u_{m_1+j}$ are as defined before. From Lemma C.8, we then obtain:

$$\max\{z \mid (x_1, z) \in A_s\} = \max\{z \mid (x_1, z) \in A_{s_1} \diamond ... \diamond A_{s_q}\}$$

$$= \max\{z \mid (x_{m_1+1} + \Big(\sum_{j=1}^{m_1} u_j \lambda^{3k+m_1-j} - \sum_{j=0}^{m_1-1} \lambda^j x_{m_1-j}^2\Big)\varepsilon + \sum_{j=2}^{m_1} a_j^{(1)}\varepsilon^j, z) \in A_{s_2} \diamond ... \diamond A_{s_q}\}$$

$$= x_{m+1} + \Big(\Big(\sum_{j=1}^{m_1} u_j \lambda^{3k+m_1-j} - \sum_{j=0}^{m_1-1} \lambda^j x_{m_1-j}^2\Big)\lambda^{m-m_1}$$

$$+ \sum_{j=1}^{m-m_1} u_{m_1+j}\lambda^{3k+m-m_1-j} - \sum_{j=0}^{m-m_1-1} \lambda^j x_{m-j}^2\Big)\varepsilon + \sum_{j=2}^{m} c_j\varepsilon^j$$

$$= x_{m+1} + \Big(\Big(\sum_{j=1}^{m_1} u_j \lambda^{3k+m-j} - \sum_{j=0}^{m_1-1} \lambda^{j+m-m_1} x_{m_1-j}^2\Big)$$

$$+ \sum_{j=m_1+1}^{m} u_j\lambda^{3k+m-j} - \sum_{j=0}^{m-m_1-1} \lambda^j x_{m-j}^2\Big)\varepsilon + \sum_{j=2}^{m} c_j\varepsilon^j$$

$$= x_{m+1} + \Big(\sum_{j=1}^{m} u_j\lambda^{3k+m-j} - \sum_{j=0}^{m-1} \lambda^j x_{m-j}^2\Big)\varepsilon + \sum_{j=2}^{m} c_j\varepsilon^j$$

for some constants $c_2, ..., c_m$ which do not depend on $\varepsilon$. $\qquad\square$

**Lemma E.9.** *Suppose* $(x_1, z^*) \in A_r$, *where:*

$$z^* = x_{k+1} + \Big(\sum_{i=2}^{d} \lambda^{4k-z_i} - \sum_{j=0}^{k-1} \lambda^j x_{k-j}^2\Big)\varepsilon + \sum_{i=2}^{k} a_i'\varepsilon^i \qquad (19)$$

*Assuming $\varepsilon$ is sufficiently small, it follows that there must exist a rule $s_1 \circ ... \circ s_p \subseteq r$ entailed by $\mathcal{P}$ such that $(x_1, z^*) \in A_{s_1} \diamond ... \diamond A_{s_p}$ and $\omega(s_1) + ... + \omega(s_p) > k$.*

*Proof.* Suppose $(x_1, z^*) \in A_r$. From Lemma E.8, we know that there exist relations $s_2, ..., s_p \in \{r_1, ..., r_d\}$ such that $\mathcal{P} \models r_1 \circ s_2 \circ ... \circ s_p \subseteq s$ and

$$\max\{z \mid (x_1, z) \in A_s\} \leq x_{m+1} + \Big(\sum_{j=1}^{m} u_j\lambda^{3k+m-j} - \sum_{j=0}^{m-1} \lambda^j x_{m-j}^2\Big)\varepsilon + \sum_{j=2}^{m} a_j\varepsilon^j$$

with $u_j$ defined as in Lemma E.8. Since we assumed $(x_1, z^*) \in A_r$, there must exist such relations $s_2, ..., s_p$ for which:

$$z^* \leq x_{m+1} + \Big(\sum_{j=1}^{m} u_j\lambda^{3k+m-j} - \sum_{j=0}^{m-1} \lambda^j x_{m-j}^2\Big)\varepsilon + \sum_{j=2}^{m} a_j\varepsilon^j \qquad (20)$$

If $m = \omega(r_1) + \omega(s_2) + ...\omega(s_p) < k = \omega(r_1) + \omega(r_2) + ... + \omega(r_d)$, then this is impossible for sufficiently small $\varepsilon$, given that $x_{m+1} < x_{k+1}$.

Now assume $m = k$. Assuming $\varepsilon$ is sufficiently small, (20) can only hold if

$$\sum_{j=1}^{m} u_j\lambda^{3k+m-j} = \sum_{j=1}^{m} u_j\lambda^{4k-j} \geq \sum_{i=2}^{d} \lambda^{4k-z_i}$$

Note that $I_{r_1} \cup I_{s_2} \cup ... \cup I_{s_p} \subseteq \{z_2, ..., z_d\}$. Hence the latter condition is equivalent with:

$$\sum_{j=1}^{m} u_j\lambda^{4k-j} = \sum_{i=2}^{d} \lambda^{4k-z_i}$$

In other words, we need to show that $u_{z_2} = u_{z_3} = ... = u_{z_d} = 1$. However, $u_{z_2} = 1$ is only possible if $s_2 = r_2$. Given that $s_2 = r_2$, we similarly find that $u_{z_3} = 1$ is only possible if $s_r = r_3$. Continuing in this way, we find that $r_1 \circ r_2 \circ ... \circ r_d$

must be a prefix of $r_1 \circ s_2 \circ ... \circ s_p$. Since we assumed $m = k$ and all relations have a non-zero weight, it follows that $r_1 \circ r_2 \circ ... \circ r_d = r_1 \circ s_2 \circ ... \circ s_p$, which is a contradiction since $\mathcal{P} \not\models r_1 \circ r_2 \circ ... \circ r_d \subseteq r$ while we assumed $\mathcal{P} \models r_1 \circ s_2 \circ ... \circ s_p \subseteq r$. Hence the only possibility is that $m > k$. $\qquad\square$

**Lemma E.10.** *Let $s \in \mathcal{R}$ and suppose $s$ is balanced by $\omega$. It holds that*

$$A_s \subseteq X_*^{\uparrow \omega(s)}$$

*where $X_* = R_{I_*}$ and $I_* = \{z_2, ..., z_d\}$.*

*Proof.* Note that for every $i \in \{1, ..., d\}$ we have $X_{r_i} \subseteq X_* \subseteq X_*^{\uparrow \omega(s)}$, and we trivially also have $\{(x^*, x^*)\} \subseteq X_* \subseteq X_*^{\uparrow \omega(s)}$. We thus immediately find that the result is satisfied when $s \in \mathcal{R}_0$. Suppose the result already holds for every relation in $\mathcal{R}_0 \cup ... \cup \mathcal{R}_{i-1}$ and suppose $s \in \mathcal{R}_i$. For every rule $t_1 \circ ... \circ t_q \subseteq s$ in $\mathcal{P}$ we then have:

$$\begin{aligned}
A_{t_1} \diamond ... \diamond A_{t_q} &\subseteq X_*^{\uparrow \omega(t_1)} \diamond ... \diamond X_*^{\uparrow \omega(t_q)} \\
&= X_*^{\uparrow \omega(t_1) + ... + \omega(t_q)} \\
&= X_*^{\uparrow \omega(s)}
\end{aligned}$$

$\qquad\square$

Finally, we show the following counterpart to Lemma D.6.

**Lemma E.11.** *Suppose $\mathcal{P}$ contains the rule $s_1 \circ ... \circ s_p \subseteq r$. Assuming $\varepsilon$ is sufficiently small, we have that:*

$$(x_1, z^*) \notin A_{s_1} \diamond ... \diamond A_{s_p}$$

*Proof.* Note that by assumption, we have that $s_1, ... s_p$ are balanced by $\omega$. By Lemma E.10, we thus have that

$$A_{s_1} \diamond ... \diamond A_{s_p} \subseteq X_*^{\uparrow m}$$

with $m = \omega(s_1) + ... + \omega(s_p)$ and $X_*$ defined as in Lemma E.10. Using Lemmas C.12 and D.5, and the fact that $m - 1 \geq k$, we find

$$(x_1, z) \in X_*^{\uparrow m-1} \Rightarrow z > x_k + \frac{1}{2}\lambda^{k-1}$$

It follows that:

$$(x_1, z^*) \in A_s \Rightarrow \exists z > x_k + \frac{1}{2}\lambda^{k-1} . (z, z^*) \in X_*$$

However, for $z > x_k + \frac{1}{2}\lambda^{k-1}$, we have:

$$\begin{aligned}
(z, z') \in X_* &\Rightarrow (z, z') \in U(x_k, y_k, x^*, x^*) \\
&\Leftrightarrow z' - y_k \leq \frac{x^* - y_k}{x^* - x_k}(z - x_k) \\
&\Rightarrow z' \leq y_k + \frac{1}{2}\lambda^{k-1}\frac{x^* - y_k}{x^* - x_k} \\
&\Rightarrow z' < z^*
\end{aligned}$$

where the last step follow from the fact that $x^* - y_k < 0$ (and the assumption that $\varepsilon$ is sufficiently small). $\qquad\square$

Lemmas E.9 and E.11 together imply that $(x_1, z^*) \notin A_r$. Together with Lemma E.3, we can thus conclude that $A_{r_1} \diamond ... \diamond A_{r_k} \not\subseteq A_r$.

## F. Checking the Conditions from Proposition 5.4

We show how we can check, in polynomial time, whether the conditions from Proposition 5.4 are satisfied for a given rule base. Let $\mathcal{P}$ be an acyclic set of closed path rules and let $\mathcal{R}$ be the set of relations appearing in $\mathcal{P}$. Let us define the mapping $depth : \mathcal{R} \to \mathbb{N}$ as follows. If $r$ does not appear in the head of any rule, then $depth(r) = 0$. Otherwise, we define:

$$depth(r) = 1 + \max\{depth(s_i) \,|\, (s_1 \circ ... \circ s_k \subseteq r) \in \mathcal{P}, i \in \{1, ..., k\}\}$$

For each $r \in \mathcal{R}$, we can construct the set $D_r \subseteq \mathcal{R}$ of relations on which $r$ depends. Specifically, let us define $D_r^{(0)} = \{r\}$. For $i > 0$ we define:

$$D_r^{(i)} = D_r^{(i-1)} \cup \bigcup\{D_{r_1}^{(i-1)} \cup ... \cup D_{r_k}^{(i-1)} \,|\, (r_1 \circ ... \circ r_k \subseteq r) \in \mathcal{P}\}$$

Note that the sets $D_r^{(i)}$ can be computed in polynomial time. Indeed, for each $r \in \mathcal{R}$, the update process converges after at most $depth(r)$ iterations, where $depth(r) \leq |\mathcal{P}|$. Moreover, the time needed for computing the sets $D_r^{(i)}$ in each iteration is clearly polynomial in $|\mathcal{R}|$. Let us write $D_r$ for the sets that are obtained upon convergence.

To check whether a relation $r \in \mathcal{R}$ is regular in $\mathcal{P}$, we can use the following result.

**Lemma F.1.** *Let $r \in \mathcal{R}$. It holds that $r$ is regular in $\mathcal{P}$ iff for each rule $s_1 \circ ... \circ s_\ell \subseteq s$ from $\mathcal{P}$, with $s \in D_r$, we have that $D_{s_i} \cap D_{s_j} = \emptyset$ for $i \neq j$.*

*Proof.* Suppose $\mathcal{P}$ contains a rule $s_1 \circ ... \circ s_\ell \subseteq s$, with $s \in D_r$, such that $t \in D_{s_i} \cap D_{s_j}$ for some relation $t$, with $i \neq j$. Then $\mathcal{P}$ entails a rule with $s_i$ in the head where $t$ appears in the body, as well as a rule with $s_j$ in the head and $t$ in the body. Since $\mathcal{P}$ contains the rule $s_1 \circ ... \circ s_\ell \subseteq s$, this means that $\mathcal{P}$ entails a rule with $s$ in the head, where $t$ appears at least twice in the body. Furthermore, since $s \in D_r$, $\mathcal{P}$ must entail a rule with $s$ in the body and $r$ in the head, from which it follows that $\mathcal{P}$ entails a rule with $r$ in the head, where $t$ appears at least twice in the body. This means that $r$ cannot be regular. We have thus shown, by contraposition, that when $r$ is regular, it must be the case that for each rule $s_1 \circ ... \circ s_\ell \subseteq s$ from $\mathcal{P}$, with $s \in D_r$, we have that $D_{s_i} \cap D_{s_j} = \emptyset$ for $i \neq j$.

Now suppose that $r$ is not regular. Because $\mathcal{P}$ is acyclic, we already know that no rules of the form $r_1 \circ ... \circ r \circ ... \circ r_k \subseteq r$ can be entailed from $\mathcal{P}$. If $r$ is not regular, it must therefore be the case that $\mathcal{P}$ entails a rule of the form $r_1 \circ ... \circ s \circ ... \circ s \circ ... \circ r_k \subseteq r$. Let $t_1 \circ ... \circ t_\ell \subseteq r$ be the rule from $\mathcal{P}$ which was used to derive the latter rule. Then either it must be the case that $s \in D_{t_i} \cap D_{t_j}$ for some $i \neq j$, or there must be some $i \in \{1, ..., \ell\}$ such that $\mathcal{P}$ entails a rule of the form $t_1' \circ ... \circ s \circ ... \circ s \circ ... \circ t_{\ell'}' \subseteq t_i$. In the latter case, we can simply continue the same argument until we end up with a rule $t_1'' \circ ... \circ t_{\ell''}'' \subseteq t''$ for which $D_{t_i''} \cap D_{t_j''} \neq \emptyset$ for some $i \neq j$ (with $t'' \in D_r$). In particular, when $r$ is not regular, there always exists some rule $s_1 \circ ... \circ s_\ell \subseteq s$ from $\mathcal{P}$, with $s \in D_r$, such that $D_{s_i} \cap D_{s_j} \neq \emptyset$ for some $i \neq j$. $\qquad\square$

To check whether the second condition from Proposition 5.4 is satisfied for a relation $r \in \mathcal{R}$, we construct a system of linear constraints $E_r$ as follows. We associate with each relation $s$ a variable $w_s$. For each relation $s \in D_r \setminus \{r\}$ and each rule $s_1 \circ ... \circ s_k \subseteq s$ in $\mathcal{P}$ we add the equation $w_{s_1} + ... + w_{s_k} = w_s$ to $E_r$. Furthermore, for every relation $s \in \mathcal{R}$, we add the inequality $w_s \geq 1$ to $E_r$. Note that the feasibility of $E_r$ can be checked in polynomial time using linear programming. We have the following result.

**Lemma F.2.** *The second condition from Proposition 5.4 is satisfied for $r$ iff the set of constraints $E_r$ is feasible.*

*Proof.* Suppose $E_r$ is feasible. Since all coefficients appearing in $E_r$ are integers, $E_r$ must have a rational solution. Let us write $w_s^*$ for the value of $w_s$ in such a rational solution. There must be some integer $c \geq 1$ such that $cw_s^*$ is integer for every $s \in \mathcal{R}$. Clearly, $w_s' = cw_s^*$ then defines an integer solution of $E_r$. Let us define $\omega_r : \mathcal{R} \to \mathbb{N} \setminus \{0\}$ as $\omega_r(s) = w_s'$.

Let $r_1 \circ ... \circ r_k \subseteq r$ be a rule entailed by $\mathcal{P}$. We show using induction that $\omega_r$ balances $s$ for every $s \in D_{r_i}$ and $i \in \{1, ..., k\}$. It then follows that $\omega_r$ balances $r_1, ..., r_k$, meaning that the second condition from Proposition 5.4 is satisfied.

Let $s \in D_{r_i}$. If $depth(s) = 0$ then we trivially have that $s$ is balanced. Now assume $depth(s) > 0$. If $\mathcal{P}$ contains the rule $s_1 \circ ... \circ s_\ell \subseteq s$, then we have $\omega_r(s_1) + ... + \omega_r(s_\ell) = \omega_r(s)$ by construction of $\omega_r$, given that $E_r$ contains the equation $w_{s_1} + ... + w_{s_\ell} = w_s$ (noting that $s \in D_{r_i}$ implies $s \in D_r \setminus \{r\}$). Now assume there is a rule $s_1 \circ ... \circ s_\ell \subseteq s$ which is entailed by $\mathcal{P}$, but not itself contained in $\mathcal{P}$. Then there must be some rule $t_1 \circ ... \circ t_m \subseteq s$ in $\mathcal{P}$, such that $\mathcal{P}$ entails $s_1 \circ ... \circ s_{i_1} \subseteq t_1, ...., s_{i_{m-1}+1} \circ ... \circ s_\ell \subseteq t_m$. Noting that $depth(t_1), ..., depth(t_m) < depth(s)$, by the induction hypothesis

we have $\omega_r(s_1) + ... + \omega_r(s_{i_1}) = \omega_r(t_1), ...., \omega_r(s_{i_{m-1}+1}) + ... + \omega_r(s_\ell) = \omega_r(t_m)$, while by construction of $\omega_r$ we have $\omega_r(t_1) + ... + \omega_r(t_m) = \omega_r(s)$. It follows that $\omega_r(s_1) + ... + \omega_r(s_\ell) = \omega_r(s)$.

Finally, we consider the converse direction. Suppose the second condition from Proposition 5.4 is satisfied for $r$ and some weighting $\omega_r : \mathcal{R} \to \mathbb{N} \setminus \{0\}$. Then clearly $w_s = \omega_r(s)$ defines a solution of $E_r$, meaning that $E_r$ is feasible. $\square$

# G. Proof of Proposition 6.1

Let $(\eta, \eta')$ be a relation embedding satisfying the three given rules. Let $v$ be an entity embedding such that $(v, \eta)$ captures the given relational facts. Let $v_i(e)$ denote the $i^{\text{th}}$ coordinate of the vector $v(e)$, for an entity $e$. Suppose $(v(a), v(c)) \notin \eta'(t)$. Then there must be some coordinate $i$ such that $(v_i(a), v_i(c)) \notin \eta'_i(t)$. Let us fix such a coordinate $i$. For the ease of presentation, let us write $v_i(a) = x$, $v_i(b_{jk}) = y_{jk}$ and $v_i(c) = z$, and for any relation $r$, $\eta_i(r) = A_r$ and $\eta'_i(r) = A'_r$. Since $(v, \eta)$ captures the given relational facts, we thus know that:

$$\begin{array}{llll} (x, y_{12}) \in A_{r_1} & (y_{12}, z) \in A_{s_2} & (x, y_{13}) \in A_{r_1} & (y_{13}, z) \in A_{s_3} \\ (x, y_{21}) \in A_{r_2} & (y_{21}, z) \in A_{s_1} & (x, y_{23}) \in A_{r_2} & (y_{23}, z) \in A_{s_3} \\ (x, y_{31}) \in A_{r_3} & (y_{31}, z) \in A_{s_1} & (x, y_{32}) \in A_{r_3} & (y_{32}, z) \in A_{s_2} \end{array}$$

Note that we cannot have that $y_{12}$ is between $y_{21}$ and $y_{31}$, since the convexity of $A_{s_1}$ would then imply $y_{12} \in A_{s_1}$ and thus we would obtain $(x, z) \in A_{r_1} \diamond A_{s_1}$, which is not possible given that the rule $r_1 \circ s_1 \subseteq t$ is captured by $(\eta, \eta')$ and $(x, z) \notin A'_t$. Using the same argument, we can also derive the following constraints:

1. $y_{12}$ is not between $y_{21}$ and $y_{31}$

2. $y_{13}$ is not between $y_{21}$ and $y_{31}$

3. $y_{21}$ is not between $y_{12}$ and $y_{32}$

4. $y_{23}$ is not between $y_{12}$ and $y_{32}$

5. $y_{31}$ is not between $y_{13}$ and $y_{23}$

6. $y_{32}$ is not between $y_{13}$ and $y_{23}$

Similarly, we also cannot have that $y_{12}$ is between $y_{21}$ and $y_{23}$ since otherwise, the convexity of $A_{r_2}$ would imply $y_{12} \in A_{r_2}$ and thus $(x, z) \in A_{r_2} \diamond A_{s_2} \subseteq A'_t$. This gives us:

7. $y_{12}$ is not between $y_{21}$ and $y_{23}$

8. $y_{13}$ is not between $y_{31}$ and $y_{32}$

9. $y_{21}$ is not between $y_{12}$ and $y_{13}$

10. $y_{23}$ is not between $y_{31}$ and $y_{32}$

11. $y_{31}$ is not between $y_{12}$ and $y_{13}$

12. $y_{32}$ is not between $y_{21}$ and $y_{23}$

W.l.o.g. we can assume that $y_{21} \leq y_{31}$ (since the case where $y_{31} \leq y_{21}$ immediately follows by symmetry). Because of condition 1, we must then have $y_{12} < y_{21}$ or $y_{31} < y_{12}$. Let us first consider the case where $y_{12} < y_{21}$. We then have

13. $y_{12} < y_{21} \leq y_{31}$

We infer:

14. From 13 and 7 we obtain $y_{12} < y_{23}$.

15. From 13 and 11 we obtain $y_{13} < y_{31}$.

16. From 14 and 4 we obtain $y_{32} < y_{23}$.

17. From 15 and 8 we obtain $y_{13} < y_{32}$.

18. From 17 and 16 we obtain $y_{13} < y_{32} < y_{23}$, which is in contradiction with 6

Let us now consider the case where $y_{31} < y_{12}$. We then have:

19. $y_{21} \le y_{31} < y_{12}$

Entirely analogously as in the previous case, we infer:

20. From 19 and 7 we obtain $y_{23} < y_{12}$

21. From 19 and 11 we obtain $y_{31} < y_{13}$

22. From 20 and 4 we obtain $y_{23} < y_{32}$

23. From 21 and 8 we obtain $y_{32} < y_{13}$

24. From 22 and 23 we obtain $y_{23} < y_{32} < y_{13}$, which is in contradiction with 6.

## H. Proof of Proposition 6.2

The result follows from the following observation.

**Lemma H.1.** *Let $A \subseteq \mathbb{R}^2$ be such that $A = inv(A)$. It holds that:*

$$(A = inv(A)) \Rightarrow (A \subseteq A \diamond A \diamond A)$$

*Proof.* Suppose $(x, y) \in A$. Since $A = inv(A)$ this implies $(y, x) \in A$ and thus $(x, x) \in A \diamond A$ and $(x, y) \in A \diamond A \diamond A$. $\square$

If $(\eta, \eta')$ captures $r^{\uparrow i} \subseteq s$, for $i \geq 3$, then we have for each coordinate $j$:

$$\eta_j(r)^{\uparrow i} \subseteq \eta'_j(s)$$

where $\uparrow$ is defined for regions as in Section E.2, i.e. slightly abusing notation, we have $\eta_j(r^{\uparrow i}) = \eta_j(r)^{\uparrow i}$. If $r$ is symmetric in $\eta$, then it follows from Lemma H.1 that $\eta_j(r)^{\uparrow(i-2)} \subseteq \eta_j(r)^{\uparrow i}$, and thus:

$$\eta_j(r)^{\uparrow(i-2)} \subseteq \eta'_j(s)$$

## I. Experimental Setup

All models were implemented using PyKEEN[2]. They were trained against a binary cross-entropy loss with self-adversarial negative sampling, as defined by Sun et al. (2019) in a 1-vs-all setting: every entity $f$ such that $r(e, f) \in \mathcal{I}_{\mathcal{P}+}$ is scored against every other entity $f' \in \mathcal{E} \setminus \{f\}$. Optimization is done with Adam at a learning rate of 0.1. The graph $\mathcal{I}_{\mathcal{P}+}$ is split in two equal components for validation: half of the graph is used for early stopping on the mean reciprocal rank metric (evaluated every 5 epochs, with a patience of 50 epochs); the other half is used for evaluation on the hits@1 metric.

The polygon model can be seen as a generalization of ExpressivE. Both models rely on the following function that scales and translates an input vector ($\mathbf{x}, \mathbf{s}, \mathbf{u} \in \mathbb{R}^n$):

$$f_{\mathbf{s},\mathbf{u}}(\mathbf{x}) = \mathbf{s} \odot \mathbf{x} + \mathbf{u}$$

where we write $\odot$ for the component-wise product. The scoring function of ExpressivE, in its simplest form, is defined as:

$$s_r(\mathbf{e}, \mathbf{f}) = -\|f_{\mathbf{s},\mathbf{u}}(\mathbf{e} \oplus \mathbf{f}) - \mathbf{f} \oplus \mathbf{e}\|$$

---

[2]https://pykeen.readthedocs.io/en/stable/

capturing regions $\eta(r)$ parameterized by $\mathbf{s}, \mathbf{u} \in \mathbb{R}^{2n}$ and defined as follows:

$$\eta_i(r) = \{(x, y) \mid |s_i x + u_i - y| \leq \theta_i\} \cap \{(x, y) \mid |s_{n+i} y + u_{n+i} - x| \leq \theta_{n+i}\}$$

The definition we give here for $s_r(e, f)$ corresponds to what Pavlovic & Sallinger (2023) refer to as *functional* ExpressivE. In its functional form, width parameters $\theta_i$ and $\theta_{n+i}$ are not trainable. The more extensive formulation of ExpressivE makes them trainable but at a theoretical level, the two formulations are equivalent. In our experiments, we use functional ExpressivE.

ExpressivE regions always have parallel bounds, which is not true of arbitrary polygons. We generalize their definition and introduce four types of base regions:

$$L_y(s, u) = \{(x, y) \mid sx + u - y \leq 0\}$$
$$L_x(s, u) = \{(x, y) \mid sy + u - x \leq 0\}$$
$$U_y(s, u) = \{(x, y) \mid sx + u - y \geq 0\}$$
$$U_x(s, u) = \{(x, y) \mid sy + u - x \geq 0\}$$

A polygon is the intersection of any combination of these base regions. In particular, ExpressivE regions can then be rewritten as:

$$\eta_i(r) = L_y(s_i, u_i - \theta_i) \cap U_y(s_i, u_i + \theta_i) \cap L_x(s_{n+i}, u_{n+i} - \theta_{n+i}) \cap U_x(s_{n+i}, u_{n+i} + \theta_{n+i})$$

In our experiments, polygons with 4 and 8 edges are defined by 1 and 2 regions of each type, respectively, and polygons with 6 and 10 edges have, in addition, regions of type $L_y$ and $U_y$. Polygons with 2 edges have regions of type $U_x$ and $U_y$.

To train embeddings for a polygon with $k$ edges, we use the following scoring function:

$$s_r(\mathbf{e}, \mathbf{f}) = -\|\text{ReLU}(f_{\mathbf{s},\mathbf{u}}(\mathbf{x}) - \mathbf{y})\|$$

where $\mathbf{s}, \mathbf{u} \in \mathbb{R}^{kn}$ and $\mathbf{x}, \mathbf{y}$ depend on base region types. For instance, for polygons with 4 edges, we set $\mathbf{x} = \mathbf{e} \oplus \mathbf{f} \oplus \mathbf{e} \oplus \mathbf{f}$ and $\mathbf{y} = \mathbf{f} \oplus \mathbf{e} \oplus -\mathbf{f} \oplus -\mathbf{e}$, encoding regions of type $L_y(s, u)$, $L_x(s, u)$, $U_y(-s, -u)$, and $U_x(-s, -u)$. This formulation ensures that whenever a constraint is satisfied at coordinate $i \leq kn$, it does not affect the score, i.e. $\text{ReLU}(f_{s_i, u_i}(x_i) - y_i) = 0$.

## J. Impact of Dimensionality

Increasing dimensionality does not consistently improve results. In Table 3, we show the difference in hits@1 between models with $d = 10$ and models with $d = 50$ (all other hyperparameters left unchanged). ExpressivE is the only model for which the difference is positive across all datasets, after averaging over 20 runs. When looking at maximum hits@1, however, higher-dimensional ExpressivE embeddings perform worse on five datasets.

*Table 3.* Comparison between results for higher-dimensional embeddings ($d = 50$) and those from Table 2 ($d = 10$), reported as the difference in hits@1 (positive if higher-dimensional embeddings perform better). Between parentheses, we report whether the minimum and maximum values are higher ($\uparrow$) or lower ($\downarrow$) for $d = 50$.

|  | PERM2 | PERM3 | MIX2 | MIX3 | REP12 | REP13 | COMB |
|---|---|---|---|---|---|---|---|
| Octagons | 0.00 ($\downarrow - \downarrow$) | 0.20 ($\uparrow - \downarrow$) | -0.19 ($\downarrow - \downarrow$) | -0.02 ($\downarrow - \downarrow$) | -0.03 ($\downarrow - \downarrow$) | 0.03 ($\uparrow - \downarrow$) | -0.12 ($\downarrow - \downarrow$) |
| ExpressivE | 0.01 ($\uparrow - \downarrow$) | 0.03 ($\uparrow - \downarrow$) | 0.05 ($\uparrow - \downarrow$) | 0.11 ($\uparrow - \downarrow$) | 0.09 ($\downarrow - \uparrow$) | 0.17 ($\downarrow - \downarrow$) | 0.12 ($\uparrow - \uparrow$) |
| Polygons (2) | 0.04 ($\uparrow - \downarrow$) | -0.02 ($\downarrow - \downarrow$) | 0.03 ($\uparrow - \downarrow$) | -0.04 ($\downarrow - \downarrow$) | -0.03 ($\downarrow - \downarrow$) | -0.02 ($\downarrow - \downarrow$) | 0.06 ($\uparrow - \downarrow$) |
| Polygons (4) | 0.01 ($\uparrow - \downarrow$) | 0.12 ($\downarrow - \downarrow$) | 0.13 ($\uparrow - \downarrow$) | 0.03 ($\uparrow - \uparrow$) | 0.07 ($\uparrow - \uparrow$) | -0.06 ($\downarrow - \downarrow$) | -0.01 ($\downarrow - \uparrow$) |
| Polygons (6) | 0.00 ($\downarrow - \downarrow$) | 0.10 ($\uparrow - \downarrow$) | 0.05 ($\uparrow - \downarrow$) | -0.02 ($\downarrow - \downarrow$) | 0.17 ($\uparrow - \uparrow$) | 0.01 ($\uparrow - \downarrow$) | 0.00 ($\uparrow - \downarrow$) |
| Polygons (8) | 0.03 ($\uparrow - \downarrow$) | 0.14 ($\uparrow - \downarrow$) | 0.13 ($\uparrow - \uparrow$) | 0.01 ($\downarrow - \uparrow$) | -0.02 ($\uparrow - \downarrow$) | -0.03 ($\downarrow - \downarrow$) | -0.01 ($\uparrow - \downarrow$) |
| Polygons (10) | 0.00 ($\downarrow - \downarrow$) | -0.02 ($\uparrow - \downarrow$) | 0.07 ($\uparrow - \uparrow$) | 0.01 ($\uparrow - \downarrow$) | 0.09 ($\uparrow - \uparrow$) | 0.04 ($\downarrow - \uparrow$) | -0.08 ($\uparrow - \downarrow$) |

