# OpenReview forum: "Faithful Relational Reasoning with Region-based Embeddings: Expressivity of Convex Coordinate-wise Models"
_ICML.cc/2026/Conference — ICML 2026 regular_

### Official Review · Reviewer_WNA9 · 2026-03-11

**Soundness:** 3
**Presentation:** 3
**Significance:** 3
**Originality:** 3
**Overall Recommendation:** 4
**Confidence:** 3

**Summary:**

The authors analyze expressivity in convex coordinate wise knowledge graph embeddings. Non convex models capture rules but lack practicality while convexity forces spurious dependencies. They prove specific conditions allowing polygon embeddings to handle acyclic rules without unwanted entailments. The novel FAIRE benchmark demonstrates existing models fail these theoretical guarantees empirically.

**Compliance With Llm Reviewing Policy:**

Affirmed.

**Key Questions For Authors:**

Could you elaborate on why empirical training runs exhibited extreme variance and whether you attempted different initialization strategies to stabilize the polygon model.

Please explain how strict balancing conditions required for accordant embeddings might be relaxed or approximated for large scale knowledge graphs where perfect regularity is impossible.

What specific inductive biases do you suspect are necessary to bridge the gap between theoretical expressivity results and actual learnability via gradient descent.

**Limitations:**

The authors acknowledge their models fail empirically despite theoretical guarantees. They must discuss computational overhead when scaling non accordant frameworks to massive graphs. Addressing societal impacts of unpredictable neurosymbolic systems would strengthen the paper.

**Strengths And Weaknesses:**

Theoretical proofs bounding model expressivity are mathematically rigorous. However high training variance obscures whether empirical failures stem from fundamental limitations or poor optimization. Using standard binary cross entropy loss is insufficient for training complex polygons making failure claims premature.

The narrative transitions smoothly from concepts to complex proofs. Conversely the mathematical parameterized polygon construction is exceedingly dense. It lacks adequate visual aids to help readers intuitively grasp how these shapes avoid earlier convexity limitations.

Addressing expressivity gaps tackles relevant symbolic reasoning bottlenecks. Practical impact is blunted because strict balancing conditions are restrictive and rarely applicable to noisy datasets violating acyclic assumptions. Since theoretically capable models fail practically the real world utility of these guarantees remains questionable.

Introducing the FAIRE benchmark to test spurious rule capture is creative. Yet core techniques build heavily on recent prior work regarding regular rule bases and octagon embeddings diminishing overall novelty.

---

> ### Author Rebuttal · Authors · 2026-03-30
>
> We thank the reviewer for their careful reading of the paper and constructive feedback.
>
> > Q1 Could you elaborate on why empirical training runs exhibited extreme variance and whether you attempted different initialization strategies to stabilize the polygon model.
>
> We have experimented with a number of strategies for improving stability, including changing the number of dimensions, changing the negative sampling strategy, and relaxing our early stopping criterion. These changes have led to improved results, but have not addressed the instability of the process (see response to Reviewer gikF, Q3). We believe the key to better stability is to regularize the polygons, for instance to encourage them to be more similar to the ones that are used in the proof (see response to Q3). Preliminary experiments with different initialization strategies have not yielded meaningful improvements, but we believe that these may indeed also have a role to play.
>
> > Q2 Please explain how strict balancing conditions required for accordant embeddings might be relaxed or approximated for large scale knowledge graphs where perfect regularity is impossible.
>
> In the standard link prediction setting, the ideal rule base that we want the model to learn would typically have cyclical dependencies, and would thus not satisfy the conditions of Proposition 5.4. More generally, given the theoretical limitations we have identified in the paper, it is unlikely that convex coordinate-wise embedding models can achieve perfect reasoning in such settings. In practice, however, strong empirical results may still be possible by learning an approximation of this ideal rule base, capturing the most important rules while satisfying the conditions of Proposition 5.4. Indeed, convex coordinate-wise models such as ExpressivE perform well on many real-world KGs, despite being even more restricted in terms of theoretical expressivity. When GNN-based encoders are used for learning entity embeddings (as in ReshufflE), another possible strategy would be to learn embeddings that are “approximately accordant” in some sense (e.g. where the embeddings used by the encoder and decoder are similar but not always identical). Finally, we also want to highlight that our results may have practical significance outside the traditional link prediction setting, e.g. where we use entity embeddings for reasoning within neuro-symbolic architectures for planning (Stahlberg et al., 2025) or abstract visual reasoning (Webb et al., 2024), which is something we would like to explore in future work.
>
> > Q3 What specific inductive biases do you suspect are necessary to bridge the gap between theoretical expressivity results and actual learnability via gradient descent.
>
> In ExpressivE, parallelograms are modelled using piece-wise linear scoring functions. The parameterisation of these piece-wise linear functions is tied to the distance between the two corresponding parallel lines (a trick which was borrowed from BoxE). This provides an important inductive bias, which encourages regions to be compact (and makes the boundaries “fuzzy” when suitable compact regions cannot be found). Since our polygons are not defined in terms of pairs of parallel lines, we cannot use the same approach.
>
> Another point that can be seen from the experiments is that increasing the number of edges does not reliably improve the results. This confirms that suitable strategies for mitigating overfitting may be needed. Essentially we would like to have a loss function which encourages the polygons to be as simple as possible. One possibility (which aligns with the constructions from the proof) would be to encourage most of the bounding lines to be increasing, and to encourage the slopes of all these increasing lines to be similar.
>
> As the focus of this paper is on the theoretical expressivity results, rather than introducing a new KG embedding model, a more extensive study of how polygons can be learned was left for future work.

---

> > ### Author Rebuttal · Reviewer_WNA9 · 2026-04-03
> >
> > Thanks for responses.

---

### Official Review · Reviewer_pmrf · 2026-03-11

**Soundness:** 4
**Presentation:** 3
**Significance:** 4
**Originality:** 3
**Overall Recommendation:** 6
**Confidence:** 3

**Summary:**

The paper characterizes embedding models as region-based embeddings, therefore focussing on the existence or absence of a finite set of relations, and prove under which conditions a model is theoretically expressive enough to capture sequences of relations, with conditions on the nature of the sequence of rules and the geometry of the relational embedding space.

**Compliance With Llm Reviewing Policy:**

Affirmed.

**Key Questions For Authors:**

1. lines 242-245, column 1: the authors state that standard strategies for learning KG embeddings can only be used for learning accordant rules. While this statement is consistent with experimental observations later in the article, but why are scoring function-based losses unable to learn non-accordant rules?

**Limitations:**

yes

**Strengths And Weaknesses:**

## Soundness
All theoretical claims are properly documented and the proofs are correct, to the best of my knowledge.

The experiment at the end of the paper is well deigned, grounded in the theoretical framework proposed by the authors, and allow to properly judge the gap between theoretical expressivity and what is achieved by state of the art training strategies.


## Presentation
The narrative is relatively easy to follow, with a logical flow and overall sufficient illustrations and examples.

Some proofs may benefit from a bit of clarification, in particular the conclusion driven from Proposition 4.2 (l201-203 column 2). The corresponding thought process is clarified in the last paragraph of the page (page 4), even if it is not put in relation with l201-203 column 2.

The paper position itself clearly and its contributions are easy to identify and relate to previous results from the literature.

## Significance
The definition of the expressivity of models is a hard but crucial problem to tackle to be able to judge whether the inability of a model to perform well on a task is intrinsic to the model (insufficient expressivity) among other potential applications.
The paper propose both a theoretical contribution of high impact potential, and raises an issue that should be tackled in further work by the community.
In terms of impact, the authors provide key insight on the capabilities of cost-efficient approaches based on relational embeddings. Given the widespread use of graph-based models and relation-based applications, for instance through RAG or other state of the art approaches, better understanding of model capabilities can help choose more appropriate models depending on the nature of relation chains tho manipulate, and potentially re-kindle interest in simpler models.

## Originality
To the best of my knowledge, the theoretical contributions claimed by the authors are novel and complement previous results by providing a more general framework. The perspectives and observations provided are a significant advancement in identifying the current bottleneck in relational embedding models.

---

> ### Author Rebuttal · Authors · 2026-03-30
>
> We thank the reviewer for their careful reading of the paper and constructive feedback.
>
> > Some proofs may benefit from a bit of clarification, in particular the conclusion driven from Proposition 4.2 (l201-203 column 2). The corresponding thought process is clarified in the last paragraph of the page (page 4), even if it is not put in relation with l201-203 column 2.
>
> Thanks for the suggestion. We will clarify this point.
>
> > Q1 lines 242-245, column 1: the authors state that standard strategies for learning KG embeddings can only be used for learning accordant rules. While this statement is consistent with experimental observations later in the article, but why are scoring function-based losses unable to learn non-accordant rules?
>
> Standard strategies for learning KG embeddings only learn a single representation for each relation. When we learn a region-based model in this way, it must therefore be accordant.

---

> > ### Author Rebuttal · Reviewer_pmrf · 2026-04-03
> >
> > I thank the authors for their answer to my question, and maintain my initial score.

---

### Official Review · Reviewer_TJR1 · 2026-03-12

**Soundness:** 2
**Presentation:** 3
**Significance:** 2
**Originality:** 3
**Overall Recommendation:** 3
**Confidence:** 4

**Summary:**

The paper studies the expressivity of convex coordinate-wise region-based embeddings for relational reasoning over knowledge graphs. The central question is whether such models can faithfully capture arbitrary sets of closed path rules — meaning they capture exactly the rules entailed by a given rule base, no more and no less. The main contributions are a series of theoretical propositions characterizing what these models can and cannot do, plus a synthetic benchmark (FAIRE) for empirical evaluation.

**Compliance With Llm Reviewing Policy:**

Affirmed.

**Key Questions For Authors:**

1) The polygon model is provably expressive enough to achieve perfect results on all FAIRE datasets, yet empirically it performs worst among all models. Can you provide any analysis of why training fails to find the theoretically guaranteed configurations? Do the learned regions bear any resemblance to the constructions in the proofs?

2) Proposition 5.4 provides sufficient conditions for faithful rule capture, and Proposition 4.2 shows these conditions cannot always be met. What is the precise boundary between the two? Can you characterize which acyclic rule bases fall outside the scope of Proposition 5.4, and how common are such cases in practice?

3) Proposition 6.1 demonstrates that spurious semantic dependencies can be unavoidable under specific conditions. Do these conditions arise naturally in real knowledge graphs, or are they pathological? Is the result a special case of a more general impossibility theorem?

4) The variance across runs is extremely high throughout Table 2. Have you investigated whether this reflects sensitivity to initialization, instability in the optimization landscape, or something specific to the FAIRE benchmark design? Does averaging across such unstable runs produce meaningful conclusions?

**Limitations:**

No.
The paper's impact statement addresses societal consequences but says nothing meaningful about technical limitations. The authors should explicitly acknowledge: (1) the unexplained gap between theoretical expressivity and empirical performance, which undermines the practical relevance of the positive results; (2) the restrictiveness of the sufficient conditions in Proposition 5.4 and how many practical rule bases fall outside their scope; (3) the purely synthetic nature of FAIRE and its uncertain relevance to real-world knowledge graphs; and (4) the extreme instability across runs as a limitation of both the models and the experimental methodology.

**Strengths And Weaknesses:**

**Strengths**:
1) The theoretical framework is well-structured, and the progression from negative results (Section 4) to positive results (Section 5) is logical

2) The distinction between accordant and non-accordant embeddings is clearly motivated and practically meaningful

3) The FAIRE benchmark addresses a real gap, existing benchmarks are ill-suited for evaluating faithful reasoning

4) The proofs, while highly technical, appear carefully constructed'

**Weaknesses**

1) The polygon model provably should work, but empirically doesn't , and the paper has no explanation.
This is the single most damaging issue in the paper. The authors prove (Proposition 5.4) that the polygon model is expressively sufficient to achieve perfect hits@1 on all FAIRE datasets under the right conditions. Yet Table 2 shows the polygon model is often the worst performing model, sometimes barely above random chance (e.g., 0.16 on COMB). The paper's only response to this striking contradiction is one sentence: "theoretical constructions sometimes rely on configurations that may be difficult to learn in practice." This is deeply unsatisfying for two reasons. First, it undermines the practical relevance of essentially all the positive theoretical results in the paper.  if the theoretically expressive model cannot be trained to exploit its expressivity, what is the value of proving it is expressive? Second, it raises the question of whether the FAIRE benchmark is actually evaluating what the paper claims it evaluates. Without understanding why training fails, it is impossible to know whether the failure reflects a fundamental limitation of the model class, an optimization problem, or simply a poor experimental setup.

2) Proposition 4.2 shows certain rule bases cannot be faithfully captured by accordant convex coordinate-wise models, while Proposition 5.4 shows certain rule bases can under special conditions. The paper never characterizes which rule bases fall into which category. This means the reader cannot answer the most natural practical question the paper raises: for a given rule base, can a coordinate-wise model handle it faithfully or not? Without this characterization, the theoretical contributions are significantly less actionable than they appear.

3) Proposition 6.1 shows that convex coordinate-wise models may be forced to capture spurious semantic dependencies. However, the paper never establishes whether the specific conditions triggering this problem arise commonly in practice or represent a pathological edge case. If common, this is a severe and broadly important limitation. If rare, it is a theoretical curiosity. The paper presents it as the former without justification, and does not investigate whether it is actually a special case of a more general impossibility result.

4) The polygon model is theoretically proven to be expressive enough to achieve perfect results on all FAIRE datasets, yet empirically it is often the worst performing model. The paper offers no investigation into why — no loss landscape analysis, no inspection of learned regions, no ablations over initialization or learning rate. This is the most important empirical finding in the paper and it is left completely undiagnosed.

5) Across 5 runs, many models show hits@1 ranges spanning nearly the entire possible scale (e.g., 0.08 to 0.90 for Polygons on PERM3). This level of instability fundamentally undermines the reliability of the conclusions drawn from Table 2, yet the paper simply reports it without any attempt to understand or address it. Five runs is almost certainly insufficient under such variance

6) The paper does not systematically map each FAIRE dataset to the relevant theoretical propositions. For example, it is not clearly stated which datasets fall within the scope of Proposition 5.4 and which do not. Without this mapping, the experiments cannot serve their stated purpose of empirically validating the theoretical findings.

---

> ### Author Rebuttal · Authors · 2026-03-30
>
> We thank the reviewer for their careful reading of the paper and constructive feedback.
>
> > The paper does not systematically map each FAIRE dataset to the relevant theoretical propositions. For example, it is not clearly stated which datasets fall within the scope of Proposition 5.4 and which do not.
>
> In each of the datasets, the relations which appear in the body of a rule do not appear as the head of another rule. This means that the relations which appear in the body of a rule are trivially balanced. In particular, all rule bases thus satisfy the conditions of Proposition 5.4. In addition, four of these rule bases only involve regular relations, which means that they can also be captured using octagons. This is explained in the second paragraph of Section 7.
>
> > Q1 Can you provide any analysis of why training fails to find the theoretically guaranteed configurations? Do the learned regions bear any resemblance to the constructions in the proofs?
>
> Learning regions appears to be inherently more difficult than learning simpler geometric objects (e.g. vector translations). Successful models such as ExpressivE rely on a carefully constructed parameterisation, which imposes a strong inductive bias (mostly to reduce the width of the regions, which makes them behave more like traditional embeddings); see also our response to Reviewer WNA9. However, note that in the new experiments (see response to Reviewer gikF), there is always a polygon configuration which achieves strong results (i.e. where the max hits@1 is high). The main challenge is thus to improve the stability of the learning process, which we have left as an important challenge for future work. The aim of our analysis was mostly to highlight that besides the theoretical considerations that we focus on in the paper, there are also important unsolved empirical questions. The learned polygons indeed look quite different from the ones that are used in the proofs, which gives a hint as to how an effective regularization strategy might look.
>
> > Q2 Proposition 5.4 provides sufficient conditions for faithful rule capture, and Proposition 4.2 shows these conditions cannot always be met. What is the precise boundary between the two? Can you characterize which acyclic rule bases fall outside the scope of Proposition 5.4, and how common are such cases in practice?
>
> For a given rule base, it is straightforward to verify whether it satisfies the conditions of Proposition 5.4. We need to verify, for each relation r, that it is either regular or balanced. To check the former, we can first construct for each relation $r$ a corresponding set $D_r$ where $s\in D_r$ if the rule base entails some rule $s_1\circ …\circ s_n \subseteq r$ with $s\in \{s_1,...,s_n\}$. These sets $D_r$ can be straightforwardly constructed in polynomial time. Then $r$ is regular if for every rule $r_1\circ …\circ r_m\subseteq r$ in the rule base, we have that $D_{r_i}\cap D_{r_j}=\emptyset$ for $i\neq j$. Testing whether $r$ is balanced involves testing the feasibility of a system of linear equations (one for each rule with a relation from $D_r$ in the head). To make this clearer, we will add these procedures to the paper.
>
> It is likely that there are rule bases which do not satisfy these conditions but which can still be faithfully captured using accordant convex coordinate-wise embeddings. Whether conditions can be found that are both sufficient and necessary remains an open question.
>
> > Q3 Proposition 6.1 demonstrates that spurious semantic dependencies can be unavoidable under specific conditions. Do these conditions arise naturally in real knowledge graphs, or are they pathological? Is the result a special case of a more general impossibility theorem?
>
> The fundamental limitation of working with convex regions arises from Helly’s theorem. For the specific case of two-dimensional regions (which is what we are working with in coordinate-wise models), we simply have that the convexity requirement induces some inequalities, which may be inconsistent (as in the proof of Proposition 6.1). We will clarify this point.
>
> We indeed assume that the conditions under which spurious inferences occur are extremely rare in most domains, as they require a relatively large set of facts which are closely interrelated, in a specific way. Empirically analysing how common this issue is in practice is not straightforward, as we have no ground truth rule bases for most KGs. We also believe such an analysis would be outside the scope of this paper.
>
> > Q4 The variance across runs is extremely high throughout Table 2. Have you investigated whether this reflects sensitivity to initialization, instability in the optimization landscape, or something specific to the FAIRE benchmark design? Does averaging across such unstable runs produce meaningful conclusions?
>
> Please see our responses to Reviewer WNA9 (Q1) and Reviewer gikF (Q3).

---

### Official Review · Reviewer_gikF · 2026-03-13

**Soundness:** 3
**Presentation:** 3
**Significance:** 3
**Originality:** 3
**Overall Recommendation:** 4
**Confidence:** 4

**Summary:**

This paper investigates the expressive power of convex coordinate models for relational reasoning. The core theoretical result formally proves that imposing a convex structure intrinsically bounds model expressivity. The authors establish positive results showing that specific convex models can capture bounded acyclic rules under regular or balanced conditions. Besides, the empirical evaluation introduces a new synthetic benchmark named FAIRE to test whether models can avoid spurious inferences. The experiments reveal that these models still struggle to reliably learn and faithfully capture these rule bases in practice.

**Compliance With Llm Reviewing Policy:**

Affirmed.

**Key Questions For Authors:**

1.	Proposition 6.1 shows convex coordinate models capture spurious dependencies even when faithful to closed path rules. Does this imply strictly faithful reasoning requires cross coordinate comparisons or can non convex coordinate models fully resolve this?

2.	Your experiments use 10 dimensional embeddings which is theoretically sufficient but may cause optimization issues. Could you provide additional results using higher dimensions like 50 or 100 to check if the poor empirical performance stems from optimization landscapes rather than fundamental architectural flaws?

3.	Training with standard binary cross entropy and random negative sampling might not provide the precise gradients needed to sculpt exact convex regions against FAIRE distractor rules. Could you add an experiment using hard negative mining targeting these specific distractors to see if explicit penalization improves empirical performance?

**Limitations:**

yes

**Strengths And Weaknesses:**

**Strengths:**

1.	The perspective of analyzing expressivity specifically through the lens of faithful rule reasoning is a novel and highly relevant angle for the neurosymbolic AI and knowledge graph embedding communities. The goal of capturing desired rules while strictly avoiding spurious inferences is highly valuable.

2.	The theoretical claims are mathematically rigorous and well supported by extensive proofs in the appendix. The formalization of faithful capturing provides a solid and reusable framework for future theoretical analyses of embedding models.

3.	The findings address an important limitation in current highly efficient coordinate models. By clearly demarcating what can and cannot be learned, the paper provides valuable guidance for future architecture designs.


**Weaknesses:**

1.	While the FAIRE benchmark is a great addition, the empirical results in Table 2 show extremely high variance and a general failure of the theoretical polygon models. The paper attributes this to standard strategies being insufficient, but a deeper empirical investigation into why the optimization fails would significantly strengthen the bridge between theory and practice.

2.	Some of the theoretical conditions such as the balanced relations in Definition 5.3 are a bit abstract. While Example 5.5 helps, a brief discussion on how common or restrictive these conditions are in real world knowledge graph ontologies would improve readability and practical context.

---

> ### Author Rebuttal · Authors · 2026-03-30
>
> We thank the reviewer for their careful reading of the paper and constructive feedback.
>
> > Q1 Proposition 6.1 shows convex coordinate models capture spurious dependencies even when faithful to closed path rules. Does this imply strictly faithful reasoning requires cross coordinate comparisons or can non convex coordinate models fully resolve this?
>
> Proposition 6.1 only applies to convex coordinate-wise embeddings. The construction from the proof of Proposition 4.1 can be used to avoid spurious inferences when non-convex (coordinate-wise) regions are allowed. Similarly, from the results in the ReshufflE paper, we know that spurious facts can be avoided by using cross-coordinate comparisons. We will clarify these points in the paper.
>
> > Q2 Your experiments use 10 dimensional embeddings which is theoretically sufficient but may cause optimization issues. Could you provide additional results using higher dimensions like 50 or 100 to check if the poor empirical performance stems from optimization landscapes rather than fundamental architectural flaws?
>
> Increasing dimensionality does not consistently improve results. In the table below, we show the difference in hits@1 between d=10 and d=50, averaged over the 7 datasets. The difference is positive if hits@1 is higher with d=50. We also note that while the polygon models have marginally better results on average, they each have lower hits@1 for d=50 on at least one dataset. Increasing dimensionality only consistently benefits ExpressivE.
>
> | Model | Mean | Min | Max |
> | --- | --- | --- | --- |
> | Octagons | -0.02 | 0.04 | -0.12 |
> | ExpressivE | 0.08 | 0.10 | 0.03 |
> | Polygons (2) | 0.00 | 0.05 | -0.05 |
> | Polygons (4) | 0.04 | 0.09 | 0.02 |
> | Polygons (6) | 0.04 | 0.13 | -0.05 |
> | Polygons (8) | 0.03 | 0.10 | 0.01 |
> | Polygons (10) | 0.02 | 0.05 | -0.03 |
>
> > Q3 Training with standard binary cross entropy and random negative sampling might not provide the precise gradients needed to sculpt exact convex regions against FAIRE distractor rules. Could you add an experiment using hard negative mining targeting these specific distractors to see if explicit penalization improves empirical performance?
>
> We reran the experiments with a number of changes. First, we now use a “1-vs-all” setting, where all candidate entities are scored for a query $r(e, ?)$. This has a similar effect as the reviewer’s suggestion and prevents rule instances from overlapping in the embedding space. Second, we noticed that early stopping was applied rather aggressively, so we now use an increased value for the patience hyperparameter. Finally, we now test configurations with up to 10 edges, and we report results that are aggregated over 20 runs instead of 5 (showing average, minimum and maximum values).
>
> | Model | Perm2 | Perm3 | Mix2 | Mix3 | Rep12 | Rep13 | Comb |
> | --- | --- | --- |--- | --- | --- |--- | --- |
> | Octagons | 1.000 (1.000-1.000) | 0.575 (0.000-1.000) | 0.466 (0.000-1.000) | 0.250 (0.030-0.480) | 0.417 (0.110-0.580) | 0.603 (0.350-1.000) | 0.170 (0.000-0.500) |
> | ExpressivE | 0.994 (0.920-1.000) | 0.970 (0.700-1.000) | 0.935 (0.820-1.000) | 0.709 (0.240-1.000) | 0.583 (0.380-0.850) | 0.693 (0.300-1.000) | 0.320 (0.150-0.610) |
> | Polygons (2) | 0.957 (0.440-1.000) | 0.600 (0.260-0.940) | 0.774 (0.290-1.000) | 0.500 (0.230-0.960) | 0.368 (0.200-0.550) | 0.530 (0.410-0.620) | 0.281 (0.080-0.820) |
> | Polygons (4) | 0.990 (0.800-1.000) | 0.745 (0.200-1.000) | 0.763 (0.200-1.000) | 0.446 (0.220-0.730) | 0.398 (0.240-0.540) | 0.620 (0.440-0.870) | 0.347 (0.140-0.650) |
> | Polygons (6) | 1.000 (1.000-1.000) | 0.686 (0.000-1.000) | 0.779 (0.230-1.000) | 0.574 (0.350-0.820) | 0.349 (0.140-0.610) | 0.509 (0.200-0.850) | 0.402 (0.030-0.990) |
> | Polygons (8) | 0.972 (0.600-1.000) | 0.714 (0.220-1.000) | 0.784 (0.260-0.980) | 0.438 (0.220-0.760) | 0.421 (0.160-0.620) | 0.629 (0.340-0.900) | 0.337 (0.020-0.690) |
> | Polygons (10) | 1.000 (1.000-1.000) | 0.893 (0.560-1.000) | 0.744 (0.290-0.970) | 0.472 (0.170-0.920) | 0.407 (0.240-0.680) | 0.532 (0.380-0.810) | 0.436 (0.080-0.880) |
>
> For the hardest dataset (Comb), we now see that polygons clearly outperform ExpressivE and octagons. Another finding is that the optimal number of edges differs across datasets (and should therefore be treated as a hyperparameter in future work based on polygon embeddings). However, results remain highly unstable, consistent with our findings in the paper.

---

> > ### Author Rebuttal · Reviewer_gikF · 2026-04-03
> >
> > I thank the authors for their answer to my question, and maintain my initial score.

---

### Decision · Program_Chairs · 2026-04-30

**Decision:**

Accept (regular)

**Comment:**

The paper studies the expressivity of convex coordinate-wise region-based embeddings for relational reasoning over knowledge graphs.  The core theoretical result formally proves that imposing a convex structure intrinsically bounds model expressivity and show that specific convex models can capture bounded acyclic rules under regular or balanced conditions. A new synthetic benchmark is also introduced to test whether models can avoid spurious inferences.

The paper recieved 4 reviews with 3 accepting and 1 tending towards rejection with scores ranging from 3 to 6. The reviews raised several concerns with the most important being the experimental results showing high variance casting doubt on the robustness of the polygon models. AS reviewer TJR1 pointed out and I quote "The authors prove (Proposition 5.4) that the polygon model is expressively sufficient to achieve perfect hits@1 on all FAIRE datasets under the right conditions. Yet Table 2 shows the polygon model is often the worst performing model, sometimes barely above random chance."

The rebuttal did provide some answers but the authors agree that there are stability issues in the learning process and thus although some robust polygon solutions are found these are atleast not guarenteed empirically and leave this for future work. I think this is a big issue since a strong theoretical claim should not be dependent on such constraints atleast when not explicitly mentioned in the theoretical proof. Overall, I like the paper, think that the theory is strong and thus recommend a weak acceptance (Weak due to the empirical unstability). I leave it to the PCs to either accept or reject the paper.